# ProFiT: Unsupervised Fine-Tuning of Tabular Models via Proxy Tasks for Label-Scarce Anomaly Detection

## Abstract

Anomaly detection in tabular data is crucial for applications such as fraud prevention and risk control, yet it remains challenging due to heterogeneous features, class imbalance, and limited labeled anomalies. Although pretrained tabular in context learning (TICL) models reduce label dependence, the inductive biases they develop on synthetic tasks are often misaligned with the actual data distributions encountered in downstream scenarios. Effective adaptation to new domains is thus difficult when labels are scarce. We propose **ProFiT**, an unsupervised fine-tuning framework that leverages only unlabeled target-domain data to adjust pretrained tabular models. ProFiT constructs a variety of proxy tasks by sampling different features as targets and using correlated features as inputs, encouraging the model to capture the underlying structure of the new data. To improve training effectiveness, we introduce a consistency regularizer that aligns the predictions from two different proxy views using Jensen–Shannon divergence. Experiments on tabular anomaly detection benchmarks show that ProFiT outperforms weakly-supervised and unsupervised methods, as well as vanilla TICL models. ProFiT offers a practical way to improve tabular anomaly detection under limited labeled data conditions and vast amounts of unlabeled data.

## 1 Introduction

Anomaly detection (Chandola et al., 2009; Li et al., 2023) in tabular data plays a central role in numerous high-stakes applications, ranging from fraud detection and financial risk control (Dornadula & Geetha, 2019) to anti-crawling systems and cybersecurity monitoring (Lazarevic et al., 2003). In these domains, the ability to identify anomalous behaviors within high-dimensional structured data is critical to ensuring business security. Despite its practical importance, tabular anomaly detection remains challenging due to multiple factors, including the heterogeneity of feature types, the imbalance of class distributions, and the scarcity of reliable labels. Moreover, anomaly patterns are often rare and context-dependent, which further complicates the construction of robust detection systems that can generalize to evolving environments (Aggarwal, 2015; Zong et al., 2018; Pang et al., 2021b). These challenges demand approaches that are both data-efficient and adaptable, while maintaining robustness under real-world operational constraints.

Supervised methods that treat anomaly detection as binary classification are commonly used (Han et al., 2022). Tree-based pipelines (e.g., Random Forests (Breiman, 2001), XGBoost (Chen & Guestrin, 2016)) often deliver strong performance for tabular data when trained with sufficient in-domain labels. However, in practice, anomalous labels are exceedingly scarce and their annotation is costly, whereas vast amounts of unlabeled data are typically available. Recent pretrained tabular in context learning models , such as TabPFN (Hollmann et al., 2022) and MotherNet (Mueller et al., 2025), adopt a meta-learning paradigm in which the models are trained on a large collection of synthetic datasets. Through this pretraining process, they acquire transferable inductive biases, allowing them to generalize to new downstream tasks with only few-shot labeled samples. However, their generalization to downstream anomaly detection tasks remains limited, as the pretrained representations often mismatch the target distribution, thereby necessitating additional fine-tuning for effective adaptation. Unfortunately, this requirement conflicts with the scarcity of high-quality labels in practice, and few existing studies have explored how to leverage the abundance of unlabeled

data to fine-tune such tabular in context learning pretrained models, enabling them to better capture downstream task-specific distributions

In this paper, we address this gap by proposing a novel **pro**xy-task-based unsupervised **fi**ne-**t**uning framework (**ProFiT**) for tabular anomaly detection. Instead of relying on labeled anomalies, ProFiT leverages the intrinsic structure of tabular data to construct a diverse set of predictive proxy tasks. Specifically, we randomly designate one feature as the prediction target and sample correlated subsets of the remaining features as inputs, thereby generating large-scale heterogeneous prediction tasks. These tasks serve as proxy tasks for the downstream anomaly detection problem, enabling the model to learn the underlying distributional characteristics required for effective generalization. We provide theoretical support that clarifies why ProFiT is effective. Under a standard latent factor model, we establish a regret identity and a finite-sample transfer bound showing that minimizing the task-averaged proxy risk controls the excess risk to Bayes on unseen downstream labels. To further enhance effectiveness, we propose a consistency regularization strategy. For the same prediction target, we construct two distinct proxy subsets of input features and encourage consistency between their predictions using Jensen–Shannon divergence. We evaluate ProFiT on multiple benchmark anomaly detection datasets, where it achieves superior performance compared to existing methods.

Our contributions can be summarized as follows:

- We propose ProFiT, a proxy task fine-tuning framework for tabular anomaly detection that leverages unlabeled data to construct proxy tasks, enabling the model to capture distributional structure without relying on annotated anomalies.
- We establish a regret identity and a finite sample transfer bound, showing that minimizing the average proxy risk across tasks effectively controls excess risk on unseen downstream labels.
- We introduce a consistency regularization strategy to enhance training effectiveness, and demonstrate through extensive experiments on benchmark datasets that ProFiT outperforms existing SOTA methods.

## 2 RELATED WORK

**Anomaly Detection**   Current popular deep anomaly detection on tabular data methods are unsupervised approach (Pang et al., 2021a). These methods typically relies on distance- or density-based scoring (e.g., KNN (Ramaswamy et al., 2000), LOF (Breunig et al., 2000)) and one-class classification (e.g., iForest (Liu et al., 2008), OCSVM (Schölkopf et al., 1999)), with deep variants (e.g., DeepSVDD (Ruff et al., 2018), DIF (Xu et al., 2023a)) improving high-dimensional feature extraction. However, these methods hinge on strong priors and effectively model only the normal class, lacking any guidance about anomalies; as a result, performance plateaus when anomaly semantics are context-dependent or data are contaminated (Shou et al., 2025).

To bridge this gap, weakly-supervised anomaly detection assumes a small set of labeled anomalies amid abundant unlabeled data (Durani et al., 2025). Early hybrids like XGBOD (Zhao & Hryniewicki, 2018) convert unsupervised scores into meta-features for a downstream classifier, while end-to-end approaches learn anomaly-aware representations directly: DevNet (Pang et al., 2021a) regularizes unlabeled scores toward a Gaussian prior and enlarges known anomalies, DeepSAD (Ruff et al., 2020) pushes labeled anomalies away from a normal hypersphere, and FeaWAD (Zhou et al., 2022) applies weak supervision in an autoencoded latent space. To cope with extremely sparse labels and noise, PReNet (Ren et al., 2019) iteratively self-trains on pseudo-labels, RoSAS (Xu et al., 2023b) uses robust continuous supervision, a dual-kernel design enforces compactness vs. separation with light- and heavy-tailed kernels (Durani et al., 2025), and READ (Shou et al., 2025) frames subset selection as reinforcement learning to emphasize boundary normals and suspected anomalies. In this work, we adopt this setting and operate with only a handful of labeled anomalies.

**Tabular In Context Learning Model**   Tabular in context learning (TICL) frames tabular prediction builds on the principles of meta-learning. The model learns from a large collection of meta-tasks, each composed of a support set and a query set. This process can be viewed as learning a mapping from "task to prediction" across a wide range of heterogeneous meta-tasks. each with a small support set and a query set, thereby acquiring cross-task transferability. As a result, the model can rapidly adapt to a target task with only a few samples during inference.

A variety of approaches have been proposed under this paradigm. TabPFN (Hollmann et al., 2022) encodes each row of a table as a token and predicts query labels by modeling attention among tokens. Building on this, TabPFN v2 (Hollmann et al., 2025) introduces two-way attention to simultaneously capture feature-wise and instance-wise interactions. To handle large-scale datasets, TabFlex (Zeng et al., 2025) replaces the softmax attention in TabPFN with linear attention, enabling scalability to larger data regimes. Another line of research explores hypernetwork architectures (Ha et al., 2017). Methods such as HyperFast (Bonet et al., 2024) and MotherNet (Mueller et al., 2025) train a hypernetwork via meta-learning to generate a set of MLP parameters for each task. At inference time, predictions are obtained by a simple forward pass through the generated MLP, which greatly improves efficiency.

Despite these advances, existing methods typically rely on constructing a large number of diverse meta-tasks during pretraining, which in turn requires extensive labeled data. To mitigate this requirement, many studies employ synthetic tabular datasets for training. However, this inevitably introduces distributional discrepancies with downstream tasks, a challenge that becomes particularly acute in anomaly detection. In this setting, the data distribution is highly imbalanced and labeled samples are scarce. Consequently, how to effectively fine-tune TICL models in an unsupervised manner remains an underexplored yet crucial research direction.

## 3 METHOD

### 3.1 PROBLEM SETTING

We focus on weakly-supervised anomaly detection in tabular data. Let each sample in the feature space $\mathbf{x} = [x^{(1)}, \ldots, x^{(d)}] \in \mathcal{X} \subset \mathbb{R}^d$. The training dataset consists of a large unlabeled set $\mathcal{D}_U = \{\mathbf{x}_1, \ldots, \mathbf{x}_N\}$ and a small labeled anomaly set $\mathcal{D}_L = \{(\mathbf{x}_{N+1}, 1), \ldots, (\mathbf{x}_{N+K}, 1)\}$ with $K \ll N$. Our objective is to learn a scoring function $f_\theta : \mathcal{X} \to [0, 1]$, which assigns higher scores to anomalous instances in unseen test dataset.

### 3.2 FRAMEWORK OVERVIEW

Our method integrates proxy-based fine-tuning with tabular in context learning. As illustrated in Figure 1, the framework has three stages:

- **Proxy Task Sampling.** From the training data, we construct proxy tasks by conditioning on a feature subset $\mathbf{X}_S \in \mathbb{R}^{S \times d}$ and predicting the held-out feature $\mathbf{X}_{:,t} \in \mathbb{R}^d$. Detailed sampling strategies are provided in subsection 3.5. To unify the learning objective, we treat the prediction of $\mathbf{X}_{:,t}$ as a classification task. For categorical columns, we directly utilize the original class labels as targets. For numerical columns, following the methodology in MotherNet, we perform discretization by sorting the values, randomly selecting quantiles as boundaries, and binning the data to convert continuous values into categorical indices. These tasks expose cross-feature dependencies and provide transferable supervision.

- **Proxy-Based Fine-tuning.** For each proxy target, multiple predictor subsets yield diverse tasks. Support and query sets are drawn from the same samples across these tasks, enabling in-context training. As detailed in subsection 3.6 and algorithm 1, the model is optimized with a classification loss for proxy prediction and a consistency loss to align different tasks sharing the same prediction target.

- **Inference.** During inference, we follow the weakly supervised anomaly detection setting described above. As shown in Algorithm 5, the labeled anomalies in $\mathcal{D}_L$ are directly incorporated into the few-shot support set used to condition the fine-tuned TICL model. Specifically, we first apply the unsupervised detector $\mathcal{A}$ to all instances in $\mathcal{D}_U$ to obtain anomaly scores and identify a subset of samples that are deemed normal. We then uniformly sample $K$ instances from this normal subset and treat them as pseudo-labeled normal data. These pseudo-normals are then combined with the labeled anomalies to construct the support set $S = \mathcal{D}_L \cup \{x_u^{(0)}, 0\}_{u=1}^K$. Feeding this support set into the fine-tuned TICL model $T_\theta$, the model generates the downstream MLP parameters, which are then used to evaluate $f_\theta$ for each test instance.

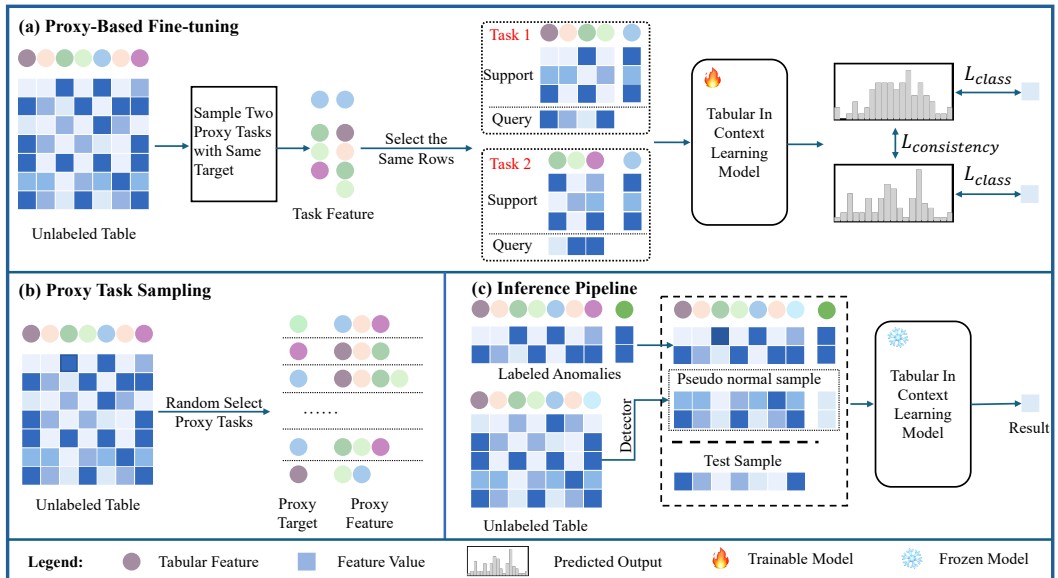

Figure 1: Overview of ProFiT, covering both training and inference. (a) Training pipeline: Given unlabeled tabular data, one feature is randomly selected as the prediction target, while two partially overlapping feature subsets are sampled from the remaining features to construct proxy tasks. For each task, rows are sampled to form paired inputs. Cross-entropy loss is applied to ensure prediction accuracy, and a Jensen–Shannon divergence term enforces consistency between the two proxy predictions. (b) Proxy-task sampling: Target features are selected based on inter-sample correlations to avoid trivial shortcuts or unlearnable cases caused by irrelevant features. (c) Inference stage: A small set of labeled samples is used as positives, while negatives are sampled from the unlabeled pool to form a support set. The model then predicts the downstream task using this constructed support-query setting.

Our core idea is to learn latent representations associated with the raw data through proxy tasks. By averaging over a diverse set of such tasks, the resulting representation preserves sufficient latent information to generalize to downstream tasks with *unknown but predictable* labels, thereby offering both practical effectiveness and theoretical guarantees for anomaly detection. In subsection 3.3 and subsection 3.4, we present our theoretical analysis, offering guarantees for the proposed method under limited-sample scenarios.

### 3.3 Consistency and Generalization of the Proxy-Task Framework

Let $\mathbf{X} \in \mathbb{R}^{n \times d}$ be the training data and $\mathcal{D} = \{1, \ldots, d\}$ denote the set of all column indices. A *proxy-task* is defined by a triple $\tau = (\mathbf{X}, S, t)$ where $S \subset \mathcal{D}$ is a feature index set and $t \in \mathcal{D} \setminus S$ is a target index. For a given proxy-task $\tau$, we denote

$$\mathbf{X}_S \in \mathbb{R}^{n \times |S|}, \qquad \mathbf{y}_t := \psi_t(\mathbf{X}_{:,t}) \in \mathbb{R}^n,$$

where $\psi_t$ is a transformation function applied to the $t$-th feature: it is the identity mapping if $\mathbf{X}_{:,t}$ is categorical, and a rank-based (quantile) discretizer if $\mathbf{X}_{:,t}$ is continuous. A downstream task is denoted $\tau^\star = (\mathbf{X}, S, t^\star)$ with $t^\star \notin \mathcal{D}$, representing an *unknown but predictable* label not included in the feature set. We assume proxy-tasks are drawn from a distribution $Q$ over index pairs $(S, t)$.

To enable this transfer across labels, we adopt a standard multi-view latent-factor assumption. Under this assumption, there is a latent factor (or factors) shared among multiple "views" (e.g. feature subsets or labels) such that learning in one view carries over to others.

**Assumption 3.1** (Latent factor model (Shanmugam, 2001) for task). There exists a latent $\mathbf{u} \in \mathcal{U} \subset \mathbb{R}^r$, where $r \leq d$ such that, conditional on $\mathbf{u}$, columns are independent: $\mathbf{X}_i \perp \mathbf{X}_{-i} \mid \mathbf{u}$. Each column satisfies $\mathbf{X}_i = g_i(\mathbf{u}, \boldsymbol{\epsilon}_i)$ with independent noise $\boldsymbol{\epsilon}_i$. Specifically, for any $\tau = (\mathbf{X}, S, t)$, we have $\mathbf{y}_t \perp \mathbf{X}_S \mid \mathbf{u}$.

**Assumption 3.2** (Downstream compatibility). The unseen downstream label is also generated from $\mathbf{u}$ and $\mathbf{y}^\star \perp \mathbf{X}^\star \mid \mathbf{u}$.

**Remark 3.3.** These two assumptions play complementary roles: (i) Assumption 3.1 defines a shared latent space $\mathcal{U}$; (ii) Assumption 3.2 links downstream labels to the same $\mathcal{U}$. Together, they justify that solving diverse proxy-tasks learns a representation transferable to unseen labels.

To further elucidate the generalization mechanism of our proxy tasks, we now make precise how solving diverse proxy tasks enables transfer. We start with a excess risk identity that connects representation quality to conditional mutual information, thereby formalizing the generalization mechanism of proxy tasks.

**Lemma 3.4.** *Let $f$ be a representation encoder and consider a fixed proxy-task $\tau = (\mathbf{X}, S, t)$. Under log-loss, the Bayes-optimal risk of any predictor $h$ on $f(\mathbf{X}_S)$ equals the conditional entropy:*

$$\inf_h \ \mathbb{E}\big[ -\log h(f(\mathbf{X}_S))[\mathbf{y}_t]\big] = H(\mathbf{y}_t \mid f(\mathbf{X}_S)).$$

*Define the excess risk of $f$ on $\tau$ by*

$$\Delta_f(\tau) := H(\mathbf{y}_t \mid f(\mathbf{X}_S)) - H(\mathbf{y}_t \mid \mathbf{X}_S).$$

*where $H(\cdot)$ denotes Shannon entropy and $I(\cdot, \cdot)$ denotes mutual information. Then $\Delta_f(\tau) = I(\mathbf{y}_t; \mathbf{X}_S \mid f(\mathbf{X}_S)) \geq 0$, under Assumption 3.1, $\Delta_f(\tau) = I(\mathbf{y}_t; \mathbf{u} \mid f(\mathbf{X}_S))$.*

Averaging across tasks $\tau \sim Q$, we obtain the task-averaged excess:

$$\overline{\Delta}_f \ := \ \mathbb{E}_{\tau \sim Q}\big[\Delta_f(\tau)\big] = \mathbb{E}_{\tau \sim Q} I(\mathbf{y}_t; \mathbf{u} \mid f(\mathbf{X}_S)). \tag{1}$$

Minimizing $\overline{\Delta}_f$ in the training of proxy task therefore encourages $f(\mathbf{X})$ to be a sufficient statistic for the latent factor $\mathbf{u}$. Since downstream labels also depend on $\mathbf{u}$ (Assumption 3.2), the learned representation is naturally suited for transfer. We now formalize this intuition, for any downstream task $\tau^\star = (\mathbf{X}, S^\star, t^\star)$, let $h_f^\star$ denote a predictor that achieves the lowest possible error of the learned representation $f(\mathbf{X}_{S^\star})$, then:

**Theorem 3.5** (Sufficiency-driven transfer). *Under Assumption 3.1 and Assumption 3.2,*

$$\mathcal{R}_{\tau^\star}(f, h_f^\star) - \mathcal{R}_{\tau^\star}^{\text{Bayes}} \ = \ \Delta_f(\tau^\star) \ \leq \ \Gamma(Q, \tau^\star) \cdot \overline{\Delta}_f, \tag{2}$$

*where the term $\mathcal{R}\tau^{\star\text{Bayes}}$ is the Bayes risk, the theoretical minimum error achievable for task $\tau^\star$, representing the task's inherent difficulty. $\Gamma(Q, \tau^\star) \geq 1$ is a compatibility constant measuring how well the proxy-tasks sampled from $Q$ align with the latent directions relevant to $t^\star$.*

In short, Theorem 3.5 shows that a representation minimizing proxy-task risk generalizes reliably to downstream tasks, with at most a bounded loss gap. Appendix C contains the proofs of Lemma 3.4 and Theorem 3.5.

### 3.4 TRANSFER BOUNDS FOR FINITE SAMPLE TASKS

The population result in subsection 3.3 shows that task-averaged sufficiency of the representation transfers to unseen labels. We now analyze the realistic and finite sample setting to guide the design of our method. We posit that each proxy target is generated by a (possibly nonlinear) factor model

$$\mathbf{y}_t \ = \ g_t(\mathbf{u}) \ + \ \boldsymbol{\epsilon}_t, \tag{3}$$

where $g_t : \mathbb{R}^r \to \mathbb{R}^d$ is differentiable and $\boldsymbol{\epsilon}_t$ specifies the observation model $p_t(\mathbf{y} \mid g_t(\mathbf{u}))$. Let $\mathbf{F}_t(\mathbf{u})$ denote the Fisher information matrix (FIM) with respect to $\mathbf{u}$ for task $t$. We summarize information coverage across proxy tasks by the task and latent averaged FIM

$$M_Q \ := \ \mathbb{E}_{\tau \sim Q, \, \mathbf{u} \sim p(\mathbf{u})}\big[\mathbf{F}_t(\mathbf{u})\big], \tag{4}$$

and write $\mu := \lambda_{\min}(M_Q)$. The scalar $\mu > 0$ quantifies coverage: larger $\mu$ means no latent direction is systematically neglected by $Q$.

**Theorem 3.6** (Transfer in Non-linear Models). *If the Average FIM is positive definite, i.e. $M_Q \succeq \mu I_r$ with $\mu > 0$, then for any downstream task $\tau^\star$,*

$$\Delta_f(\tau^\star) \;\leq\; \frac{1}{\mu}\, \overline{\Delta}_f. \tag{5}$$

*In words, the worst-case gap of the learned representation on any unseen label is controlled by the task-averaged excess on proxy-tasks, scaled by $1/\mu$. The constant $1/\mu$ is a concrete counterpart of the abstract transfer constant in Theorem 3.5.*

Theorem 3.6 controls the *representation-induced* part of the downstream excess. In practice, however, the proxy observation models and the downstream one may have different Bayes risks (irreducible noise). To make this explicit, decompose

$$\mathrm{disc}(Q, \tau^\star) := \big| H_{\tau^\star}(\mathbf{y}^\star \mid \mathbf{u}) \;-\; \mathbb{E}_{\tau \sim Q} H_\tau(\mathbf{y}_t \mid \mathbf{u}) \big|.$$

This term is zero when observation models are matched or calibrated in difficulty. We now pass from distribution-level quantities to their empirical counterparts. Training on $M$ proxy tasks with $n$ samples by empirical risk minimization yields parameters $\widehat{\theta}$ and encoder $\widehat{f}$, the following theorem gives the finite-sample analogue of our transfer bound.

**Theorem 3.7** (End-to-end transfer under nonlinear mechanisms (finite-sample)). *With probability at least $1 - \delta$,*

$$\mathcal{R}_{\tau^\star}(\widehat{f}, \widehat{h}^\star) - \mathcal{R}_{\tau^\star}^{\mathrm{Bayes}} \;\leq\; \frac{1}{\mu}\Big( \underbrace{\widehat{L}_Q(\widehat{\theta}) - \widehat{H}_Q}_{\textit{Empirical Excess}} + \underbrace{\mathrm{Gen}(M, n, \delta)}_{\textit{Generalization Gap}} \Big) + \underbrace{\mathrm{disc}(Q, \tau^\star)}_{\textit{Task Mismatch}}. \tag{6}$$

*Were, $\widehat{L}_Q - \widehat{H}_Q$ is the empirical proxy excess (empirical log-loss minus the empirical conditional-entropy baseline $\widehat{H}(\mathbf{y}_t \mid \mathbf{X}_S)$, both averaged over tasks/samples); $\mathrm{Gen}(M, n, \delta)$ is a high-probability ($\geq 1 - \delta$) bound on the distribution–empirical gap for the proxy risk.*

The proofs of Theorem 3.6 and Theorem 3.7 can be found in Appendix C.

## 3.5 SAMPLING STRATEGY

To construct an informative and compact feature set of proxy task, we adopt a sampling strategy based on the *minimum Redundancy and Maximum Relevance (mRMR)* principle. Across tasks, target coordinates $t \in \mathcal{D}$ are sampled approximately uniformly over $\mathcal{D}$; for a fixed target coordinate $t \in \mathcal{D}$ and candidates $\tilde{\mathcal{D}} = \mathcal{D} \setminus \{t\}$. Let $\mathbf{C} \in \mathbb{R}^{d \times d}$ be the absolute correlation matrix between features (with $\mathbf{C}_{i,i} = 0$). We build a compact proxy set $S \subseteq \tilde{\mathcal{D}}$ of size $k = \kappa(d)$ by a greedy mRMR procedure:

- **Initialization:** The first feature added to $S$ is the one most strongly correlated with the target

$$v^* = \arg\max_{v \in \tilde{\mathcal{D}}} \mathbf{C}_{v,t}.$$

- **Greedy step:** For each $v \in \tilde{\mathcal{D}} \setminus S$, score it according to

$$\mathrm{score}(v) = \mathbf{C}_{v,t} - \frac{1}{|S|} \sum_{u \in S} \mathbf{C}_{v,u},$$

  and add the $v$ with the largest score to $S$. Repeat until $|S| = k$ or no candidates remain. Where the first term reflects the relevance to the target, and the second term penalizes redundancy with the already selected features.

- **Fill (optional):** If $|S| < k$, sample the remainder uniformly from $\tilde{\mathcal{D}} \setminus S$.

The resulting subset $S$ maintains high target relevance while suppressing internal redundancy, thereby promoting diversity and informativeness of the constructed proxy tasks. In practice, increasing the number of sampled proxy tasks $M$ tightens the end-to-end transfer bounds via concentration and averaging effects: (i) the empirical average information $\widehat{M}_Q$ concentrates to $M_Q$, which increases the observed coverage $\widehat{\mu} = \lambda_{\min}(\widehat{M}_Q)$ and stabilizes transfer and (ii) this sampling reduces task mismatch $\mathrm{disc}(Q, \tau^\star)$ by aligning proxy-task difficulty via more predictable and calibrated views of $t$. Together, these effects reduce the gap to the Bayes risk.

### 3.6 TRAINING OBJECTIVE

We optimize a two-part objective tailored to proxy-task adaptation: a *cross entropy supervision loss* and a *cross-subset consistency loss* measured by the Jensen–Shannon (JS) divergence. These two components are controlled by distinct coefficients, respectively capturing label alignment and representation stability across feature projections.

**Cross-entropy on proxy tasks**  For any proxy-task sample $(\mathbf{x}_{S_j}, \tilde{y}_i^{(t)}, S_j) \in (\mathbf{X}, S, t)$, the classification cross-entropy is

$$L_Q^{\text{CE}}(\theta) := \frac{1}{M} \sum_{j=1}^{M} \frac{1}{n} \sum_{i=1}^{n} \ell_{\text{CE}}(\mathbf{x}_i, \tilde{y}_i^{(t)}, S_j; \theta), \tag{7}$$

which directly aligns predictions with proxy labels and contracts the conditional entropy at the task level. Here we instantiate $\widehat{L}_Q$ in Theorem 3.7 by the CE risk, so $\widehat{L}_Q^{\text{CE}}(\theta) - \widehat{H}_Q$ is exactly that term; since $\widehat{H}_Q$ is constant w.r.t. $\theta$, minimizing $\widehat{L}_Q^{\text{CE}}$ is equivalent to minimizing the empirical excess.

**Cross-subset consistency via JS divergence**  We minimize the expected JS divergence between two logits (predictive distributions) for the same $(\mathbf{x}_i, t)$ under independently sampled subsets $S_1, S_2$:

$$L_Q^{\text{JS}}(\theta) = \frac{1}{M} \sum_{j=1}^{M} \frac{1}{n} \sum_{i=1}^{n} \mathbb{E}_{S_1, S_2 \sim \mathcal{S}(t_j)} \Big[ D_{\text{JS}}\big(\hat{p}_\theta(\cdot \mid \mathbf{x}_i, S_1) \,\|\, \hat{p}_\theta(\cdot \mid \mathbf{x}_i, S_2)\big) \Big].$$

This consistency regularizer explicitly controls the variability of predictions under view perturbations (changing $S$), improving algorithmic stability and prediction smoothness. In the finite-sample bound of Theorem 3.7, this translates into a tighter generalization gap term, i.e., it reduces $\text{Gen}(M, n, \delta)$.

The final empirical risk minimization objective during training is

$$\mathcal{L}(\theta) = \lambda_{\text{CE}} L_Q^{\text{CE}} + \lambda_{\text{JS}} L_Q^{\text{JS}}. \tag{8}$$

Here $\lambda_{\text{CE}}, \lambda_{\text{JS}} > 0$ independently modulate the two losses. This design jointly lowers conditional risk on proxy tasks, improves robustness to feature-subset perturbations, thereby enhancing transfer to unseen labels and distributions downstream. The full training algorithm is presented in Appendix D.

## 4 EXPERIMENTS

**Datasets**  We select 35 real world tabular datasets widely used in anomaly detection tasks, Sourced from ODDS (Rayana, 2016) and ADbench (Han et al., 2022). These datasets span various domains, including healthcare, internet services, finance, etc. They feature a combination of numerical and categorical attributes and exhibit diverse statistical properties, with sizes ranging from 129 to 619,326 samples, dimensions from 3 to 1,555, and anomaly ratios from 0.03 % to 39.91 %. A detailed statistical summary for each dataset can be found in Table 3, which shows the number of samples, the dimension, and the number of anomalies of each dataset used.

**Evaluation Metrics**  For our evaluation protocol, we follow the settings RoSAS (Xu et al., 2023b). Each dataset is partitioned into training and test subsets at a 7:3 ratio. A constraint is imposed on the training data, where the number of {5,10,20,30} labeled anomalies are utilized; should the number of available anomalies be less than this threshold, all are included. Evaluation is performed on the held-out test set and the performance of the models is assessed based on two primary metrics: the Area Under the Precision-Recall Curve (AUCPR) and the F1-score.

**Baselines**  To evaluate the performance of our method on real-world datasets, we benchmark it against 8 state-of-the-art baselines for anomaly detection. These include two classic unsupervised methods, **iForest** (Liu et al., 2008) and **DeepSVDD** (Ruff et al., 2018), and six weakly supervised methods: **DevNet** (Pang et al., 2021a), **DeepSAD** (Ruff et al., 2020), **FeaWAD** (Zhou et al., 2022), **PReNet** (Ren et al., 2019), **RoSAS** (Xu et al., 2023b), and **READ** (Shou et al., 2025). The implementation of baselines is sourced from PyOD library (Zhao et al., 2019), DeepOD library (Xu et al.,

2023a) or their official code repository. To ensure a fair comparison, all baseline methods share the same experimental conditions, including but not limited to training-test splits, data preprocessing pipelines, and evaluation metrics. Every experiment runs three times and we report the mean results throughout this paper.

**Implementation Details**   We fine-tune the tabular in-context learning model, MotherNet, using our proposed ProFiT framework. The fine-tuning process runs for 100 epochs, with each epoch comprising 256 iterations and a batch size of 64 per iteration. For each proxy task, we sample between $\min(100, \text{len(datasets)}/2)$ and $\min(200, \text{len(datasets)})$ instances. Among them, 70% are designated as the support set, while the remaining 30% constitute the query set. Regarding the targets of proxy tasks, categorical features are retained through identity mapping, whereas numerical features are ranked and discretized into categorical variables based on quantile intervals. Fine-tuning is performed using the AdamW optimizer, with an initial learning rate of $3 \times 10^{-5}$, which is gradually decayed following a cosine annealing schedule across epochs. In our experiments, we set $\lambda_{\text{CE}} = 1$ and $\lambda_{\text{JS}} = 20$ in Equation 8. At the inference stage, iForest (Liu et al., 2008) is employed to generate pseudo-labels ($\mathcal{A}$ in algorithm 5) for normal samples.

## 4.1 MAIN RESULTS

Table 1 summarizes the experimental results on 35 benchmark datasets, where the number of labeled anomalies is 5 (owing to space limits, we provide only the dataset-wise average F1 score. The complete per-dataset F1 results are available in Appendix I for the complete AUCPR and F1 results of different shot settings). Overall, our method achieves the best or second-best performance on the vast majority of datasets, and significantly outperforms all baselines in terms of average metrics. In particular, compared to the previous state-of-the-art method READ, our approach improves the average AUCPR by more than 7.5% and the average F1 score by over 5.6%. We report the average performance of different methods across multiple datasets under varying numbers of labeled anomalies, and present the results as boxplots in Figure 2. The boxes represent the interquartile range, the whiskers denote the overall spread, and the red triangles indicate the median values. As shown, our method consistently achieves higher median performance in most cases, with a more compact distribution, demonstrating notable stability and robustness. This suggests that the proposed approach is not only effective under a single experimental condition but also advantageous in more comprehensive scenarios across diverse datasets and labeling scales.

## 4.2 ABLATION STUDIES

**Effectiveness of Proxy-based Fine-tuning**   To assess the effectiveness of the proposed unsupervised fine-tuning method, we compare its performance improvements over MotherNet across multiple datasets, measured by AUCPR and F1 score. As shown in Figure 3, although slight performance drops are observed on a few datasets, our unsupervised fine-tuning method consistently enhances MotherNet on the majority of datasets (see Appendix J for the complete results). On several particularly challenging benchmarks, the method achieves gains exceeding 8%. These results demonstrate that unsupervised fine-tuning enables the model to better capture task-specific data distributions, thereby

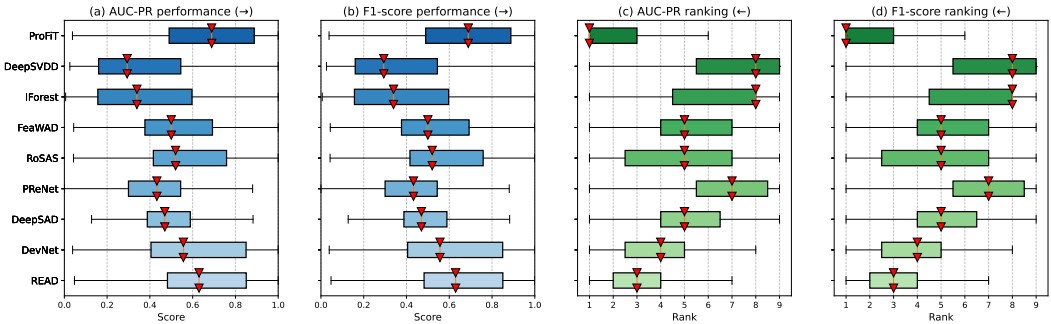

Figure 2: Boxplot comparison of different methods in terms of AUCPR and F1 score.

Table 1: The AUCPR and average F1 performance of all methods across different datasets.

| Dataset | READ | DevNet | DeepSAD | PReNet | RoSAS | FeaWAD | iForest | DeepSVDD | ProFiT |
|---|---|---|---|---|---|---|---|---|---|
| ALOI | 0.0408 | 0.0407 | **0.0528** | 0.0313 | 0.0358 | 0.0433 | 0.0352 | 0.0421 | 0.0382 |
| Annthyroid | 0.6104 | 0.5812 | 0.1905 | 0.172 | 0.4675 | 0.3565 | 0.3393 | 0.1917 | **0.6664** |
| Breastw | 0.7778 | **0.9909** | 0.8555 | 0.5598 | 0.5196 | 0.988 | 0.9841 | 0.8743 | 0.9902 |
| Cardio | 0.5009 | 0.5981 | 0.1849 | 0.2274 | 0.2997 | **0.7015** | 0.5551 | 0.6053 | 0.5007 |
| Cardiotocography | 0.4658 | 0.529 | 0.288 | 0.2804 | 0.3671 | 0.657 | 0.4312 | 0.4359 | **0.7123** |
| Celeba | 0.0973 | 0.0558 | 0.0548 | † | 0.0961 | **0.1807** | 0.0739 | 0.0762 | 0.0302 |
| Census | 0.1118 | 0.0828 | 0.1096 | † | 0.1078 | **0.1425** | 0.0812 | 0.0829 | 0.1403 |
| Donors | 0.9115 | 0.1678 | 0.2219 | † | 0.7604 | 0.5568 | 0.1306 | 0.2046 | **0.9423** |
| Fault | 0.3877 | 0.3591 | **0.4609** | 0.3801 | 0.3997 | 0.2767 | 0.4068 | 0.3387 | 0.4366 |
| Http | **0.9991** | 0.9869 | 0.9834 | † | 0.9985 | 0.8631 | 0.9884 | 0.379 | 0.9842 |
| InternetAds | 0.5112 | 0.3477 | 0.2379 | 0.2443 | 0.3273 | 0.6082 | 0.5318 | 0.2836 | **0.6529** |
| Ionosphere | 0.7171 | 0.7581 | **0.8432** | 0.6458 | 0.6126 | 0.4734 | 0.8121 | 0.7265 | 0.8367 |
| Landsat | 0.3706 | **0.4383** | 0.3236 | 0.2562 | 0.3356 | 0.3073 | 0.1825 | 0.3059 | 0.3676 |
| Letter | 0.1435 | 0.1669 | 0.1945 | 0.1709 | **0.2216** | 0.1024 | 0.1284 | 0.1136 | 0.0751 |
| Magic | 0.5742 | 0.4155 | 0.558 | 0.4305 | 0.549 | 0.5036 | 0.6351 | **0.6445** | 0.4848 |
| Mammography | 0.4277 | 0.3769 | 0.1447 | 0.1645 | 0.2984 | 0.4525 | 0.2295 | 0.201 | **0.5183** |
| Mnist | 0.4766 | 0.2058 | 0.2193 | 0.1347 | 0.2513 | **0.6529** | 0.2667 | 0.2937 | 0.3292 |
| Optdigits | **0.9882** | 0.9752 | 0.27 | 0.253 | 0.7037 | 0.2894 | 0.0583 | 0.0257 | 0.8951 |
| PageBlocks | **0.5515** | 0.4734 | 0.3994 | 0.2195 | 0.2468 | 0.4109 | 0.5231 | 0.5431 | 0.4816 |
| Pendigits | 0.9355 | 0.7879 | 0.2745 | 0.3967 | **0.9518** | 0.6667 | 0.2044 | 0.1068 | 0.8186 |
| Pima | 0.4356 | 0.5485 | 0.3954 | 0.4224 | 0.3408 | 0.4882 | 0.5318 | 0.546 | **0.6497** |
| Satellite | 0.5806 | 0.5624 | 0.37 | 0.5217 | 0.4425 | 0.2667 | 0.6895 | 0.5706 | **0.8066** |
| Satimage-2 | 0.8665 | **0.9198** | 0.3248 | 0.487 | 0.5777 | 0.8795 | 0.879 | 0.1916 | 0.8741 |
| Shuttle | 0.7531 | 0.5455 | 0.294 | 0.2274 | 0.7531 | 0.938 | **0.9783** | 0.9121 | 0.9736 |
| Skin | 0.7703 | 0.6931 | 0.2785 | † | **0.9235** | 0.4775 | 0.2609 | 0.1852 | 0.7938 |
| Smtp | **0.478** | 0.1739 | 0.3423 | 0.2469 | 0.1744 | 0.4761 | 0.006 | 0.3423 | 0.4331 |
| SpamBase | 0.6511 | 0.6263 | 0.3945 | 0.4313 | 0.573 | 0.3606 | 0.5061 | 0.3929 | **0.8387** |
| Thyroid | 0.9034 | 0.8619 | 0.2156 | 0.5686 | 0.7702 | 0.4178 | 0.559 | 0.274 | **0.9236** |
| Vertebral | 0.613 | 0.3034 | 0.4718 | 0.6375 | 0.5487 | 0.2713 | 0.1241 | 0.1047 | **0.6621** |
| WBC | **1.0** | 0.9167 | 0.9167 | 0.7 | 0.3618 | 0.9167 | **1.0** | **1.0** | **1.0** |
| WDBC | 0.7143 | **1.0** | 0.5119 | 0.6884 | 0.6979 | **1.0** | 0.8333 | 0.8095 | **1.0** |
| Wilt | 0.38 | 0.5067 | 0.085 | 0.2315 | **0.7442** | 0.0421 | 0.0469 | 0.0366 | 0.4468 |
| Wine | **1.0** | **1.0** | **1.0** | **1.0** | **1.0** | **1.0** | 0.2143 | 0.1288 | **1.0** |
| WPBC | 0.3837 | 0.3731 | **0.5251** | 0.4074 | 0.4717 | 0.3059 | 0.2583 | 0.2774 | 0.4325 |
| Yeast | 0.4335 | **0.4468** | 0.3386 | 0.4159 | 0.4237 | 0.367 | 0.3107 | 0.3103 | 0.3710 |
| Average | 0.5875 | 0.5376 | 0.3809 | 0.3851 | 0.4958 | 0.4983 | 0.4227 | 0.3588 | **0.6316** |
| Average Rank | 3.2286 | 4.2 | 5.8286 | 7.0 | 4.8286 | 4.6286 | 5.4286 | 6.0 | **2.8571** |
| Average F1 | 0.5876 | 0.5224 | 0.3620 | 0.3679 | 0.4872 | 0.4554 | 0.3908 | 0.3302 | **0.6148** |
| Average F1 Rank | 2.9429 | 4.0571 | 5.8 | 6.6 | 4.4286 | 4.6 | 5.4286 | 5.8286 | **2.6857** |

† Indicates that no result was available within 12 hours.

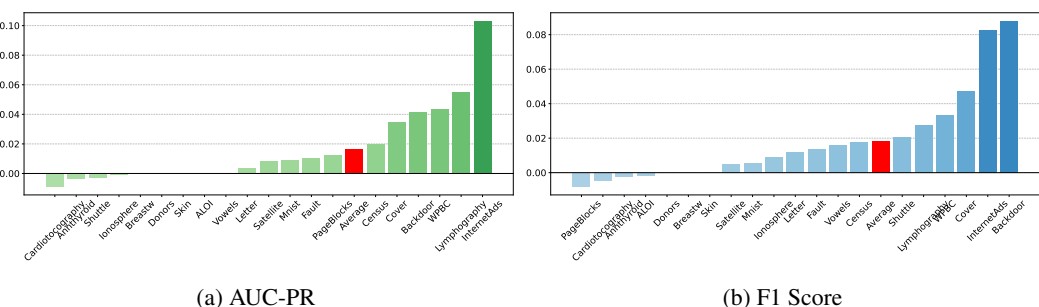

(a) AUC-PR         (b) F1 Score

Figure 3: Performance improvement before and after fine-tuning

substantially improving its anomaly detection capability, while further confirming the practicality and robustness of the proposed approach.

Beyond the overall improvements, we further analyze the conditions under which our unsupervised fine-tuning method is most effective. Our investigation reveals that the gains brought by ProFiT are closely related to two factors: the presence of meaningful latent factor structure and the richness of the feature space. ProFiT yields the most substantial improvements on datasets that exhibit both clear latent structure and sufficient feature dimensionality (e.g., Lymphography, WPBC), enabling proxy tasks to effectively exploit the underlying relationships. When latent structure is present but

the feature dimensionality is very small (e.g., breastw, Skin), the improvement is limited due to the restricted expressiveness of proxy-task modeling. Conversely, even in the absence of strong latent structure, datasets with many features (e.g., InternetAds, Backdoor, Census) still benefit from ProFiT, as the high dimensionality supports diverse and informative proxy-task construction. In contrast, datasets lacking both latent structure and sufficient features (e.g., ALOI, Shuttle) show minimal gains. Overall, these findings clarify the applicability of our method and show that ProFiT is particularly effective when either latent structure or feature richness provides adequate signal for unsupervised fine-tuning. The detailed data analysis can be found in Appendix E.

Table 2: Performance comparison AUCPR, avg F1, avg Rank of all methods across different numbers of anomalies $K$

| # Labeled | Metric | READ | DevNet | DeepSAD | PReNet | RoSAS | FeaWAD | iForest | DeepSVDD | ProFiT |
|---|---|---|---|---|---|---|---|---|---|---|
| $K = 5$ | PR | 0.5875 | 0.5376 | 0.3809 | 0.3851 | 0.4958 | 0.4983 | 0.4227 | 0.3588 | **0.6316** |
| | rank | 3.2286 | 4.2 | 5.8286 | 7.0 | 4.8286 | 4.6286 | 5.4286 | 6.0 | **2.8571** |
| | F1 | 0.5876 | 0.5224 | 0.362 | 0.3679 | 0.4872 | 0.4554 | 0.3908 | 0.3302 | **0.6148** |
| | Rank | 2.9429 | 4.0571 | 5.8 | 6.6 | 4.4286 | 4.6 | 5.4286 | 5.8286 | **2.6857** |
| $K = 10$ | PR | 0.6343 | 0.5914 | 0.4430 | 0.4176 | 0.5559 | 0.5151 | 0.4227 | 0.3588 | **0.6393** |
| | Rank | **2.9429** | 3.6571 | 5.4571 | 7.0 | 4.7429 | 4.5429 | 6.0 | 6.4857 | 3.0857 |
| | F1 | **0.6203** | 0.5651 | 0.4274 | 0.3888 | 0.4985 | 0.4926 | 0.3948 | 0.3338 | 0.6170 |
| | Rank | **2.7429** | 3.4571 | 5.5143 | 7.0571 | 4.3714 | 4.0286 | 5.8286 | 6.2 | 2.8286 |
| $K = 20$ | PR | 0.6864 | 0.6395 | 0.5241 | 0.4989 | 0.6116 | 0.5581 | 0.4227 | 0.3616 | **0.6969** |
| | Rank | **2.6571** | 3.5143 | 5.4857 | 6.7714 | 4.0286 | 4.5429 | 6.5429 | 7.2 | 2.8286 |
| | F1 | **0.6677** | 0.6267 | 0.4986 | 0.4711 | 0.5879 | 0.5335 | 0.3948 | 0.3338 | 0.6674 |
| | Rank | 2.6571 | 3.2286 | 5.6286 | 6.4857 | 4.0286 | 4.3429 | 6.5714 | 7.0857 | **2.4571** |
| $K = 30$ | PR | 0.6309 | 0.5759 | 0.6910 | 0.6690 | 0.5891 | 0.5509 | 0.4227 | 0.3588 | **0.7033** |
| | Rank | 4.1143 | 4.7714 | 2.9429 | 4.1429 | 5.0857 | 5.7429 | 6.6571 | 7.3714 | **2.7429** |
| | F1 | 0.6186 | 0.5504 | 0.6739 | 0.6339 | 0.5666 | 0.4925 | 0.3948 | 0.3338 | **0.6789** |
| | Rank | 3.7143 | 4.6 | 2.7429 | 3.6286 | 4.9429 | 6.0 | 6.8 | 7.0571 | **2.6857** |

**Different Numbers of Labeled Anomalies** To further evaluate the robustness and generalization ability of our method, we conduct ablation analysis under different numbers of labeled anomalies ($K = 5, 10, 20, 30$). As shown in Table 2, iForest and DeepSVDD are unsupervised methods, and their performance does not vary with the number of labeled samples. In contrast, existing weakly-supervised methods perform poorly under low-shot settings, while our method consistently outperforms all baselines across different shots. Particularly in the extremely low-shot case ($K = 5$), our method achieves significantly higher AUCPR and F1 scores than other weakly-supervised anomaly detection methods, demonstrating strong few-shot generalization ability. As $K$ increases, the performance of our method continues to improve, and it still maintains the best AUCPR and F1 under the high-shot setting ($K = 30$), indicating its robustness with sufficient supervision.

## 5 CONCLUSION

We introduced ProFiT, an unsupervised fine tuning framework that adapts pretrained tabular in context learning models to anomaly detection when labels are scarce by training on automatically constructed proxy tasks. By predicting a held out feature from mRMR selected and correlated feature subsets and enforcing cross subset consistency with a Jensen–Shannon divergence regularizer, ProFiT learns representations that align with target domain structure without additional anomaly labels. Our analysis explains why proxy task learning transfers: we derive a regret identity that links proxy risk to conditional entropy, prove sufficiency driven transfer with bounded excess risk, and provide finite sample bounds that highlight the role of information coverage across tasks. Empirically, across 35 benchmarks, ProFiT surpasses weakly-supervised and unsupervised baselines as well as vanilla TICL, with notable gains in average AUCPR and F1 score, and stable improvements among different labeled anomalies. These results show that adaptation using only unlabeled data can narrow the distribution gap that limits pretrained tabular models in practice. Looking forward, ProFiT suggests directions such as adaptive proxy task scheduling, integration with limited supervision, and extensions to settings with concept drift or multi table relational structure.

**Ethics Statement.** This study uses only publicly available datasets. No private, sensitive, or personally identifiable information is involved, and therefore no ethics approval was required.

**Reproducibility Statement.** We have made efforts to ensure the reproducibility of our work. The complete proof process of the theoretical results can be found in the Appendix C. For the experimental part, all datasets used in this study are publicly available, and details regarding dataset access and processing are provided in the supplementary materials (with dataset downloads referenced from Han et al. (2022) and (Rayana, 2016)). The source code implementing our methods will be released publicly upon acceptance.

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

## A USE OF LLMs

In this paper, we employed a Large Language Model (LLM) to assist with text polishing and expression refinement. Specifically, we used LLM, whose primary role was to improve the fluency of language, enhance the academic style of writing, and increase the readability of the text.

It should be emphasized that:

1. The use of LLM in this paper was strictly limited to language refinement and expression optimization. All research ideas, experimental design, data analysis, and conclusions were independently carried out by the researchers.

2. All model-generated content was reviewed and, where necessary, modified by the authors to ensure appropriateness within the research context and compliance with academic standards.

3. The model was not used for data fabrication or manipulation of experimental results. The scientific validity and originality of the research remain entirely the responsibility of the research team.

## B DATASETS DETAILS

**Datasets**   We select 35 real world tabular datasets widely used in anomaly detection tasks, Sourced from ODDS (Rayana, 2016) and ADbench (Han et al., 2022). These datasets span various domains, including healthcare, internet services, finance, etc. They feature a combination of numerical and categorical attributes and exhibit diverse statistical properties, with sizes ranging from 129 to 619,326 samples, dimensions from 3 to 1,555, and anomaly ratios from 0.03 % to 39.91 %. A detailed statistical summary for each dataset can be found in Table 3, which shows the number of samples, the dimension, and the number of anomalies of each dataset used.

## C OMITTED PROOFS

In this section, we present the proofs of  Theorem 3.5,  Theorem 3.6, and  Theorem 3.7.

We begin with a lemma:

**Lemma C.1.** *For random variables* $(\mathbf{y}, \mathbf{u}, \mathbf{Z})$,

$$I(\mathbf{y}; \mathbf{u} \mid \mathbf{Z}) \ = \ \mathbb{E}_{\mathbf{Z}}\Big[D_{\mathrm{KL}}\big(p(\mathbf{y} \mid \mathbf{u}, \mathbf{Z}) \,\|\, p(\mathbf{y} \mid \mathbf{Z})\big)\Big] \ = \ \inf_{q(\cdot|\mathbf{Z})} \mathbb{E}_{\mathbf{Z}, \mathbf{u}}\Big[D_{\mathrm{KL}}\big(p(\mathbf{y} \mid \mathbf{u}, \mathbf{Z}) \,\|\, q(\mathbf{y} \mid \mathbf{Z})\big)\Big].$$

*Equivalently,*

$$I(\mathbf{y}; \mathbf{u} \mid \mathbf{Z})) = \sup_{\phi} \ \mathbb{E}_{\mathbf{y}, \mathbf{u}, \mathbf{Z}}[\phi(\mathbf{y}, \mathbf{u}, \mathbf{Z})] - \mathbb{E}_{\mathbf{y}, \mathbf{Z}}\big[\log \mathbb{E}_{\mathbf{u}}[\exp\{\phi(\mathbf{y}, \mathbf{u}, \mathbf{Z})\} \mid \mathbf{y}, \mathbf{Z}]\big],$$

*where $\phi$ ranges over integrable scoring functions for anomaly detection.*

*Proof.* The first two equalities are the conditional KL form of $I(\cdot; \cdot \mid \cdot)$ and the optimal choice of baseline $q(\cdot \mid \mathbf{Z})$. The last line follows from the Donsker–Varadhan variational representation of KL, applied conditionally on $(\mathbf{Z}, \mathbf{y})$. □

### C.1 PROOF OF LEMMA 3.4

*Proof.* Let $f$ be a representation encoder that maps inputs $\mathbf{X}$ to latent representations $f(\mathbf{X})$. For a fixed pseudo-task $\tau = (\mathbf{X}, S, t)$ with log-loss, consider predictors $h$ that take the representation $f(\mathbf{X}_S)$ as input and output a probability distribution over the target values $\mathbf{y}_t$. The best achievable performance of such a predictor is

$$\inf_{h} \ \mathbb{E}\big[-\log h(f(\mathbf{X}_S))[\mathbf{y}_t]\big] = H(\mathbf{y}_t \mid f(\mathbf{X}_S)).$$

where $H(\cdot)$ denotes Shannon entropy and $I(\cdot, \cdot)$ denotes mutual information, define the *excess risk* of $f$ on $\tau$ as

$$\Delta_f(\tau) := H(\mathbf{y}_t \mid f(\mathbf{X}_S)) - H(\mathbf{y}_t \mid \mathbf{X}_S),$$

Table 3: Statistics of the benchmark datasets

| Dataset | Instances | Dimensions | Anomalies | Anomaly Ratio (%) |
|---|---|---|---|---|
| ALOI | 49534 | 27 | 1508 | 3.04 % |
| Annthyroid | 7200 | 6 | 534 | 7.42 % |
| Breastw | 683 | 9 | 239 | 34.99 % |
| Cardio | 1831 | 21 | 176 | 9.61 % |
| Cardiotocography | 2114 | 21 | 466 | 22.04 % |
| Celeba | 202599 | 39 | 4547 | 2.24 % |
| Census | 299285 | 500 | 18568 | 6.20 % |
| Donors | 619326 | 10 | 36710 | 5.93 % |
| Fault | 1941 | 27 | 673 | 34.67 % |
| Http | 567498 | 3 | 2211 | 0.39 % |
| InternetAds | 1966 | 1555 | 368 | 18.72 % |
| Ionosphere | 351 | 32 | 126 | 35.90 % |
| Landsat | 6435 | 36 | 1333 | 20.71 % |
| Letter | 1600 | 32 | 100 | 6.25 % |
| Magic | 19020 | 10 | 6688 | 35.16 % |
| Mammography | 11183 | 6 | 260 | 2.32 % |
| Mnist | 7603 | 100 | 700 | 9.21 % |
| Optdigits | 5216 | 64 | 150 | 2.88 % |
| PageBlocks | 5393 | 10 | 510 | 9.46 % |
| Pendigits | 6870 | 16 | 156 | 2.27 % |
| Pima | 768 | 8 | 268 | 34.90 % |
| Satellite | 6435 | 36 | 2036 | 31.64 % |
| Satimage-2 | 5803 | 36 | 71 | 1.22 % |
| Shuttle | 49097 | 9 | 3511 | 7.15 % |
| Skin | 245057 | 3 | 50859 | 20.75 % |
| Smtp | 95156 | 3 | 30 | 0.03 % |
| SpamBase | 4207 | 57 | 1679 | 39.91 % |
| Thyroid | 3772 | 6 | 93 | 2.47 % |
| Vertebral | 240 | 6 | 30 | 12.50 % |
| WBC | 223 | 9 | 10 | 4.48 % |
| WDBC | 367 | 30 | 10 | 2.72 % |
| Wilt | 4819 | 5 | 257 | 5.33 % |
| Wine | 129 | 13 | 10 | 7.75 % |
| WPBC | 198 | 33 | 47 | 23.74 % |
| Yeast | 1484 | 8 | 507 | 34.16 % |

which measures the information loss caused by compressing $\mathbf{X}_S$ into $f(\mathbf{X}_S)$. By the chain rule of mutual information (MacKay, 2003),

$$\Delta_f(\tau) = I(\mathbf{y}_t; \mathbf{X}_S \mid f(\mathbf{X}_S)) \geq 0.$$

Under the latent factor assumption (Assumption 3.1), we have $H(\mathbf{y}_t \mid \mathbf{X}_S) = H(\mathbf{y}_t \mid \mathbf{u})$, so the excess risk reduces to

$$\Delta_f(\tau) = H(\mathbf{y}_t \mid f(\mathbf{X}_S)) - H(\mathbf{y}_t \mid \mathbf{u}) = I(\mathbf{y}_t; \mathbf{u} \mid f(\mathbf{X}_S)).$$

$\square$

This highlights a key point: the performance gap of $f$ is exactly the task-relevant information about $\mathbf{u}$ that $f(\mathbf{X}_S)$ fails to preserve.

## C.2 PROOF OF THEOREM 3.5

*Proof.* By Theorem 3.4 and Assumption 3.2,

$$\mathcal{R}_{\tau^\star}(f, h_f^\star) - \mathcal{R}_{\tau^\star}^{\mathrm{Bayes}} = H(\mathbf{y}^\star \mid f(\mathbf{X}_{S^\star})) - H(\mathbf{y}^\star \mid \mathbf{u}) = I(\mathbf{y}^\star; \mathbf{u} \mid f(\mathbf{X}_{S^\star})) = \Delta_f(\tau^\star).$$

We compare $\Delta_f(\tau^\star)$ with the $Q$-average $\overline{\Delta}_f$. Using Lemma C.1,

$$\Delta_f(\tau^\star) = \inf_{q^\star} \mathbb{E}\big[D_{\mathrm{KL}}\big(p(\mathbf{y}^\star \mid \mathbf{u}, f(\mathbf{X}_{S^\star})) \,\big\|\, q^\star(\cdot \mid f(\mathbf{X}_{S^\star}))\big)\big].$$

An analogous expression holds for each $\tau = (\mathbf{X}, S, t)$. If the proxy-task sampling $Q$ sufficiently covers the latent directions relevant to $t^\star$, there exists a finite constant

$$\Gamma(Q, \tau^\star) := \sup_f \frac{I(\mathbf{y}^\star; \mathbf{u} \mid f(\mathbf{X}_{S^\star}))}{\mathbb{E}_{\tau \sim Q} I(\mathbf{y}_t; \mathbf{u} \mid f(\mathbf{X}_S))} \; \in \; [1, \infty)$$

such that $I(\mathbf{y}^\star; \mathbf{u} \mid f(\mathbf{X}_{S^\star})) \leq \Gamma(Q, \tau^\star) \, \mathbb{E}_{\tau \sim Q} I(\mathbf{y}_t; \mathbf{u} \mid f(\mathbf{X}_S))$. Hence

$$\mathcal{R}_{\tau^\star}(f, h_f^\star) - \mathcal{R}_{\tau^\star}^{\mathrm{Bayes}} = \Delta_f(\tau^\star) \; \leq \; \Gamma(Q, \tau^\star) \cdot \overline{\Delta}_f.$$

When $Q$ richly excites all latent directions, $\Gamma(Q, \tau^\star)$ is close to 1. $\qquad\square$

## C.3 PROOF OF THEOREM 3.6

*Proof.* Assume the factor model $\mathbf{y}_t = g_t(\mathbf{u}) + \boldsymbol{\epsilon}_t$ with Fisher information $\mathbf{F}_t(\mathbf{u})$ for $\mathbf{u}$. Fix $Z := f(\mathbf{X}_S)$. For regular observation models (e.g., smooth exponential families), a local second-order expansion of the conditional log-likelihood gives, for some estimator $\hat{\mathbf{u}}(Z)$ and some $\tilde{\mathbf{u}}$ between $\mathbf{u}$ and $\hat{\mathbf{u}}(Z)$,

$$D_{\mathrm{KL}}\big(p(\mathbf{y}_t \mid \mathbf{u}, Z) \,\|\, p(\mathbf{y}_t \mid Z)\big) \; \gtrsim \; \tfrac{1}{2} (\mathbf{u} - \hat{\mathbf{u}}(Z))^\top \mathbf{F}_t(\tilde{\mathbf{u}}) (\mathbf{u} - \hat{\mathbf{u}}(Z)).$$

Taking expectations over $(\mathbf{u}, Z)$ and optimizing the choice of $\hat{\mathbf{u}}$ yields a constant (absorbed into the information scale) such that

$$I(\mathbf{y}_t; \mathbf{u} \mid Z) \; \geq \; \mathbb{E}\Big[(\mathbf{u} - \hat{\mathbf{u}}(Z))^\top \mathbb{E}[\mathbf{F}_t(\mathbf{u}) \mid Z] (\mathbf{u} - \hat{\mathbf{u}}(Z))\Big].$$

Averaging over $\tau \sim Q$ and exchanging expectations,

$$\overline{\Delta}_f = \mathbb{E}_{\tau \sim Q} I(\mathbf{y}_t; \mathbf{u} \mid Z) \; \geq \; \mathbb{E}\Big[(\mathbf{u} - \hat{\mathbf{u}}(Z))^\top \Big(\mathbb{E}_{\tau \sim Q, \mathbf{u}}[\mathbf{F}_t(\mathbf{u})]\Big) (\mathbf{u} - \hat{\mathbf{u}}(Z))\Big] = \mathbb{E}\big[(\mathbf{u} - \hat{\mathbf{u}}(Z))^\top M_Q (\mathbf{u} - \hat{\mathbf{u}}(Z))\big].$$

By $M_Q \succeq \mu I_r$,

$$\overline{\Delta}_f \; \geq \; \mu \, \mathbb{E}\big[\|\mathbf{u} - \hat{\mathbf{u}}(Z)\|_2^2\big].$$

For the target task $\tau^\star$,

$$\Delta_f(\tau^\star) = I(\mathbf{y}^\star; \mathbf{u} \mid f(\mathbf{X}_{S^\star})) \; \lesssim \; \mathbb{E}\big[\|\mathbf{u} - \hat{\mathbf{u}}(f(\mathbf{X}_{S^\star}))\|_2^2\big].$$

Using the same $\hat{\mathbf{u}}$ on both sides and combining with the previous lower bound,

$$\Delta_f(\tau^\star) \; \leq \; \frac{1}{\mu} \, \overline{\Delta}_f.$$

$\qquad\square$

## C.4 PROOF OF THEOREM 3.7 (FINITE-SAMPLE END-TO-END TRANSFER)

*Proof.* Let $\widehat{\theta}$ and $\widehat{f}$ be obtained by empirical risk minimization over $m$ proxy tasks with $n$ samples each.

**(i) Empirical excess.** By the regret identity, the empirical cross-entropy risk $\widehat{L}_Q(\theta)$ differs from the empirical conditional-entropy baseline $\widehat{H}_Q$ exactly by the empirical excess:

$$\widehat{L}_Q(\theta) - \widehat{H}_Q \; = \; \text{empirical proxy excess.}$$

**(ii) Generalization gap.** Let $L_Q(\theta)$ denote the population cross-entropy risk. Standard uniform convergence (e.g., PAC-Bayes or localized Rademacher) gives with probability at least $1 - \delta$:

$$L_Q(\widehat{\theta}) - H_Q \; \leq \; \widehat{L}_Q(\widehat{\theta}) - \widehat{H}_Q \; + \; \mathrm{Gen}(M, n, \delta).$$

In a PAC-Bayes form with prior $p = \mathcal{N}(\theta_0, \sigma^2 I)$ and posterior approximated by a point mass at $\widehat{\theta}$,

$$\mathrm{Gen}(M, n, \delta) \; \lesssim \; \sqrt{\frac{D_{\mathrm{KL}}(\delta_{\widehat{\theta}}\|p) + \log(1/\delta)}{Mn}} = \sqrt{\frac{\|\widehat{\theta} - \theta_0\|_2^2/(2\sigma^2) + \log(1/\delta)}{Mn}},$$

Thus, with probability $\geq 1 - \delta$,

$$\mathbb{E}_{\tau \sim Q}\big[I(\mathbf{y}_t; \mathbf{u} \mid \widehat{f}(\mathbf{X}_S))\big] = L_Q(\widehat{\theta}) - H_Q \; \leq \; \widehat{L}_Q(\widehat{\theta}) - \widehat{H}_Q + \mathrm{Gen}(M, n, \delta).$$

**(iii) From average to target task; accounting for mismatch.** Decompose the target risk gap into a representation-induced part and an irreducible mismatch:

$$\mathcal{R}_{\tau^\star}(\widehat{f}, \widehat{h}^\star) - \mathcal{R}_{\tau^\star}^{\text{Bayes}} = \underbrace{\left(H_{\tau^\star}(\mathbf{y}^\star \mid \widehat{f}) - H_{\tau^\star}(\mathbf{y}^\star \mid \mathbf{u})\right)}_{\Delta_{\widehat{f}}(\tau^\star)} + \underbrace{\left(H_{\tau^\star}(\mathbf{y}^\star \mid \mathbf{u}) - \mathbb{E}_{\tau \sim Q} H_\tau(\mathbf{y}_t \mid \mathbf{u})\right)}_{\text{disc}(Q, \tau^\star)}.$$

Apply Theorem 3.6 to the first term:

$$\Delta_{\widehat{f}}(\tau^\star) \leq \frac{1}{\mu} \mathbb{E}_{\tau \sim Q} I(\mathbf{y}_t; \mathbf{u} \mid \widehat{f}(\mathbf{X}_S)) \leq \frac{1}{\mu} \left(\widehat{L}_Q(\widehat{\theta}) - \widehat{H}_Q + \text{Gen}(M, n, \delta)\right).$$

Combining both parts completes the proof:

$$\mathcal{R}_{\tau^\star}(\widehat{f}, \widehat{h}^\star) - \mathcal{R}_{\tau^\star}^{\text{Bayes}} \leq \frac{1}{\mu} \left(\widehat{L}_Q(\widehat{\theta}) - \widehat{H}_Q + \text{Gen}(M, n, \delta)\right) + \text{disc}(Q, \tau^\star).$$

$\square$

## D  TRAINING AND INFERENCE ALGORITHM

---

**Algorithm 1:** ProFiT: Unsupervised Proxy-Task Fine-tuning for Tabular Anomaly Detection

---

**Input:** Unlabeled train samples $\mathbf{X} = \{\mathbf{x}_i\}_{i=1}^N \in \mathbb{R}^{N \times d}$; feature set $\mathcal{D} = \{1, \ldots, d\}$; TICL model $\mathcal{T}_\theta$; maximal subset size $k_{\max}$; proxy tasks per epoch $M$; loss weights $\lambda_{\text{CE}}, \lambda_{\text{JS}}$; minibatch size $n$; learning rate $\eta$; identity mapping or quantile function $\varepsilon$; S2 keep ratio $\rho_{\text{keep}}$; minimal Jaccard distance $\delta$; maximal refinement trials $T_{\max}$

**Output:** Fine-tuned model $\mathcal{T}_\theta$.

1 **Precompute** absolute correlation matrix $\mathbf{C} \in \mathbb{R}^{d \times d}$ of features (set $\mathbf{C}_{ii} = 0$).
2 **for** *epoch* $= 1, 2, \ldots$ **do**
3      $L_{\text{CE}} \leftarrow 0, \quad L_{\text{JS}} \leftarrow 0$
4      **for** $j = 1$ **to** $M$ **do**
5          Sample a target feature $t \sim \text{Unif}(\mathcal{D})$.
6          $(S_1, S_2, k) \leftarrow \text{BUILDTWOSUBSETS}(\mathbf{C}, t, \mathcal{D}, k_{\max}, \rho_{\text{keep}}, \delta, T_{\max})$.
7          Sample indices $\mathcal{I} \subset \{1, \ldots, N\}$ with $|\mathcal{I}| = n$.
8          **foreach** $i \in \mathcal{I}$ **do**
9              $\tilde{y}_i^{(t)} \leftarrow \varepsilon(\mathbf{X}_{i,t})$;
10            $\widehat{p}_\theta(\cdot \mid \mathbf{x}_i, S_1) \leftarrow \mathcal{T}_\theta(\mathbf{x}_i^{(S_1)}); \quad \widehat{p}_\theta(\cdot \mid \mathbf{x}_i, S_2) \leftarrow \mathcal{T}_\theta(\mathbf{x}_i^{(S_2)})$;
11      Calculate the loss $\mathcal{L}$ by Eq. (8).
12      Update $\theta \leftarrow \theta - \eta \nabla_\theta \mathcal{L}$.
13 **return** $\mathcal{T}_\theta$.

---

---

**Algorithm 2:** BUILDTWOSUBSETS for a target feature $t$

---

**Input:** Correlation matrix $\mathbf{C}$; target index $t$; feature set $\mathcal{D}$; procedure BUILDS1; keep ratio $\rho_{\text{keep}}$
**Output:** Two subsets $S_1, S_2$ and their size $k$.

1   $(S_1, k) \leftarrow \text{BUILDS1}(\mathbf{C}, t, \mathcal{D})$
2   $\mathcal{C} \leftarrow \mathcal{D} \setminus \{t\}$
3   Sort $S_1$ in descending order of $\mathbf{C}_{i,t}$ and denote the ordered list by $S_1^{\text{sorted}}$.
4   $\text{keep}_k \leftarrow \max\left(1, \min(k-1, \lfloor\rho_{\text{keep}}k\rfloor)\right)$
5   $\text{keep} \leftarrow$ first $\text{keep}_k$ elements of $S_1^{\text{sorted}}$.
6   $\text{need} \leftarrow k - \text{keep}_k$
7   $\mathcal{R} \leftarrow \mathcal{C} \setminus \text{keep}$
8   **if** *need > 0* **then**
9     $\text{add} \leftarrow \text{MRMRSELECT}(\mathbf{C}, t, \mathcal{R}, \text{need})$
10 **else**
11     $\text{add} \leftarrow \emptyset$
12 **if** *|add| < need* **then**
13     $\mathcal{E} \leftarrow \mathcal{R} \setminus \text{add}$
14     **if** $\mathcal{E} \neq \emptyset$ **then**
15       Randomly sample $\min(|\mathcal{E}|, \text{need} - |\text{add}|)$ indices from $\mathcal{E}$ and append to add.

16 $S_2 \leftarrow \text{keep} \cup \text{add}$
17 **if** *|S_2| < k* **then**
18     $\mathcal{E} \leftarrow \mathcal{C} \setminus S_2$
19     **if** $\mathcal{E} \neq \emptyset$ **then**
20       Randomly sample $k - |S_2|$ indices from $\mathcal{E}$ and add to $S_2$.

21 **return** $S_1, S_2, k$.

---

**Algorithm 3:** BUILDS1 for a target feature $t$

---

**Input:** Correlation matrix $\mathbf{C}$; target index $t$; feature set $\mathcal{D}$; procedure $\text{CHOOSEK}(D)$
**Output:** Subset $S_1$ and its size $k$.

1   $\text{cand} \leftarrow \mathcal{D} \setminus \{t\}$
2   $k \leftarrow \text{CHOOSEK}(|\mathcal{D}|)$
3   $S_1 \leftarrow \text{MRMRSELECT}(\mathbf{C}, t, \text{cand}, k)$
4   **if** *|S_1| < k* **then**
5     $\mathcal{E} \leftarrow \text{cand} \setminus S_1$
6     **if** $\mathcal{E} \neq \emptyset$ **then**
7       Randomly sample $k - |S_1|$ indices from $\mathcal{E}$ and add to $S_1$.

8 **return** $S_1, k$

---

---

**Algorithm 4:** MRMRSELECT for a target feature $t$

**Input:** Correlation matrix $\mathbf{C}$; target index $t$; candidate set $\mathcal{R}$; subset size $k$
**Output:** Selected subset $S$.

1 **if** $|\mathcal{R}| = 0$ *or* $k = 0$ **then**
2     **return** $\emptyset$
3 Compute $\mathrm{rel}[v] \leftarrow \mathbf{C}_{v,t}$ for all $v \in \mathcal{R}$.
4 Let $O$ be $\mathcal{R}$ sorted in descending order of rel.
5 $S \leftarrow \{O[1]\}$
6 **for** $i = 2$ **to** $\min(k, |O|)$ **do**
7     best $\leftarrow$ None,   best_score $\leftarrow -\infty$
8     **foreach** $v \in O$ **do**
9        **if** $v \in S$ **then**
10           **continue**
11        redundancy $\leftarrow \mathrm{mean}(\mathbf{C}_{v,u} : u \in S)$
12        score $\leftarrow \mathbf{C}_{v,t} -$ redundancy
13        **if** score $>$ *best_score* **then**
14           best $\leftarrow v$,   best_score $\leftarrow$ score
15     **if** *best $\neq$ None* **then**
16        $S \leftarrow S \cup \{$best$\}$
17 **return** $S$

---

**Algorithm 5:** ProFiT Inference with TICL $\mathcal{T}_\theta$ and Pseudo-Normals from Unsupervised Detector

**Input:** Fine-tuned TICL $\mathcal{T}_\theta$; unsupervised anomaly detector $\mathcal{A}$; labeled anomalies
     $\mathcal{D}_L = \{(\mathbf{x}_l, 1)\}_{l=1}^K$; unlabeled pool $\mathcal{D}_U = \{\mathbf{x}_u\}_{u=1}^N$; pseudo-normal count $k$; test set
     $\mathcal{D}_{\text{test}} = \{\mathbf{x}_j^{\text{test}}\}_{j=1}^{N_{\text{test}}}$.
**Output:** Anomaly scores $\{s(\mathbf{x}_j^{\text{test}})\}_{j=1}^{N_{\text{test}}}$.

1 **for** $\mathbf{x}_i$ *in* $\mathcal{D}_U$ **do**
2     $a_i \leftarrow \mathcal{A}(\mathbf{x}_i)$
3 $q_{0.8} \leftarrow \mathrm{Percentile}(\{a_i\}_{i=1}^N, 80)$
4 $\mathcal{C} \leftarrow \{\mathbf{x}_i \in \mathcal{D}_U \mid a_i \leq q_{0.8}\}$
5 Select $K$ samples $\{\mathbf{x}_u^{(0)}\}_{u=1}^K$ uniformly at random from $\mathcal{C}$ (without replacement)
6 Assign pseudo-labels: $\{(\mathbf{x}_u^{(0)}, 0)\}_{u=1}^K$
7 $\mathcal{S} \leftarrow \{\mathcal{D}_L \cup \{(\mathbf{x}_u^{(0)}, 0)\}_{u=1}^K$
8 **Obtain downstream MLP parameters:** $\phi \leftarrow \mathcal{T}_\theta(\mathcal{S})$
9 **for** $j = 1, \ldots, N_{test}$ **do**
10     $s(\mathbf{x}_j^{\text{test}}) \leftarrow \mathrm{MLP}_\phi(\mathbf{x}_j^{\text{test}})$
11 **return** $\{s(\mathbf{x}_j^{\text{test}})\}_{j=1}^{N_{\text{test}}}$

---

# E   EFFECTIVENESS OF PROFIT

As shown in Figure 3, beyond the overall improvements, we further analyze the conditions under which our unsupervised fine-tuning method is most effective. As shown in Figure 3, beyond the overall improvements, we further analyze the conditions under which our unsupervised fine-tuning method is most effective. To quantify the extent to which a dataset exhibits underlying latent factors, we compute two correlation-based statistics. First, we measure the original correlation, defined as the mean absolute pairwise correlation of the raw feature correlation matrix. Second, after extracting latent factors using Factor Analysis and reconstructing the data, we obtain the residual matrix and compute the residual correlation, i.e., the mean absolute pairwise correlation of the residual features. Based on these two quantities, we define the latent factor strength as

$$LFS = 1 - \frac{\text{residual correlation}}{\text{original correlation}},$$

which reflects the proportion of the original feature dependence that can be explained by latent factors.

As shown in Table 4, our investigation reveals that the gains brought by ProFiT are closely related to two factors: the presence of meaningful latent factor structure and the richness of the feature space. ProFiT yields the most substantial improvements on datasets that exhibit both clear latent structure and sufficient feature dimensionality (e.g., Lymphography, WPBC), enabling proxy tasks to effectively exploit the underlying relationships. When latent structure is present but the feature dimensionality is very small (e.g., breastw, Skin), the improvement is limited due to the restricted expressiveness of proxy-task modeling. Conversely, even in the absence of strong latent structure, datasets with many features (e.g., InternetAds, Backdoor, Census) still benefit from ProFiT, as the high dimensionality supports diverse and informative proxy-task construction. In contrast, datasets lacking both latent structure and sufficient features (e.g., ALOI, Shuttle) show minimal gains. Overall, these findings clarify the applicability of our method and show that ProFiT is particularly effective when either latent structure or feature richness provides adequate signal for unsupervised fine-tuning.

Table 4: Relation between performance improvement, latent factor strength, and feature dimensionality across datasets

| Datasets | Dim | Original Corr | Residual Corr | LFS | F1 Impr. | PR Impr. |
|---|---|---|---|---|---|---|
| Lymphography | 18 | 0.1680 | 0.1075 | 36.01% | 0.0278 | 0.0554 |
| WPBC | 33 | 0.2905 | 0.1290 | 55.59% | 0.0334 | 0.0435 |
| Breastw | 9 | 0.6019 | 0.1372 | 77.21% | 0.0000 | 0.0000 |
| Skin | 3 | 0.6961 | 0.4181 | 39.94% | 0.0000 | 0.0000 |
| InternetAds | 1555 | 0.0183 | 0.0183 | 0.00% | 0.0825 | 0.1031 |
| Backdoor | 196 | 0.1061 | 0.1392 | -31.20% | 0.0879 | 0.0414 |
| Census | 500 | 0.0297 | 0.0273 | 8.08% | 0.0179 | 0.0199 |
| ALOI | 27 | 0.0946 | 0.0870 | 8.03% | -0.0017 | 0.0002 |
| Shuttle | 9 | 0.1885 | 0.1778 | 5.68% | 0.0204 | -0.0032 |

## F  ANALYSIS OF NORMAL SAMPLE PSEUDO-LABELING

In the anomaly detection task, the number of abnormal samples is generally much lower than that of normal samples. We conducted experiments on 35 datasets to investigate the impact of different pseudo-labeling strategies for normal samples. The results, shown in Table 5, demonstrate that the ProFiT fine-tuned model significantly improves upon the baseline MotherNet model, regardless of the pseudo-labeling strategy used for normal samples. This shows that ProFiT is robust to different pseudo-labeling methods.

1. **IForest Topk**: This strategy uses iForest to select the K samples with the lowest anomaly scores from the unlabeled data. While these samples have the highest confidence, their diversity is limited, leading to relatively lower model performance.

2. **IForest RandomK 80%:** This approach randomly selects K samples from the lowest 80% of the anomaly scores. Although it may introduce noise, the diversity of samples significantly improves, resulting in a substantial boost in model performance.

3. **Random:** This method selects K samples directly from all unlabeled samples, without any filtering. Interestingly, this random sampling outperforms the other methods, including the iForest-based approaches, in terms of average performance. This highlights the robustness of the model to different pseudo-labeling strategies and emphasizes the importance of sample diversity over strict accuracy.

Furthermore, we attempted to extend our method to an Unsupervised approach, where both normal and abnormal samples were pseudo-labeled using IForest. This approach led to a significant performance drop. The reason is that abnormal samples are much fewer than normal samples in the dataset, and accurate pseudo-labeling is crucial for guiding the model. When using an unsupervised detector, the accuracy of the labeled abnormal samples is too low, which negatively impacts model performance. In contrast, the large number of normal samples in the dataset can tolerate the noise introduced by the

unsupervised pseudo-labeling process without significantly affecting model performance. Therefore, accurate labeling of abnormal samples is essential for effective model performance, while normal sample diversity plays a significant role in improving robustness.

Table 5: Performance (F1 and PR) of different pseudo-labeling strategies for support set selection on MotherNet and ProFiT.

| Method | F1 | | PR | |
|---|---|---|---|---|
| | MotherNet | ProFiT | MotherNet | ProFiT |
| IForest Topk | 0.5677 | 0.5573 | 0.5904 | 0.6085 |
| IForest RandomK 80% | 0.6696 | 0.6770 | 0.6954 | 0.7084 |
| Random | 0.7014 | 0.7128 | 0.7354 | 0.7475 |
| Unsupervised | 0.3402 | 0.3260 | 0.3660 | 0.3610 |

## G  EXTENSION TO OTHER TASK

ProFiT is not limited to the unsupervised fine-tuning setting used for anomaly detection. In principle, the method can be applied to a broader range of representation learning scenarios. We primarily chose the anomaly detection setup due to the support–query nature of TICL models, which naturally aligns with settings involving extremely limited labeled data. This makes anomaly detection a representative and meaningful testbed for evaluating the benefits of our fine-tuning paradigm.

To assess the generality of our approach, we further evaluated ProFiT on several general-purpose tabular classification benchmarks. In addition to the anomaly detection tasks reported in the main paper, we selected a subset of datasets from the CC70 benchmark suite and conducted corresponding experiments. As shown in Table 6, the results show that ProFiT consistently improves performance on standard tabular classification tasks as well. These findings suggest that the proposed method possesses a certain degree of task generality beyond anomaly detection, and can potentially serve as a more universal fine-tuning strategy for tabular representation models.

Table 6: F1 score and AUCPR on general classification tasks before and after applying ProFiT.

| Dataset | F1 | | PR | |
|---|---|---|---|---|
| | MotherNet | ProFiT | MotherNet | ProFiT |
| PC4 | 0.3333 | 0.3889 | 0.3687 | 0.4584 |
| KC2 | 0.4545 | 0.4545 | 0.4304 | 0.4735 |
| KC1 | 0.3438 | 0.3438 | 0.3672 | 0.4150 |
| PC1 | 0.4286 | 0.4286 | 0.5169 | 0.5304 |
| BankMarketing | 0.4178 | 0.4178 | 0.3786 | 0.3852 |
| Nomao | 0.7584 | 0.7422 | 0.7494 | 0.7948 |
| Dresses Sales | 0.4762 | 0.6667 | 0.5124 | 0.6795 |
| Credit Approval | 0.8462 | 0.7692 | 0.8673 | 0.8688 |
| Sick | 0.5217 | 0.6957 | 0.5638 | 0.7406 |
| Bioresponse | 0.6029 | 0.6029 | 0.5840 | 0.5974 |
| Spambase | 0.6868 | 0.6923 | 0.7911 | 0.8087 |
| PhishingWebsites | 0.8920 | 0.8920 | 0.9488 | 0.9563 |
| Tic Tac Toe | 0.7460 | 0.7302 | 0.7146 | 0.7327 |
| **Average** | 0.5776 | **0.6019** | 0.5995 | **0.6493** |

## H  SEMI-SUPERVISED TAD BASELINES

We additionally evaluate our method against two representative semi-supervised tabular anomaly detection approaches: MCM Yin et al. (2024) and DRL Ye et al. (2025). Both baselines are implemented using their official code releases and recommended hyperparameters. To ensure a

Table 7: F1 and PR comparison of DRL, MCM, and ProFiT across datasets.

| Dataset | F1 Score | | | PR | | |
|---|---|---|---|---|---|---|
| | DRL | MCM | ProFiT | DRL | MCM | ProFiT |
| ALOI | 0.0286 | **0.0813** | 0.0308 | 0.0291 | **0.0531** | 0.0382 |
| Annthyroid | 0.2023 | 0.2428 | **0.7168** | 0.1752 | 0.2196 | **0.6664** |
| Breastw | 0.8767 | **0.9589** | **0.9589** | 0.9543 | 0.9856 | **0.9902** |
| Cardio | 0.3214 | 0.4107 | **0.5893** | 0.3125 | 0.4347 | **0.5007** |
| Cardiotocography | 0.4326 | 0.4894 | **0.6099** | 0.3816 | 0.4290 | **0.7123** |
| Celeba | **0.1302** | 0.0756 | 0.0455 | **0.0605** | 0.0522 | 0.0302 |
| Census | 0.0380 | 0.0656 | **0.1608** | 0.0678 | 0.0784 | **0.1403** |
| Donors | 0.0311 | 0.0331 | **0.8600** | 0.1097 | 0.0862 | **0.9423** |
| Fault | 0.4545 | **0.4976** | 0.4258 | 0.4756 | **0.4973** | 0.4366 |
| Http | 0.0161 | 0.0587 | **0.9882** | 0.2509 | 0.6029 | **0.9842** |
| Ionosphere | **0.8108** | 0.1081 | 0.7838 | **0.8688** | 0.2408 | 0.8367 |
| Landsat | 0.2481 | 0.1830 | **0.3659** | 0.2580 | 0.2018 | **0.3676** |
| Letter | **0.3939** | 0.1515 | 0.0303 | **0.4104** | 0.1175 | 0.0751 |
| Magic | 0.4753 | **0.6408** | 0.4852 | 0.5635 | **0.7203** | 0.4848 |
| Mammography | 0.2000 | 0.3294 | **0.5412** | 0.1310 | 0.2244 | **0.5183** |
| Mnist | **0.5519** | 0.4057 | 0.3632 | **0.5544** | 0.3898 | 0.3292 |
| Optdigits | 0.0000 | 0.0000 | **0.8140** | 0.0256 | 0.0407 | **0.8951** |
| PageBlocks | 0.2986 | 0.2500 | **0.5417** | 0.3227 | 0.2168 | **0.4816** |
| Pendigits | 0.0455 | 0.1818 | **0.7273** | 0.0326 | 0.0911 | **0.8186** |
| Pima | 0.4684 | 0.5316 | **0.6329** | 0.4667 | 0.5193 | **0.6497** |
| Satellite | 0.5497 | 0.5304 | **0.6619** | 0.5864 | 0.6771 | **0.8066** |
| Satimage-2 | 0.6087 | 0.7391 | **0.8696** | 0.5295 | 0.7007 | **0.8741** |
| Shuttle | 0.9159 | 0.9039 | **0.9510** | 0.8728 | 0.7982 | **0.9736** |
| Skin | 0.1923 | 0.0022 | **0.7826** | 0.2504 | 0.1595 | **0.7938** |
| Smtp | **0.6154** | **0.6154** | 0.4615 | 0.4399 | **0.5982** | 0.4331 |
| SpamBase | 0.5172 | 0.5960 | **0.7636** | 0.5508 | 0.6046 | **0.8387** |
| Thyroid | 0.3704 | 0.4444 | **0.8889** | 0.3293 | 0.3547 | **0.9236** |
| Vertebral | 0.0000 | 0.0000 | **0.6364** | 0.1589 | 0.1270 | **0.6621** |
| WBC | 0.3333 | 0.0000 | **1.0000** | 0.4250 | 0.2619 | **1.0000** |
| WDBC | 0.6667 | **1.0000** | **1.0000** | 0.7292 | **1.0000** | **1.0000** |
| Wilt | 0.1190 | 0.0000 | **0.5714** | 0.1162 | 0.0399 | **0.4468** |
| Wine | 0.0000 | **1.0000** | **1.0000** | 0.2250 | **1.0000** | **1.0000** |
| WPBC | **0.3333** | 0.2000 | **0.3333** | 0.3444 | 0.3076 | **0.4325** |
| Yeast | 0.3312 | 0.3052 | **0.3701** | 0.3181 | 0.3090 | **0.3710** |
| Average | 0.3405 | 0.3539 | **0.6165** | 0.3626 | 0.3865 | **0.6310** |

consistent comparison, we adopt the same data splits and set the contamination level to match the true anomaly ratio of each dataset.

Semi-supervised TAD methods are commonly designed for scenarios in which only normal samples are available during training. Consequently, their performance tends to rely on clean training data and may degrade substantially when the training set contains anomalous instances. In our evaluation setting, the contamination levels are identical to the natural anomaly ratios of the datasets, which introduces a degree of noise that these methods are not optimized to handle.

Across the benchmark, ProFiT, using five labeled anomalies per dataset, achieves stronger performance than MCM and DRL on most datasets. Detailed results are provided in Table 7. These observations indicate that the proposed proxy-based fine-tuning strategy remains effective under weak supervision and naturally contaminated training conditions.

# I ADDITIONAL RESULTS

In Table 8 we display the F1 performance of ours method and comparing methods. Tables 9 to 14 show the AUCPR and F1 performance on different number of labeled anomalies, which is the detailed results of Table 2.

Table 8: F1 score and average Rank of all methods across different datasets, the numbers of labeled anomalies is 5.

| Dataset | READ | DevNet | DeepSAD | PReNet | RoSAS | FeaWAD | iForest | DeepSVDD | ProFiT |
|---|---|---|---|---|---|---|---|---|---|
| ALOI | 0.0571 | 0.0659 | **0.0879** | 0.0286 | 0.0615 | 0.0571 | 0.033 | 0.0593 | 0.0308 |
| Annthyroid | 0.6185 | 0.6705 | 0.2197 | 0.1792 | 0.5376 | 0.289 | 0.3006 | 0.2081 | **0.7168** |
| Breastw | 0.7361 | **0.9589** | 0.7945 | 0.4521 | 0.4247 | **0.9589** | 0.9315 | 0.7917 | **0.9589** |
| Cardio | 0.4643 | 0.5714 | 0.125 | 0.2321 | 0.3393 | **0.6786** | 0.5179 | 0.5893 | 0.5893 |
| Cardiotocography | 0.5319 | 0.5248 | 0.2766 | 0.234 | 0.3688 | 0.5957 | 0.4397 | 0.4823 | **0.6099** |
| Celeba | 0.1659 | 0.1071 | 0.1134 | † | 0.1848 | **0.2456** | 0.119 | 0.1155 | 0.0455 |
| Census | 0.1469 | 0.0876 | 0.1619 | † | 0.1715 | **0.1762** | 0.056 | 0.0708 | 0.1608 |
| Donors | **0.8793** | 0.1519 | 0.2694 | † | 0.6569 | 0.6098 | 0.1016 | 0.2094 | 0.86 |
| Fault | 0.3971 | 0.311 | **0.4641** | 0.3397 | 0.4115 | 0.2105 | 0.4019 | 0.3254 | 0.4258 |
| Http | **0.9985** | 0.9854 | 0.981 | † | 0.992 | 0.0127 | 0.9854 | 0.0 | 0.9882 |
| InternetAds | 0.4737 | 0.3053 | 0.2632 | 0.2105 | 0.3579 | **0.5579** | 0.4842 | 0.3263 | **0.5579** |
| Ionosphere | 0.7297 | 0.6757 | 0.7297 | 0.5405 | 0.5405 | 0.4324 | 0.7027 | 0.5946 | **0.7838** |
| Landsat | 0.3885 | **0.4185** | 0.3484 | 0.2306 | 0.3559 | 0.2581 | 0.1654 | 0.2581 | 0.3659 |
| Letter | 0.2121 | 0.2424 | 0.2424 | **0.3333** | 0.2121 | 0.0303 | 0.1818 | 0.1515 | 0.0303 |
| Magic | 0.541 | 0.3883 | 0.5035 | 0.4447 | 0.5306 | 0.5138 | 0.5395 | **0.5904** | 0.4852 |
| Mammography | 0.4941 | 0.4588 | 0.2588 | 0.2706 | 0.4235 | 0.4706 | 0.2471 | 0.3059 | **0.5412** |
| Mnist | 0.4811 | 0.2358 | 0.2406 | 0.1226 | 0.2877 | **0.5849** | 0.316 | 0.3255 | 0.3632 |
| Optdigits | **0.9535** | 0.9302 | 0.2326 | 0.2558 | 0.6279 | 0.2326 | 0.0465 | 0.0 | 0.814 |
| PageBlocks | **0.5417** | 0.4306 | 0.4097 | 0.2222 | 0.2708 | 0.3264 | 0.4306 | 0.5069 | **0.5417** |
| Pendigits | 0.8636 | 0.7727 | 0.2727 | 0.3409 | **0.9091** | 0.6136 | 0.3182 | 0.0 | 0.7273 |
| Pima | 0.4304 | 0.519 | 0.3165 | 0.3924 | 0.3418 | 0.5063 | 0.5316 | 0.5316 | **0.6329** |
| Satellite | 0.5721 | 0.5817 | 0.3349 | 0.4872 | 0.4359 | 0.1827 | 0.5849 | 0.4744 | **0.6619** |
| Satimage-2 | 0.7826 | **0.8696** | 0.3043 | 0.5217 | 0.5217 | **0.8696** | **0.8696** | 0.2609 | **0.8696** |
| Shuttle | 0.7116 | 0.4861 | 0.2957 | 0.1543 | 0.72 | 0.9187 | **0.9529** | 0.9492 | 0.951 |
| Skin | 0.8048 | 0.6602 | 0.2745 | † | **0.9014** | 0.5854 | 0.1214 | 0.0581 | 0.7826 |
| Smtp | **0.6154** | 0.2308 | 0.4615 | 0.4615 | 0.2308 | **0.6154** | 0.0 | 0.4615 | 0.4615 |
| SpamBase | 0.5576 | 0.5535 | 0.3778 | 0.4101 | 0.4808 | 0.3313 | 0.5192 | 0.398 | **0.7636** |
| Thyroid | **0.8889** | 0.7778 | 0.2593 | 0.5926 | 0.6667 | 0.4074 | 0.6296 | 0.2593 | **0.8889** |
| Vertebral | 0.5455 | 0.2727 | 0.2727 | 0.5455 | 0.4545 | 0.3636 | 0.0 | 0.0 | **0.6364** |
| WBC | **1.0** | 0.6667 | 0.6667 | 0.3333 | 0.3333 | 0.6667 | **1.0** | **1.0** | **1.0** |
| WDBC | 0.6667 | 0.3333 | 0.3333 | 0.6667 | 0.6667 | **1.0** | 0.6667 | 0.6667 | **1.0** |
| Wilt | 0.4286 | 0.5119 | 0.0595 | 0.2381 | **0.7262** | 0.0 | 0.0119 | 0.0 | 0.5714 |
| Wine | **1.0** | **1.0** | **1.0** | **1.0** | **1.0** | **1.0** | 0.0 | 0.0 | **1.0** |
| WPBC | 0.4 | 0.4 | 0.4 | 0.4 | **0.4667** | 0.2667 | 0.2 | 0.2667 | 0.3333 |
| Yeast | **0.487** | 0.461 | 0.3182 | 0.3961 | 0.4416 | 0.3701 | 0.2727 | 0.3182 | 0.3701 |
| Average | 0.5876 | 0.5224 | 0.362 | 0.3679 | 0.4872 | 0.4554 | 0.3908 | 0.3302 | **0.6148** |
| Average Rank | 2.9429 | 4.0571 | 5.8 | 6.6 | 4.4286 | 4.6 | 5.4286 | 5.8286 | **2.6857** |

† Indicates that no result was available within 12 hours.

Table 9: AUCPR and average Rank of all methods across different datasets, the numbers of labeled anomalies is 10.

| Dataset | READ | DevNet | DeepSAD | PReNet | RoSAS | FeaWAD | iForest | DeepSVDD | ProFiT |
|---|---|---|---|---|---|---|---|---|---|
| ALOI | 0.0404 | 0.0442 | **0.0779** | 0.0388 | 0.0352 | 0.0357 | 0.0352 | 0.0421 | 0.0315 |
| Annthyroid | **0.6962** | 0.5295 | 0.22 | 0.3671 | 0.6738 | 0.2122 | 0.3393 | 0.1917 | 0.4782 |
| Breastw | 0.7079 | 0.757 | 0.8183 | 0.6128 | 0.5862 | 0.223 | 0.9841 | 0.8743 | **0.9924** |
| Cardio | **0.9008** | 0.8285 | 0.3372 | 0.4417 | 0.5687 | 0.7718 | 0.5551 | 0.6053 | 0.8093 |
| Cardiotocography | 0.5864 | 0.5741 | 0.3273 | 0.3439 | 0.4998 | **0.6942** | 0.4312 | 0.4359 | 0.6384 |
| Celeba | 0.1642 | 0.0386 | 0.0806 | † | 0.1138 | 0.2192 | 0.0739 | 0.0762 | **0.2309** |
| Census | 0.0942 | 0.0502 | 0.1084 | † | 0.0885 | 0.1347 | 0.0812 | 0.0829 | **0.1555** |
| Donors | **0.6831** | 0.2024 | 0.2561 | † | 0.5906 | 0.458 | 0.1306 | 0.2046 | 0.6770 |
| Fault | 0.4908 | **0.4992** | 0.48 | 0.4417 | 0.4684 | 0.417 | 0.4068 | 0.3387 | 0.4262 |
| Http | **0.9988** | 0.9908 | 0.9928 | † | 0.9985 | **0.9985** | 0.9884 | 0.379 | 0.9853 |
| InternetAds | 0.2723 | 0.3615 | 0.3101 | 0.2325 | 0.2797 | 0.4218 | **0.5318** | 0.2836 | 0.3787 |
| Ionosphere | 0.7203 | 0.9435 | 0.8801 | 0.7149 | 0.6438 | 0.6635 | 0.8121 | 0.7265 | **0.9653** |
| Landsat | **0.5439** | 0.4283 | 0.4004 | 0.2817 | 0.4895 | 0.1688 | 0.1825 | 0.3059 | 0.2755 |
| Letter | 0.3446 | 0.2816 | **0.357** | 0.3019 | 0.3296 | 0.0782 | 0.1284 | 0.1136 | 0.0995 |
| Magic | 0.5715 | 0.4718 | 0.5518 | 0.3535 | 0.4072 | 0.5331 | 0.6351 | 0.6445 | **0.7202** |
| Mammography | **0.4968** | 0.4076 | 0.1122 | 0.2042 | 0.3421 | 0.4366 | 0.2295 | 0.201 | 0.3948 |
| Mnist | 0.5431 | 0.5125 | 0.3084 | 0.1524 | 0.5086 | 0.7266 | 0.2667 | 0.2937 | **0.7319** |
| Optdigits | 0.9738 | **0.9989** | 0.3838 | 0.302 | 0.7482 | 0.8064 | 0.0583 | 0.0257 | 0.9777 |
| PageBlocks | 0.5753 | 0.5934 | 0.3472 | 0.2724 | 0.3555 | **0.7253** | 0.5231 | 0.5431 | 0.5958 |
| Pendigits | **0.964** | 0.9511 | 0.8129 | 0.397 | 0.9451 | 0.0373 | 0.2044 | 0.1068 | 0.8193 |
| Pima | 0.438 | 0.4538 | 0.4645 | 0.4167 | 0.4709 | 0.4518 | 0.5318 | 0.546 | **0.6329** |
| Satellite | 0.6918 | **0.8211** | 0.4552 | 0.3913 | 0.5146 | 0.5371 | 0.6895 | 0.5706 | 0.7992 |
| Satimage-2 | 0.8864 | **0.8873** | 0.6293 | 0.6171 | 0.7966 | 0.877 | 0.879 | 0.1916 | 0.8797 |
| Shuttle | 0.956 | 0.9756 | 0.6493 | 0.2961 | 0.9059 | 0.9573 | **0.9783** | 0.9121 | 0.9557 |
| Skin | **0.9822** | 0.8602 | 0.3098 | † | 0.9654 | 0.3354 | 0.2609 | 0.1852 | 0.8401 |
| Smtp | 0.4772 | 0.1427 | 0.0888 | 0.2309 | 0.0541 | **0.5984** | 0.006 | 0.3423 | 0.5414 |
| SpamBase | 0.8297 | 0.6862 | 0.3963 | 0.4545 | 0.6015 | **0.8341** | 0.5061 | 0.3929 | 0.6590 |
| Thyroid | 0.8699 | 0.8237 | 0.2154 | 0.5044 | 0.7658 | 0.8764 | 0.559 | 0.274 | **0.8798** |
| Vertebral | 0.6297 | 0.3738 | 0.4761 | 0.6071 | **0.6894** | 0.1369 | 0.1241 | 0.1047 | 0.6004 |
| WBC | **1.0** | 0.8667 | 0.9167 | 0.6667 | 0.4603 | **1.0** | **1.0** | **1.0** | **1.0** |
| WDBC | 0.9167 | **1.0** | 0.8667 | 0.8333 | 0.9167 | **1.0** | 0.8333 | 0.8095 | **1.0** |
| Wilt | 0.3712 | 0.5305 | 0.0941 | 0.3659 | **0.8232** | 0.0549 | 0.0469 | 0.0366 | 0.2691 |
| Wine | **1.0** | **1.0** | **1.0** | **1.0** | **1.0** | **1.0** | 0.2143 | 0.1288 | **1.0** |
| WPBC | 0.3244 | 0.3546 | 0.4058 | 0.3509 | 0.3714 | 0.2422 | 0.2583 | 0.2774 | **0.5412** |
| Yeast | **0.4577** | 0.4572 | 0.374 | 0.3334 | 0.4476 | 0.3657 | 0.3107 | 0.3103 | 0.3925 |
| Average | 0.6343 | 0.5914 | 0.4430 | 0.4176 | 0.5559 | 0.5151 | 0.4227 | 0.3588 | **0.6393** |
| Average Rank | **2.9429** | 3.6571 | 5.4571 | 7.0 | 4.7429 | 4.5429 | 6.0 | 6.4857 | 3.0857 |

† Indicates that no result was available within 12 hours.

## J    ABLATION DETAILS

Tables 15 to 20 present the AUCPR and F1 score of MotherNet and ProFiT fine-tuning under different numbers of labeled anomalous samples.

Table 10: F1 score and average Rank of all methods across different datasets, the numbers of labeled anomalies is 10.

| Dataset | READ | DevNet | DeepSAD | PReNet | RoSAS | FeaWAD | iForest | DeepSVDD | ProFiT |
|---|---|---|---|---|---|---|---|---|---|
| ALOI | 0.0615 | 0.0703 | **0.1033** | 0.0396 | 0.0505 | 0.0527 | 0.033 | 0.0593 | 0.033 |
| Annthyroid | **0.6532** | 0.5954 | 0.2832 | 0.3642 | **0.6532** | 0.2601 | 0.3006 | 0.2081 | 0.4509 |
| Breastw | 0.6377 | 0.6712 | 0.7808 | 0.4658 | 0.5139 | 0.0274 | 0.9315 | 0.7917 | **0.9589** |
| Cardio | **0.8036** | 0.6964 | 0.3393 | 0.3929 | 0.5179 | 0.6071 | 0.5179 | 0.5893 | 0.6964 |
| Cardiotocography | 0.5603 | 0.5532 | 0.3121 | 0.305 | 0.4539 | **0.6099** | 0.4397 | 0.4823 | 0.5319 |
| Celeba | 0.2393 | 0.0609 | 0.1337 | † | 0.2246 | 0.2701 | 0.119 | 0.1155 | **0.2988** |
| Census | 0.143 | 0.0555 | 0.1482 | † | 0.1343 | 0.1641 | 0.056 | 0.0708 | **0.1919** |
| Donors | 0.7378 | 0.2378 | 0.2413 | † | 0.6079 | 0.4887 | 0.1016 | 0.2094 | **0.7379** |
| Fault | 0.445 | **0.5024** | 0.4593 | 0.4417 | 0.4593 | 0.3301 | 0.4019 | 0.3254 | 0.4641 |
| Http | 0.9927 | 0.9869 | 0.981 | † | **0.9985** | 0.9963 | 0.9854 | 0.0 | 0.9912 |
| InternetAds | 0.3158 | 0.3368 | 0.2632 | 0.2421 | 0.2842 | 0.4632 | **0.4842** | 0.3263 | 0.3789 |
| Ionosphere | 0.6486 | **0.8649** | 0.8378 | 0.5946 | 0.5946 | 0.6486 | 0.7027 | 0.5946 | **0.8649** |
| Landsat | **0.4862** | 0.3258 | 0.396 | 0.2581 | 0.4511 | 0.1754 | 0.1654 | 0.2581 | 0.3509 |
| Letter | 0.3636 | 0.3333 | **0.4242** | 0.303 | 0.3333 | 0.0606 | 0.1818 | 0.1515 | 0.1212 |
| Magic | 0.5583 | 0.4412 | 0.5 | 0.3468 | 0.3765 | 0.4896 | 0.5395 | 0.5904 | **0.6042** |
| Mammography | **0.5412** | 0.4588 | 0.2689 | 0.3176 | 0.3882 | 0.4588 | 0.2471 | 0.3059 | 0.3882 |
| Mnist | 0.5 | 0.5047 | 0.3302 | 0.1368 | 0.5472 | **0.6698** | 0.316 | 0.3255 | 0.6415 |
| Optdigits | 0.9302 | **0.9767** | 0.3488 | 0.3023 | 0.6047 | 0.8372 | 0.0465 | 0.0 | 0.9535 |
| PageBlocks | 0.5486 | 0.5694 | 0.3333 | 0.2847 | 0.3958 | **0.625** | 0.4306 | 0.5069 | 0.6042 |
| Pendigits | **0.9318** | **0.9318** | 0.7727 | 0.3864 | 0.9091 | 0.0227 | 0.3182 | 0.0 | 0.7045 |
| Pima | 0.4177 | 0.443 | 0.4557 | 0.3671 | 0.4304 | 0.5443 | 0.5316 | 0.5316 | **0.6076** |
| Satellite | 0.641 | 0.7115 | 0.4231 | 0.3446 | 0.492 | 0.4327 | 0.5849 | 0.4744 | **0.7244** |
| Satimage-2 | **0.8696** | **0.8696** | 0.6522 | 0.6087 | 0.7826 | **0.8696** | **0.8696** | 0.2609 | **0.8696** |
| Shuttle | 0.951 | **0.9575** | 0.5961 | 0.2634 | 0.8429 | 0.9529 | 0.9529 | 0.9492 | 0.9455 |
| Skin | 0.921 | 0.8454 | 0.2994 | † | **0.9361** | 0.2916 | 0.2609 | 0.1852 | 0.7945 |
| Smtp | **0.6154** | 0.0 | 0.1538 | 0.4615 | 0.0 | **0.6154** | 0.0 | 0.4615 | 0.5385 |
| SpamBase | 0.7636 | 0.6061 | 0.3737 | 0.4182 | 0.5495 | **0.7939** | 0.5192 | 0.398 | 0.604 |
| Thyroid | **0.8519** | **0.8519** | 0.2593 | 0.6296 | 0.8148 | **0.8519** | 0.6296 | 0.2593 | 0.7778 |
| Vertebral | **0.6364** | 0.3636 | 0.4545 | 0.5455 | 0.5455 | 0.0 | 0.0 | 0.0 | **0.6364** |
| WBC | **1.0** | 0.6667 | 0.6667 | 0.3333 | 0.3333 | **1.0** | **1.0** | **1.0** | **1.0** |
| WDBC | 0.6667 | **1.0** | 0.6667 | 0.6667 | 0.6667 | **1.0** | 0.6667 | 0.6667 | **1.0** |
| Wilt | 0.4524 | 0.5952 | 0.119 | 0.3333 | **0.7262** | 0.0357 | 0.0119 | 0.0 | 0.2619 |
| Wine | **1.0** | **1.0** | **1.0** | **1.0** | **1.0** | **1.0** | 0.0 | 0.0 | **1.0** |
| WPBC | 0.3333 | 0.2 | 0.2 | 0.2 | 0.2667 | 0.2667 | 0.2 | 0.2667 | **0.4667** |
| Yeast | **0.4935** | **0.4935** | 0.3831 | 0.3117 | 0.4675 | 0.3571 | 0.2727 | 0.3182 | 0.4026 |
| Average | **0.6203** | 0.5651 | 0.4274 | 0.3888 | 0.5252 | 0.4926 | 0.3948 | 0.3338 | 0.6170 |
| Average Rank | **2.7429** | 3.4571 | 5.5143 | 7.0571 | 4.3714 | 4.0286 | 5.8286 | 6.2 | 2.8286 |

† Indicates that no result was available within 12 hours.

Table 11: AUCPR and average Rank of all methods across different datasets, the numbers of labeled anomalies is 20.

| Dataset | READ | DevNet | DeepSAD | PReNet | RoSAS | FeaWAD | iForest | DeepSVDD | ProFiT |
|---|---|---|---|---|---|---|---|---|---|
| ALOI | 0.0491 | 0.0454 | **0.0651** | 0.0412 | 0.0491 | 0.0481 | 0.0352 | 0.0421 | 0.0452 |
| Annthyroid | 0.7412 | 0.6686 | 0.3531 | 0.3866 | 0.6879 | 0.3571 | 0.3393 | 0.1917 | **0.7841** |
| Breastw | 0.8009 | **0.9958** | 0.8481 | 0.5802 | 0.724 | 0.9743 | 0.9841 | 0.8743 | 0.9827 |
| Cardio | **0.9013** | 0.8524 | 0.4797 | 0.5166 | 0.6203 | 0.8847 | 0.5551 | 0.6053 | 0.8058 |
| Cardiotocography | 0.7429 | **0.8304** | 0.4213 | 0.4193 | 0.4881 | 0.7129 | 0.4312 | 0.4359 | 0.6522 |
| Celeba | 0.2198 | 0.0238 | 0.0732 | † | 0.1176 | 0.2173 | 0.0739 | 0.0762 | **0.2451** |
| Census | 0.2003 | 0.0688 | 0.0999 | † | 0.1574 | 0.2152 | 0.0812 | 0.0829 | **0.2537** |
| Donors | **0.965** | 0.3928 | 0.7455 | † | 0.7033 | 0.6194 | 0.1306 | 0.2046 | 0.8912 |
| Fault | 0.568 | 0.5813 | 0.4847 | 0.4567 | 0.5717 | 0.4299 | 0.4068 | 0.3387 | **0.6689** |
| Http | 0.9987 | 0.9897 | 0.9927 | † | 0.9985 | **0.9993** | 0.9884 | 0.379 | 0.9853 |
| InternetAds | 0.5291 | 0.6 | 0.4516 | 0.4529 | 0.4797 | 0.5371 | 0.5318 | 0.2836 | **0.8166** |
| Ionosphere | 0.8051 | 0.8872 | **0.9246** | 0.8045 | 0.7577 | 0.8036 | 0.8121 | 0.7265 | 0.8847 |
| Landsat | **0.5601** | 0.516 | 0.3655 | 0.3703 | 0.5063 | 0.3498 | 0.1825 | 0.3059 | 0.2897 |
| Letter | 0.3568 | 0.319 | **0.5015** | 0.3157 | 0.4311 | 0.1297 | 0.1284 | 0.1136 | 0.1542 |
| Magic | 0.6849 | 0.5497 | 0.5584 | 0.365 | 0.5521 | 0.6265 | 0.6351 | 0.6445 | **0.7501** |
| Mammography | **0.5206** | 0.369 | 0.1748 | 0.3394 | 0.4173 | 0.4788 | 0.2295 | 0.201 | 0.4518 |
| Mnist | 0.6706 | 0.5323 | 0.2591 | 0.2275 | 0.4319 | 0.6958 | 0.2667 | 0.2937 | **0.7868** |
| Optdigits | 0.9929 | **0.9969** | 0.5844 | 0.3338 | 0.9395 | 0.0814 | 0.0583 | 0.0257 | 0.9853 |
| PageBlocks | **0.7517** | 0.5676 | 0.407 | 0.3945 | 0.5557 | 0.6732 | 0.5231 | 0.5431 | 0.7465 |
| Pendigits | **0.9667** | 0.9638 | 0.6168 | 0.8006 | 0.9495 | 0.6224 | 0.2044 | 0.1068 | 0.6598 |
| Pima | 0.508 | **0.6763** | 0.5373 | 0.4815 | 0.5604 | 0.5936 | 0.5318 | 0.546 | 0.6590 |
| Satellite | 0.7136 | 0.7846 | 0.6053 | 0.4264 | 0.5524 | 0.6263 | 0.6895 | 0.5706 | **0.8105** |
| Satimage-2 | 0.8979 | **0.9017** | 0.5909 | 0.7791 | 0.8189 | 0.8783 | 0.879 | 0.1916 | 0.8494 |
| Shuttle | 0.9567 | 0.9586 | 0.8314 | 0.3964 | 0.9615 | 0.9552 | 0.9783 | 0.9121 | **0.9833** |
| Skin | 0.7328 | 0.8383 | 0.5576 | † | 0.8926 | 0.4551 | 0.2609 | 0.2852 | 0.7565 |
| Smtp | **0.4769** | 0.3301 | 0.3318 | 0.2763 | 0.0702 | 0.476 | 0.006 | 0.3423 | 0.3112 |
| SpamBase | 0.625 | 0.5953 | 0.4457 | 0.4701 | 0.615 | 0.5796 | 0.5061 | 0.3929 | **0.8998** |
| Thyroid | 0.8507 | 0.9199 | 0.519 | 0.7945 | **0.9515** | 0.7437 | 0.559 | 0.274 | 0.9049 |
| Vertebral | 0.5911 | 0.251 | 0.4853 | 0.6152 | 0.65 | 0.1989 | 0.1241 | 0.1047 | **0.6801** |
| WBC | **1.0** | 0.9167 | 0.8095 | 0.7 | 0.4321 | 0.9167 | **1.0** | **1.0** | **1.0** |
| WDBC | **1.0** | **1.0** | **1.0** | **1.0** | **1.0** | **1.0** | 0.8333 | 0.8095 | **1.0** |
| Wilt | 0.815 | 0.7543 | 0.3973 | 0.4935 | 0.8749 | 0.0624 | 0.0469 | 0.0366 | **0.9115** |
| Wine | **1.0** | **1.0** | **1.0** | **1.0** | **1.0** | **1.0** | 0.2143 | 0.1288 | **1.0** |
| WPBC | 0.4307 | 0.3081 | 0.4392 | 0.3606 | **0.4993** | 0.2278 | 0.2583 | 0.2774 | 0.4134 |
| Yeast | **0.4011** | 0.3981 | 0.3862 | 0.3679 | 0.3879 | 0.3628 | 0.3107 | 0.3103 | 0.3725 |
| Average | 0.6864 | 0.6395 | 0.5241 | 0.4989 | 0.6116 | 0.5581 | 0.4227 | 0.3616 | **0.6969** |
| Average Rank | **2.6571** | 3.5143 | 5.4857 | 6.7714 | 4.0286 | 4.5429 | 6.5429 | 7.2 | 2.8286 |

† Indicates that no result was available within 12 hours.

Table 12: F1 score and average Rank of all methods across different datasets, the numbers of labeled anomalies is 20.

| Dataset | READ | DevNet | DeepSAD | PReNet | RoSAS | FeaWAD | iForest | DeepSVDD | ProFiT |
|---|---|---|---|---|---|---|---|---|---|
| ALOI | 0.0637 | 0.0659 | **0.1011** | 0.0505 | 0.0879 | 0.0681 | 0.033 | 0.0593 | 0.0703 |
| Annthyroid | 0.6879 | 0.6647 | 0.3699 | 0.3815 | 0.6532 | 0.3237 | 0.3006 | 0.2081 | **0.7399** |
| Breastw | 0.726 | **0.9589** | 0.7671 | 0.4521 | 0.6759 | **0.9589** | 0.9315 | 0.7917 | **0.9589** |
| Cardio | **0.8214** | 0.75 | 0.4286 | 0.4821 | 0.5893 | 0.7857 | 0.5179 | 0.5893 | 0.6964 |
| Cardiotocography | 0.6809 | **0.7234** | 0.4184 | 0.3333 | 0.4894 | **0.7234** | 0.4397 | 0.4823 | 0.6596 |
| Celeba | 0.2827 | 0.0308 | 0.133 | † | 0.1819 | 0.2771 | 0.119 | 0.1155 | **0.2954** |
| Census | 0.2557 | 0.074 | 0.1473 | † | 0.24 | 0.2739 | 0.056 | 0.0708 | **0.2999** |
| Donors | **0.9537** | 0.3806 | 0.6845 | † | 0.6776 | 0.6822 | 0.1016 | 0.2094 | 0.8565 |
| Fault | 0.5311 | 0.5455 | 0.4833 | 0.3923 | 0.5885 | 0.4593 | 0.4019 | 0.3254 | **0.6077** |
| Http | **0.9985** | 0.9883 | 0.9723 | † | **0.9985** | **0.9985** | 0.9854 | 0.0 | 0.9919 |
| InternetAds | 0.4632 | 0.5474 | 0.3895 | 0.3895 | 0.4421 | 0.4737 | 0.4842 | 0.3263 | **0.7263** |
| Ionosphere | 0.7027 | 0.7838 | 0.8108 | 0.7297 | 0.6757 | 0.7297 | 0.7027 | 0.5946 | **0.8378** |
| Landsat | 0.594 | **0.6216** | 0.3333 | 0.3158 | 0.5338 | 0.2456 | 0.1654 | 0.2581 | 0.3308 |
| Letter | 0.3939 | 0.3939 | **0.4848** | 0.4242 | 0.4545 | 0.1212 | 0.1818 | 0.1515 | 0.2121 |
| Magic | 0.5983 | 0.5311 | 0.5119 | 0.33 | 0.5104 | 0.542 | 0.5395 | 0.5904 | **0.6655** |
| Mammography | **0.5765** | 0.4118 | 0.3206 | 0.4118 | 0.4706 | 0.4941 | 0.2471 | 0.3059 | 0.5176 |
| Mnist | 0.6509 | 0.6038 | 0.2972 | 0.217 | 0.4623 | 0.6226 | 0.316 | 0.3255 | **0.7547** |
| Optdigits | **0.9767** | 0.9535 | 0.5116 | 0.3023 | 0.8372 | 0.0 | 0.0465 | 0.0 | **0.9767** |
| PageBlocks | **0.7153** | 0.6389 | 0.3819 | 0.4514 | 0.5347 | 0.6181 | 0.4306 | 0.5069 | 0.6319 |
| Pendigits | 0.9091 | **0.9545** | 0.6136 | 0.7955 | 0.9318 | 0.5909 | 0.3182 | 0.0 | 0.6136 |
| Pima | 0.557 | **0.6709** | 0.4684 | 0.4304 | 0.5316 | 0.5443 | 0.5316 | 0.5316 | 0.5949 |
| Satellite | 0.6218 | **0.7356** | 0.5481 | 0.3798 | 0.5128 | 0.524 | 0.5849 | 0.4744 | 0.6907 |
| Satimage-2 | **0.8696** | **0.8696** | 0.6087 | 0.7391 | 0.7826 | **0.8696** | **0.8696** | 0.2609 | 0.7826 |
| Shuttle | 0.9686 | 0.9769 | 0.8142 | 0.3595 | **0.9806** | 0.9464 | 0.9529 | 0.9492 | 0.9584 |
| Skin | 0.8336 | 0.8307 | 0.4928 | † | **0.8987** | 0.5103 | 0.2609 | 0.1852 | 0.7878 |
| Smtp | **0.6154** | 0.5385 | 0.4615 | 0.5385 | 0.0 | **0.6154** | 0.0 | 0.4615 | 0.3077 |
| SpamBase | 0.5596 | 0.4808 | 0.3919 | 0.3879 | 0.5677 | 0.5758 | 0.5192 | 0.398 | **0.8081** |
| Thyroid | 0.7778 | **0.8889** | 0.4815 | 0.7778 | 0.8519 | 0.6667 | 0.6296 | 0.2593 | 0.8148 |
| Vertebral | 0.4545 | 0.1818 | 0.3636 | 0.5455 | 0.4545 | 0.1818 | 0.0 | 0.0 | **0.6364** |
| WBC | **1.0** | 0.6667 | 0.6667 | 0.3333 | 0.3333 | 0.6667 | **1.0** | **1.0** | **1.0** |
| WDBC | **1.0** | **1.0** | **1.0** | **1.0** | **1.0** | **1.0** | 0.6667 | 0.6667 | **1.0** |
| Wilt | 0.7381 | 0.7024 | 0.3571 | 0.4643 | 0.7857 | 0.0119 | 0.0119 | 0.0 | **0.8095** |
| Wine | **1.0** | **1.0** | **1.0** | **1.0** | **1.0** | **1.0** | 0.0 | 0.0 | **1.0** |
| WPBC | 0.4 | 0.3333 | 0.2667 | 0.3333 | **0.4667** | 0.2 | 0.2 | 0.2667 | 0.3333 |
| Yeast | 0.3896 | **0.4351** | 0.3701 | 0.3831 | 0.3766 | 0.3701 | 0.2727 | 0.3182 | 0.3896 |
| Average | **0.6677** | 0.6267 | 0.4986 | 0.4711 | 0.5879 | 0.5335 | 0.3948 | 0.3338 | 0.6674 |
| Average Rank | 2.6571 | 3.2286 | 5.6286 | 6.4857 | 4.0286 | 4.3429 | 6.5714 | 7.0857 | **2.4571** |

† Indicates that no result was available within 12 hours.

Table 13: AUCPR and average Rank of all methods across different datasets, the numbers of labeled anomalies is 30.

| Dataset | READ | DevNet | DeepSAD | PReNet | RoSAS | FeaWAD | iForest | DeepSVDD | ProFiT |
|---|---|---|---|---|---|---|---|---|---|
| ALOI | **0.0582** | 0.0443 | 0.0325 | 0.0397 | 0.0506 | 0.0474 | 0.0352 | 0.0421 | 0.0366 |
| Annthyroid | 0.6872 | 0.3153 | 0.7898 | 0.6371 | 0.3669 | 0.5817 | 0.3393 | 0.1917 | **0.8269** |
| Breastw | 0.9358 | 0.978 | 0.9523 | **0.9959** | 0.9403 | 0.7888 | 0.9841 | 0.8743 | 0.9781 |
| Cardio | 0.7212 | 0.9035 | 0.8771 | 0.8462 | 0.5113 | 0.5929 | 0.5551 | 0.6053 | **0.9196** |
| Cardiotocography | 0.5998 | 0.6261 | 0.7352 | 0.6988 | 0.3796 | 0.536 | 0.4312 | 0.4359 | **0.8151** |
| Celeba | 0.2123 | 0.0363 | 0.1019 | † | 0.1494 | 0.1985 | 0.0739 | 0.0762 | **0.2436** |
| Census | 0.105 | 0.1717 | 0.1914 | † | 0.0999 | 0.1716 | 0.0812 | 0.0829 | **0.2588** |
| Donors | 0.8137 | 0.9598 | 0.5913 | † | 0.9672 | 0.6787 | 0.1306 | 0.2046 | **0.9751** |
| Fault | 0.5335 | 0.428 | **0.5662** | 0.556 | 0.4948 | 0.4479 | 0.4068 | 0.3387 | 0.5282 |
| Http | 0.9985 | **0.9997** | 0.9991 | † | 0.9863 | 0.9985 | 0.9884 | 0.379 | 0.9851 |
| InternetAds | 0.3046 | **0.5899** | 0.5448 | 0.4438 | 0.3446 | 0.3174 | 0.5318 | 0.2836 | 0.5838 |
| Ionosphere | 0.8187 | 0.7768 | 0.8817 | **0.9646** | 0.9431 | 0.853 | 0.8121 | 0.7265 | 0.9478 |
| Landsat | 0.454 | 0.2015 | 0.5131 | **0.5299** | 0.4195 | 0.2812 | 0.1825 | 0.3059 | 0.3406 |
| Letter | 0.5554 | 0.0613 | 0.5936 | 0.4468 | **0.6766** | 0.5716 | 0.1284 | 0.1136 | 0.2896 |
| Magic | 0.5499 | 0.5811 | 0.6002 | 0.5818 | 0.5722 | 0.3953 | 0.6351 | 0.6445 | **0.7665** |
| Mammography | 0.4894 | 0.446 | **0.5073** | 0.3895 | 0.2314 | 0.3682 | 0.2295 | 0.201 | 0.4195 |
| Mnist | 0.6346 | 0.556 | **0.7627** | 0.6489 | 0.3368 | 0.2661 | 0.2667 | 0.2937 | 0.7401 |
| Optdigits | 0.9704 | 0.9325 | **0.9984** | 0.9962 | 0.675 | 0.5771 | 0.0583 | 0.0257 | 0.9795 |
| PageBlocks | 0.5465 | 0.6985 | **0.718** | 0.5446 | 0.5126 | 0.4025 | 0.5231 | 0.5431 | 0.6651 |
| Pendigits | **0.9764** | 0.1976 | 0.9692 | 0.9681 | 0.7529 | 0.8827 | 0.2044 | 0.1068 | 0.7104 |
| Pima | 0.5468 | 0.5476 | 0.5891 | **0.6702** | 0.5319 | 0.466 | 0.5318 | 0.546 | 0.6678 |
| Satellite | 0.5335 | 0.6737 | 0.7508 | 0.805 | 0.6106 | 0.4393 | 0.6895 | 0.5706 | **0.8375** |
| Satimage-2 | 0.8439 | 0.8798 | 0.8985 | **0.9054** | 0.8539 | 0.7976 | 0.879 | 0.1916 | 0.8734 |
| Shuttle | **0.98** | 0.9536 | 0.9552 | 0.9696 | 0.7924 | 0.6203 | 0.9783 | 0.9121 | 0.9628 |
| Skin | 0.8139 | 0.8941 | 0.7763 | † | **0.9137** | 0.5164 | 0.2609 | 0.1852 | 0.8943 |
| Smtp | 0.0708 | 0.6164 | 0.3599 | 0.209 | **0.6194** | 0.2549 | 0.006 | 0.3423 | 0.0315 |
| SpamBase | 0.5873 | 0.4547 | 0.7026 | 0.7109 | 0.4445 | 0.407 | 0.5061 | 0.3929 | **0.9024** |
| Thyroid | 0.8102 | 0.8925 | 0.8923 | 0.8674 | 0.6363 | 0.7103 | 0.559 | 0.274 | **0.9112** |
| Vertebral | 0.7003 | 0.1785 | 0.7021 | 0.3461 | 0.6552 | 0.7401 | 0.1241 | 0.1047 | **0.7536** |
| WBC | 0.4321 | 0.9167 | **1.0** | 0.9167 | 0.8095 | 0.7 | **1.0** | **1.0** | **1.0** |
| WDBC | **1.0** | **1.0** | **1.0** | **1.0** | **1.0** | **1.0** | 0.8333 | 0.8095 | **1.0** |
| Wilt | 0.8772 | 0.0611 | 0.8122 | 0.5717 | 0.5104 | 0.8383 | 0.0469 | 0.0366 | **0.8831** |
| Wine | **1.0** | **1.0** | **1.0** | **1.0** | **1.0** | **1.0** | 0.2143 | 0.1288 | **1.0** |
| WPBC | 0.4522 | 0.2425 | 0.3951 | 0.3736 | **0.4816** | 0.4249 | 0.2583 | 0.2774 | 0.4718 |
| Yeast | **0.468** | 0.3402 | 0.4263 | 0.435 | 0.3465 | 0.4104 | 0.3107 | 0.3103 | 0.4176 |
| Average | 0.6309 | 0.5759 | 0.6910 | 0.6690 | 0.5891 | 0.5509 | 0.4227 | 0.3588 | **0.7033** |
| Average Rank | 4.1143 | 4.7714 | 2.9429 | 4.1429 | 5.0857 | 5.7429 | 6.6571 | 7.3714 | **2.7429** |

† Indicates that no result was available within 12 hours.

Table 14: F1 score and average Rank of all methods across different datasets, the number of labeled anomalies is 30.

| Dataset | READ | DevNet | DeepSAD | PReNet | RoSAS | FeaWAD | iForest | DeepSVDD | ProFiT |
|---|---|---|---|---|---|---|---|---|---|
| ALOI | 0.0879 | 0.0681 | 0.044 | 0.0571 | **0.0945** | 0.0879 | 0.033 | 0.0593 | 0.0352 |
| Annthyroid | 0.7052 | 0.2948 | 0.7283 | 0.6301 | 0.3699 | 0.6243 | 0.3006 | 0.2081 | **0.7861** |
| Breastw | 0.8493 | **0.9589** | 0.8904 | **0.9589** | 0.8767 | 0.6986 | 0.9315 | 0.7917 | **0.9589** |
| Cardio | 0.6964 | 0.7857 | 0.8036 | 0.7143 | 0.4286 | 0.5357 | 0.5179 | 0.5893 | **0.8214** |
| Cardiotocography | 0.5674 | 0.6454 | 0.6525 | 0.7021 | 0.3688 | 0.4894 | 0.4397 | 0.4823 | **0.7801** |
| Celeba | **0.2891** | 0.0825 | 0.2099 | † | 0.2099 | 0.2792 | 0.119 | 0.1155 | 0.2778 |
| Census | 0.1685 | 0.1913 | 0.2496 | 0.1396 | 0.1409 | 0.0 | 0.056 | 0.0708 | **0.3088** |
| Donors | 0.8721 | 0.9057 | 0.5532 | † | **0.9264** | 0.8038 | 0.1016 | 0.2094 | 0.8834 |
| Fault | 0.5455 | 0.4402 | 0.5455 | **0.5646** | 0.445 | 0.4067 | 0.4019 | 0.3254 | 0.4689 |
| Http | 0.9971 | **0.9985** | 0.9956 | 0.9854 | 0.9839 | 0.0 | 0.9854 | 0.0 | 0.9912 |
| InternetAds | 0.3158 | 0.5474 | 0.4947 | 0.3895 | 0.3474 | 0.2947 | 0.4842 | 0.3263 | **0.5684** |
| Ionosphere | 0.7297 | 0.7297 | 0.8378 | **0.8919** | 0.8378 | 0.7838 | 0.7027 | 0.5946 | 0.8649 |
| Landsat | 0.4612 | 0.2231 | 0.5038 | 0.5238 | 0.4085 | 0.2506 | 0.1654 | 0.2581 | 0.3534 |
| Letter | 0.5758 | 0.0 | 0.6061 | 0.4545 | **0.6667** | 0.5152 | 0.1818 | 0.1515 | 0.303 |
| Magic | 0.5464 | 0.5627 | 0.586 | 0.5677 | 0.5272 | 0.3622 | 0.5395 | 0.5904 | **0.6853** |
| Mammography | 0.5294 | 0.4706 | **0.5529** | 0.4941 | 0.2706 | 0.4118 | 0.2471 | 0.3059 | 0.4471 |
| Mnist | 0.5943 | 0.5755 | **0.717** | 0.6321 | 0.3632 | 0.2311 | 0.316 | 0.3255 | 0.6981 |
| Optdigits | 0.9535 | 0.907 | 0.9767 | 0.9767 | 0.6977 | 0.5349 | 0.0465 | 0.0 | 0.9535 |
| PageBlocks | 0.5417 | 0.6319 | **0.6736** | 0.5833 | 0.5278 | 0.4375 | 0.4306 | 0.5069 | 0.6528 |
| Pendigits | **0.9545** | 0.2273 | 0.9318 | **0.9545** | 0.75 | 0.8636 | 0.3182 | 0.0 | 0.5909 |
| Pima | 0.557 | 0.5316 | 0.5443 | 0.5823 | 0.481 | 0.4684 | 0.5316 | 0.5316 | **0.5949** |
| Satellite | 0.4984 | 0.5321 | 0.7147 | **0.7612** | 0.5929 | 0.367 | 0.5849 | 0.4744 | 0.7179 |
| Satimage-2 | 0.7826 | **0.8696** | **0.8696** | **0.8696** | 0.8261 | 0.7391 | **0.8696** | 0.2609 | **0.8696** |
| Shuttle | **0.9695** | 0.9445 | 0.9538 | 0.9658 | 0.7218 | 0.5712 | 0.9529 | 0.9492 | 0.9233 |
| Skin | 0.8701 | 0.9065 | 0.7449 | † | **0.9472** | 0.6651 | 0.2609 | 0.1852 | 0.8169 |
| Smtp | 0.0 | **0.6154** | 0.4615 | 0.0 | 0.5385 | 0.0769 | 0.0 | 0.4615 | 0.4615 |
| SpamBase | 0.5374 | 0.4323 | 0.6687 | 0.6545 | 0.4 | 0.402 | 0.5192 | 0.398 | **0.8384** |
| Thyroid | 0.7778 | 0.8519 | 0.8889 | **0.9259** | 0.5556 | 0.7778 | 0.6296 | 0.2593 | 0.8148 |
| Vertebral | 0.5455 | 0.0909 | 0.5455 | 0.3636 | **0.6364** | **0.6364** | 0.0 | 0.0 | **0.6364** |
| WBC | 0.3333 | 0.6667 | **1.0** | 0.6667 | 0.6667 | 0.3333 | **1.0** | **1.0** | **1.0** |
| WDBC | **1.0** | **1.0** | **1.0** | **1.0** | **1.0** | **1.0** | 0.6667 | 0.6667 | **1.0** |
| Wilt | **0.8571** | 0.0595 | 0.7857 | 0.5476 | 0.5 | 0.8214 | 0.0119 | 0.0 | 0.7976 |
| Wine | **1.0** | **1.0** | **1.0** | **1.0** | **1.0** | **1.0** | 0.0 | 0.0 | **1.0** |
| WPBC | **0.4667** | 0.2 | 0.4 | 0.2667 | 0.4 | 0.3333 | 0.2 | 0.2667 | 0.4 |
| Yeast | **0.474** | 0.3182 | 0.4545 | 0.461 | 0.3247 | 0.4351 | 0.2727 | 0.3182 | 0.461 |
| Average | 0.6186 | 0.5504 | 0.6739 | 0.6339 | 0.5666 | 0.4925 | 0.3948 | 0.3338 | **0.6789** |
| Average Rank | 3.7143 | 4.6 | 2.7429 | 3.6286 | 4.9429 | 6.0 | 6.8 | 7.0571 | **2.6857** |

† Indicates that no result was available within 12 hours.

Table 15: AUCPR and F1 before and after ProFiT. the number of labeled anomalies is 1.

| Dataset | F1 Score | | AUC PR | |
|---|---|---|---|---|
| | MotherNet | ProFiT | MotherNet | ProFiT |
| ALOI | 0.0242 | 0.0264 | 0.0313 | 0.0311 |
| Annthyroid | 0.1156 | 0.0906 | 0.1597 | 0.1321 |
| Backdoor | 0.1961 | 0.1627 | 0.1753 | 0.1596 |
| Breastw | 0.2887 | 0.2887 | 0.3451 | 0.3451 |
| Cardiotocography | 0.5426 | 0.5035 | 0.5906 | 0.5263 |
| Census | 0.0908 | 0.0805 | 0.0791 | 0.0838 |
| Cover | 0.0734 | 0.1352 | 0.0451 | 0.0725 |
| Donors | 0.4619 | 0.4619 | 0.3276 | 0.3276 |
| Fault | 0.2823 | 0.3038 | 0.3341 | 0.3370 |
| InternetAds | 0.1825 | 0.1649 | 0.1821 | 0.1909 |
| Ionosphere | 0.5676 | 0.5676 | 0.6789 | 0.5808 |
| Letter | 0.1010 | 0.0707 | 0.0985 | 0.0673 |
| Lymphography | 0.1667 | 0.1667 | 0.2203 | 0.2110 |
| Mnist | 0.1274 | 0.1557 | 0.1146 | 0.1288 |
| PageBlocks | 0.1042 | 0.1296 | 0.1097 | 0.1492 |
| Satellite | 0.2447 | 0.3072 | 0.2679 | 0.3379 |
| Shuttle | 0.1325 | 0.1017 | 0.1836 | 0.1189 |
| Skin | 0.4577 | 0.4577 | 0.4602 | 0.4602 |
| Vowels | 0.0476 | 0.0238 | 0.1065 | 0.0253 |
| WPBC | 0.2667 | 0.1333 | 0.2890 | 0.2126 |
| Average | **0.2237** | 0.2166 | **0.2400** | 0.2249 |

Table 16: AUCPR and F1 before and after ProFiT. the number of labeled anomalies is 2.

| Dataset | F1 Score | | AUC PR | |
|---|---|---|---|---|
| | MotherNet | ProFiT | MotherNet | ProFiT |
| ALOI | 0.0315 | 0.0234 | 0.0325 | 0.0329 |
| Annthyroid | 0.3237 | 0.2871 | 0.3148 | 0.2594 |
| Backdoor | 0.4032 | 0.4261 | 0.3834 | 0.3384 |
| Breastw | 0.9521 | 0.9521 | 0.9835 | 0.9835 |
| Cardiotocography | 0.4184 | 0.4752 | 0.4349 | 0.4636 |
| Census | 0.1168 | 0.1772 | 0.0958 | 0.1399 |
| Cover | 0.2911 | 0.3589 | 0.2657 | 0.3350 |
| Donors | 0.2746 | 0.2746 | 0.3241 | 0.3241 |
| Fault | 0.4083 | 0.4067 | 0.4337 | 0.4358 |
| InternetAds | 0.2596 | 0.2351 | 0.2588 | 0.2362 |
| Ionosphere | 0.3694 | 0.2883 | 0.4252 | 0.3491 |
| Letter | 0.0404 | 0.0505 | 0.0772 | 0.0827 |
| Lymphography | 0.1667 | 0.1667 | 0.2558 | 0.2057 |
| Mnist | 0.1179 | 0.0723 | 0.1156 | 0.0972 |
| PageBlocks | 0.375 | 0.4028 | 0.3281 | 0.3938 |
| Satellite | 0.4316 | 0.4087 | 0.5166 | 0.4799 |
| Shuttle | 0.5342 | 0.5702 | 0.5968 | 0.6063 |
| Skin | 0.6751 | 0.6751 | 0.6944 | 0.6944 |
| Vowels | 0.3810 | 0.3571 | 0.4262 | 0.4104 |
| WPBC | 0.2444 | 0.3556 | 0.3407 | 0.3729 |
| Average | 0.3408 | **0.3482** | **0.3652** | 0.3621 |

Table 17: AUCPR and F1 before and after ProFiT. the number of labeled anomalies is 4.

| Dataset | F1 Score | | AUC PR | |
|---|---|---|---|---|
| | MotherNet | ProFiT | MotherNet | ProFiT |
| ALOI | 0.0505 | 0.0418 | 0.0377 | 0.0336 |
| Annthyroid | 0.5260 | 0.5395 | 0.5123 | 0.5404 |
| Backdoor | 0.6112 | 0.7279 | 0.6398 | 0.7152 |
| Breastw | 0.9589 | 0.9589 | 0.9854 | 0.9854 |
| Cardiotocography | 0.5626 | 0.5816 | 0.5829 | 0.5919 |
| Census | 0.1286 | 0.1938 | 0.1023 | 0.1565 |
| Cover | 0.6005 | 0.6559 | 0.6580 | 0.6923 |
| Donors | 0.5640 | 0.5640 | 0.6336 | 0.6336 |
| Fault | 0.3732 | 0.4083 | 0.3919 | 0.4068 |
| InternetAds | 0.3018 | 0.3649 | 0.2943 | 0.4107 |
| Ionosphere | 0.7658 | 0.7658 | 0.7420 | 0.8153 |
| Letter | 0.0404 | 0.0606 | 0.0755 | 0.0892 |
| Lymphography | 0.5 | 0.6667 | 0.6778 | 0.8611 |
| Mnist | 0.2531 | 0.1965 | 0.2386 | 0.2219 |
| PageBlocks | 0.4398 | 0.3889 | 0.4115 | 0.3770 |
| Satellite | 0.6277 | 0.6378 | 0.7005 | 0.7189 |
| Shuttle | 0.9156 | 0.9489 | 0.9173 | 0.9413 |
| Skin | 0.7905 | 0.7905 | 0.7587 | 0.7587 |
| Vowels | 0.3810 | 0.4286 | 0.4030 | 0.4479 |
| WPBC | 0.3111 | 0.3556 | 0.3566 | 0.3909 |
| Average | 0.4851 | **0.5138** | 0.5060 | **0.5394** |

Table 18: AUCPR and F1 before and after ProFiT. the number of labeled anomalies is 8.

| Dataset | F1 Score | | AUC PR | |
|---|---|---|---|---|
| | MotherNet | ProFiT | MotherNet | ProFiT |
| ALOI | 0.0330 | 0.0308 | 0.0347 | 0.0336 |
| Annthyroid | 0.6397 | 0.6590 | 0.6685 | 0.6750 |
| Backdoor | 0.6772 | 0.7599 | 0.7207 | 0.6875 |
| Breastw | 0.9589 | 0.9589 | 0.9884 | 0.9884 |
| Cardiotocography | 0.6217 | 0.5768 | 0.7026 | 0.6771 |
| Census | 0.1824 | 0.2054 | 0.1345 | 0.1624 |
| Cover | 0.7488 | 0.8010 | 0.8471 | 0.8948 |
| Donors | 0.7454 | 0.7454 | 0.8140 | 0.8140 |
| Fault | 0.4705 | 0.4801 | 0.4589 | 0.4766 |
| InternetAds | 0.3509 | 0.5333 | 0.3581 | 0.5675 |
| Ionosphere | 0.7838 | 0.8198 | 0.8184 | 0.8682 |
| Letter | 0.1111 | 0.1515 | 0.1216 | 0.1394 |
| Lymphography | 0.5 | 0.5 | 0.6389 | 0.6865 |
| Mnist | 0.5047 | 0.5425 | 0.5368 | 0.5631 |
| PageBlocks | 0.6204 | 0.6134 | 0.6836 | 0.6558 |
| Satellite | 0.6544 | 0.6410 | 0.7612 | 0.7606 |
| Shuttle | 0.8669 | 0.9547 | 0.9284 | 0.9553 |
| Skin | 0.7864 | 0.7864 | 0.7952 | 0.7952 |
| Vowels | 0.5714 | 0.6190 | 0.5864 | 0.6191 |
| WPBC | 0.3556 | 0.4889 | 0.3854 | 0.5232 |
| Average | 0.5592 | **0.5934** | 0.5992 | **0.6272** |

Table 19: AUCPR and F1 before and after ProFiT. the number of labeled anomalies is 16.

| Dataset | F1 Score | | AUC PR | |
|---|---|---|---|---|
| | MotherNet | ProFiT | MotherNet | ProFiT |
| ALOI | 0.0381 | 0.0359 | 0.0383 | 0.0399 |
| Annthyroid | 0.6667 | 0.6744 | 0.7133 | 0.7263 |
| Backdoor | 0.5352 | 0.7274 | 0.5750 | 0.7810 |
| Breastw | 0.9589 | 0.9589 | 0.9926 | 0.9926 |
| Cardiotocography | 0.6572 | 0.6336 | 0.6834 | 0.6844 |
| Census | 0.2504 | 0.2173 | 0.1926 | 0.1785 |
| Cover | 0.8517 | 0.8772 | 0.9381 | 0.9537 |
| Donors | 0.8394 | 0.8394 | 0.8973 | 0.8973 |
| Fault | 0.5199 | 0.5199 | 0.5305 | 0.5407 |
| InternetAds | 0.4281 | 0.5509 | 0.4194 | 0.5675 |
| Ionosphere | 0.8018 | 0.8468 | 0.8907 | 0.9114 |
| Letter | 0.1515 | 0.1616 | 0.1523 | 0.1617 |
| Lymphography | 0.5 | 0.5 | 0.6389 | 0.6778 |
| Mnist | 0.6572 | 0.7060 | 0.7238 | 0.7617 |
| PageBlocks | 0.6597 | 0.6227 | 0.6990 | 0.6806 |
| Satellite | 0.7051 | 0.6998 | 0.8179 | 0.8197 |
| Shuttle | 0.9473 | 0.9455 | 0.9681 | 0.9633 |
| Skin | 0.7953 | 0.7953 | 0.7865 | 0.7865 |
| Vowels | 0.6429 | 0.6905 | 0.7248 | 0.7643 |
| WPBC | 0.4222 | 0.3778 | 0.4054 | 0.5153 |
| Average | 0.6014 | **0.6190** | 0.6394 | **0.6702** |

Table 20: AUCPR and F1 before and after ProFiT. the number of labeled anomalies is 32.

| Dataset | F1 Score | | AUC PR | |
|---|---|---|---|---|
| | MotherNet | ProFiT | MotherNet | ProFiT |
| ALOI | 0.0344 | 0.0432 | 0.0378 | 0.0426 |
| Annthyroid | 0.6994 | 0.7071 | 0.7346 | 0.7465 |
| Backdoor | 0.6011 | 0.7474 | 0.6894 | 0.7501 |
| Breastw | 0.9589 | 0.9589 | 0.9879 | 0.9879 |
| Cardiotocography | 0.7329 | 0.7376 | 0.8050 | 0.8029 |
| Census | 0.2954 | 0.2978 | 0.2469 | 0.2496 |
| Cover | 0.8405 | 0.8624 | 0.9274 | 0.9440 |
| Donors | 0.8642 | 0.8642 | 0.9321 | 0.9321 |
| Fault | 0.5502 | 0.5678 | 0.5683 | 0.5842 |
| InternetAds | 0.4982 | 0.6667 | 0.5341 | 0.6928 |
| Ionosphere | 0.8649 | 0.9189 | 0.9392 | 0.9631 |
| Letter | 0.2828 | 0.3030 | 0.2420 | 0.2478 |
| Lymphography | 0.5 | 0.5 | 0.5833 | 0.7056 |
| Mnist | 0.7343 | 0.7547 | 0.8133 | 0.8237 |
| PageBlocks | 0.6782 | 0.6713 | 0.7017 | 0.7506 |
| Satellite | 0.7276 | 0.7254 | 0.8457 | 0.8448 |
| Shuttle | 0.9680 | 0.9658 | 0.9918 | 0.9816 |
| Skin | 0.7996 | 0.7996 | 0.8283 | 0.8283 |
| Vowels | 0.7857 | 0.7857 | 0.8754 | 0.8580 |
| WPBC | 0.4222 | 0.5111 | 0.5273 | 0.5505 |
| Average | 0.6419 | **0.6694** | 0.6906 | **0.7143** |

