# OpenReview forum: "ProFit: Unsupervised Fine-Tuning of Tabular Models via  Proxy Tasks for Label-Scarce Anomaly Detection"
_ICLR.cc/2026/Conference — Submitted to ICLR 2026_

### Official Review · Reviewer_rJk6 · 2025-10-15

**Soundness:** 3
**Presentation:** 3
**Contribution:** 2
**Rating:** 6
**Confidence:** 4

**Summary:**

This paper proposes a novel approach for unsupervised fine-tuning of existing Tabular In-Context Learning Models for anomaly detection. The proposed method focuses on the weakly supervised setting where one disposes of a small subset of anomalies and a large, unlabeled dataset.
In short, the proposed approach, ProFIT, relies on proxy tasks to refine the prediction ability of the MotherNet model. For a $d$-dimensional feature space, a target feature $t \in 1,...,d$ is selected and serves as the proxy-task's label. Among the $d$-1 remaining features, the authors propose a feature selection method that identifies the most relevant subset of features that will serve to predict the target feature.

A double-objective is used to fine-tune the TICL model based on those proxy tasks: (i) a standard cross-entropy loss on the selected target feature $t$ and (ii) an alignment loss that ensures that the logits on the same target feature $t$ produced by two sets of features $S_1$ and $S_2$ are close to each other.

The experimental results on a significant benchmark of tabular datasets demonstrate the relevance of the proposed approach, as it shows strong performance based on several threshold-dependent and non-threshold-dependent metrics.

**Strengths:**

S1. The paper is **well written** and easy to follow.

S2. **Experiments are sound and extensive**. The dataset benchmark is quite large, and the methods to which ProFIT is compared are quite numerous.

S3. **Ablation** studies are limited but **relevant**.

S4. ProFIT is **theoretically based** and well-founded.

**Weaknesses:**

**W1**. While minor, there are a few typos in the paper that are worth mentioning: 1) Line 103, Tabular In Context **Learing** Model.; 2) Line 201, $ \mathbf{X}_i = g_j(\mathbf{u}, \epsilon_i) $, where does this $j$ come from? 3) On Tables 11-16 MotherNet is referred to as MotehrNet.

**W2**. As mentioned in S1, the paper is well-written; however, there is **key information missing from the main text of the paper**.
- 1) Unless I am wrong, nowhere is it mentioned which model is fine-tuned in the experiments that are presented in the paper. One must refer to the appendix to find this crucial information.
- 2)  The authors mention how they are interested in the weakly supervised anomaly detection set-up, and even define on line 132 $D_L$ (which is never reused before page 17 of the appendix), but never explain how those labeled anomalies are used.
- 3) Overall, section 3.2. lacks crucial details on the methods.

**W3**. The proposed pretext task is a special case of mask reconstruction.

**W4**. This work is not currently reproducible, although I acknowledge that the authors plan to release the code upon acceptance.

**Questions:**

**Q1**. As mentioned in **W1**, the paper may require some clarifications to make the setup easier to understand. Overall, the authors should include Appendix B in the main text of the paper, replacing less crucial sections, such as Section 3.5.
Regarding the overall method, I have a few questions:
- (i) How are the labeled data samples used in this context? Only as in-context samples for the TICL model? If so, it might be worth mentioning it somewhere in the main part of the paper.
- (ii) Figure 1 suggests that there might be overlapping features between the two subsets used to predict the same selected target features. However, Algorithm 1 suggests otherwise. Could the authors explain which approach they used?

**Q2** It appears to me that ProFIT should be better suited for datasets with a medium to large number of features. In particular, the performance on datasets with very few features, e.g., http (3) or Skin (3), could be attributed to the sole performance of the MotherNet model. This claim is supported by Tables 11 to 16, where for the Skin dataset, there is no difference in performance between the MotherNet model and the fine-tuned one using your approach. I recommend discussing this in section 5, investigating performance variations on datasets like http, Skin, Smtp, or Wilt. Additionally, including the non-fine-tuned MotherNet model in the main tables and comparison would further underscore the relevance of your method.

**Q3**: Could you elaborate on the difference between the proposed approach and vanilla mask reconstruction? It seems to me that it consists in masking $l$ feature among the original $d$ features and predicting only one of them. Why should it work better than predicting the entire set of $l$ features? It appears, as mentioned in **W4**, as a special case of mask reconstruction.

**Q4**: (More open question) Why do the authors restrain their approach to unsupervised fine-tuning of TICL models for anomaly detection, as it seems that their approach could be applied to a much broader self-supervised setup (for representation learning, for example)?


Overall, I lean towards acceptance, and I am open to increasing my score should my questions and concerns be addressed.

---

> ### Author Response · Authors · 2025-11-23
>
> Thanks for your questions ! I will respond to your questions one by one.
>
> **Q1 Corrections and Clarifications**
>
> We thank the reviewer for the helpful comments. All typos and information mentioned in W1 and W2 have been corrected in the revised manuscript. For Q1, we have clarified the role of labeled data in the main text and updated the description to explicitly state that they are only used as in-context examples. We have also revised Algorithm 1 to remove the inconsistency noted by the reviewer and moved part of Appendix B into the main paper to improve clarity.

---

> ### Author Response · Authors · 2025-11-23
>
> **Q2: Why does ProFIT perform poorly on datasets with very few features (e.g., only 3 features)? Do these results mainly reflect the performance of the MotherNet model rather than the advantages of the ProFIT framework itself?**
>
> Thank you for your insightful comment. Following the comment, we conducted a more detailed investigation into the applicability of ProFiT by analyzing two key factors that jointly influence its effectiveness:
>
> 1. the presence of meaningful latent factor structure, estimated via FactorAnalysis;
> 2. the number of features available in the dataset.
>
>
> To address this concern, we first quantify how well the latent factor assumption holds on each dataset. We introduce a new metric, Latent Factor Strength (LFS), defined as follows:
> 1. We compute the original correlation, i.e., the mean absolute pairwise correlation of the raw feature correlation matrix.
> 2. We then perform Factor Analysis to extract latent factors and reconstruct the features. Based on the reconstruction residuals, we compute the residual correlation, i.e., the mean absolute pairwise correlation of the residual features.
> 3. We define
>
> $$
> LFS = 1- \frac{\text{residual correlation}}{\text{original correlation}},
> $$
>
> which measures the proportion of the original feature dependence that can be explained by latent factors. A higher LFS indicates that the latent factor assumption is better satisfied.
>
> By combining the latent-factor estimates, feature dimensionality, and empirical performance variations, we observed the following patterns:
>
> 1. **Clear latent factor structure + sufficient number of features**
>
> In datasets such as Lymphography and WPBC, FactorAnalysis reveals clear latent factor structure, and the feature dimensionality is adequate. In this regime, ProFiT brings significant performance improvements, since proxy tasks can effectively exploit both the underlying structure and the richer feature space.
>
> 2. **Clear latent factor structure + very few features**
>
> Datasets such as breastw and Skin exhibit a clear latent structure, but the number of features is extremely small. Although ProFiT can leverage the latent relationships, the limited feature dimensionality restricts the expressiveness of proxy tasks. As a result, the improvement over the base MotherNet is small, and in some cases (e.g., Skin), the performance is nearly identical before and after fine-tuning.
>
> 3. **No clear latent factor structure + sufficient number of features**
>
> Datasets like InternetAds, Backdoor, and Census have weak or unclear latent factor structure according to FactorAnalysis.
> However, because they contain a relatively large number of features, ProFiT is still able to construct diverse and informative proxy tasks. Consequently, we observe notable performance gains despite the lack of strong latent structure.
>
> 4. **No clear latent factor structure + very few features**
>
> In datasets such as ALOI and Shuttle, where both latent structure is weak and the number of features is limited, ProFiT brings little or no improvement, and in rare cases performance may even decrease slightly.
>
> The lack of latent structure and limited feature space jointly restrict the model’s ability to extract additional useful information beyond the pretrained MotherNet.
>
> Across all datasets, our findings show that ProFiT is particularly effective when:
>
> - The dataset contains meaningful latent factor structure; or
> - The feature dimensionality is sufficiently high to support diverse proxy-task construction.
>
> These insights have now been incorporated into Section 5 of the revised manuscript. We thank you again for encouraging us to make these applicability conditions explicit.
>
> | Datasets      | Dim  | Original Corr | Residual Corr | LFS    | F1 Impr. | PR Impr. |
> |---------------|------|---------------|---------------|--------|----------|----------|
> | Lymphography  | 18   | 0.1680        | 0.1075        | 36.01% | 0.0278   | 0.0554   |
> | WPBC          | 33   | 0.2905        | 0.1290        | 55.59% | 0.0334   | 0.0435   |
> | Breastw       | 9    | 0.6019        | 0.1372        | 77.21% | 0.0000   | 0.0000   |
> | Skin          | 3    | 0.6961        | 0.4181        | 39.94% | 0.0000   | 0.0000   |
> | InternetAds   | 1555 | 0.0183        | 0.0183        | 0.00%  | 0.0825   | 0.1031   |
> | Backdoor      | 196  | 0.1061        | 0.1392        | -31.20%| 0.0879   | 0.0414   |
> | Census        | 500  | 0.0297        | 0.0273        | 8.08%  | 0.0179   | 0.0199   |
> | OI            | 27   | 0.0946        | 0.0870        | 8.03%  | -0.0017  | 0.0002   |
> | Shuttle       | 9    | 0.1885        | 0.1778        | 5.68%  | 0.0204   | -0.0032  |

---

> ### Author Response · Authors · 2025-11-23
>
> **Q3 Difference between mask reconstruction task**
>
> Our approach is indeed inspired by self-supervised masked reconstruction, but it differs fundamentally from vanilla mask reconstruction in both formulation and purpose.
>
> First, vanilla mask reconstruction methods (e.g., MCM) aim to reconstruct  masked features and typically **rely on architectures** specifically designed for generative reconstruction. These approaches cannot be directly applied to TICL models, whose architecture requires a support–query interaction mechanism and does not naturally support full-feature generative reconstruction during fine-tuning.
>
> In contrast, ProFiT is explicitly designed to be compatible with the TICL paradigm. Instead of reconstructing all masked features, we formulate **a single-feature prediction task that fits seamlessly into the TICL support–query structure** and can be used for efficient fine-tuning without modifying the model architecture. Therefore, ProFiT is not merely a special case of standard mask reconstruction, but a proxy task tailored to the constraints and objectives of TICL-based pretrained models.
>
> Most importantly, our theoretical analysis shows that this proxy-task formulation leads to a fine-tuning objective that is **aligned with the original TICL learning**. This provides a principled justification for why ProFiT improves downstream performance, which is fundamentally different from the heuristic nature of generic mask reconstruction.
>
> **Q4: Extension to other setup**
>
> Our approach can indeed be applied to a broader range of self-supervised or representation learning scenarios. We chose to focus on unsupervised fine-tuning for anomaly detection primarily because TICL models were originally designed for support–query style tasks, and anomaly detection under extremely limited labels is a setting where this paradigm is particularly beneficial.
>
> To verify the broader applicability of our method, we additionally evaluated ProFiT on several general-purpose tabular classification benchmarks. We are conducting the corresponding experiments. Specifically, we selected several datasets from CC70 for evaluation. The results indicate that our approach can still improve model performance on non-anomaly-detection tabular datasets, suggesting that the method has a certain level of generality.
>
> | Dataset           | F1 MotherNet | F1 ProFiT | PR MotherNet | PR ProFiT |
> |-------------------|--------------|-----------|---------------|-----------|
> | PC4               | 0.3333       | 0.3889    | 0.3687        | 0.4584    |
> | KC2               | 0.4545       | 0.4545    | 0.4304        | 0.4735    |
> | KC1               | 0.3438       | 0.3438    | 0.3672        | 0.4150    |
> | PC1               | 0.4286       | 0.4286    | 0.5169        | 0.5304    |
> | BankMarketing     | 0.4178       | 0.4178    | 0.3786        | 0.3852    |
> | Nomao             | 0.7584       | 0.7422    | 0.7494        | 0.7948    |
> | Dresses Sales     | 0.4762       | 0.6667    | 0.5124        | 0.6795    |
> | Credit Approval   | 0.8462       | 0.7692    | 0.8673        | 0.8688    |
> | Sick              | 0.5217       | 0.6957    | 0.5638        | 0.7406    |
> | Bioresponse       | 0.6029       | 0.6029    | 0.5840        | 0.5974    |
> | Spambase          | 0.6868       | 0.6923    | 0.7911        | 0.8087    |
> | PhishingWebsites  | 0.8920       | 0.8920    | 0.9488        | 0.9563    |
> | Tic Tac Toe       | 0.7460       | 0.7302    | 0.7146        | 0.7327    |
> | **Average**       | 0.5776   | **0.6019**| 0.5995    | **0.6493**|

---

> ### Comment · Reviewer_rJk6 · 2025-11-25
> **Answer to Authors**
>
> Dear Authors,
>
> Thank you for adressing most of our concerns. In particular I believe that your answer to Q2 is a particularly insightful addition to your paper.
>
> As correctly mentioned by **Reviewer Bf7X**, there are some tabular AD methods that might be worth adding to your reference list and/or to the benchmark. In particular, MCM [1], NPT-AD [2], DRL [3], Disent-AD [4], ICL [5], NeuTraL-AD [6] ...etc. While those methods assume normal-only training sets, some of them have been evaluated in the presence of contamination.
>
> Nevertheless, I have updated my score to account for the additions to the original paper that address most of my concerns.
>
> [1] MCM: Masked Cell Modeling for Anomaly Detection in Tabular Data, ICLR 2024.
>
> [2] Beyond individual input for deep anomaly detection on tabular data , ICML 2024.
>
> [3] DRL: Decomposed Representation Learning for Tabular Anomaly Detection. ICLR 2025.
>
> [4] Disentangling Tabular Data Towards Better One-Class Anomaly Detection, AAAI 2025.
>
> [5] Anomaly Detection for Tabular Data with Internal Contrastive Learning, ICLR 2022.
>
> [6] Neural Transformation Learning for Anomaly Detection, ICML 2021

---

> > ### Author Response · Authors · 2025-11-25
> >
> > Dear Reviewer,
> >
> > Thank you very much for your follow-up feedback and for updating your score. We greatly appreciate your recognition of our revisions, including your positive comments on our response to Q2.
> >
> > We also acknowledge the additional related works mentioned in the discussion among reviewers. We will carefully integrate these relevant methods into our related work section and consider them in our broader comparison and revision.
> >
> > We will continue to address the remaining comments from all reviewers and further refine the manuscript to improve its clarity and completeness.
> >
> > Thank you again for your constructive feedback and support in helping us strengthen this work.

---

### Official Review · Reviewer_J37h · 2025-10-30

**Soundness:** 3
**Presentation:** 3
**Contribution:** 2
**Rating:** 2
**Confidence:** 2

**Summary:**

The paper introduced ProFiT, an unsupervised fine tuning framework that adapts pretrained tabular in context learning models to anomaly detection when labels are scarce by training on automatically constructed proxy tasks. By predicting a held out feature from mRMR selected and correlated feature subsets and enforcing cross subset consistency with a Jensen–Shannon divergence regularizer, ProFiT learns  representations that align with target domain structure without additional anomaly labels. Experiments on tabular anomaly detection benchmarks show that ProFiT outperforms weakly-supervised and unsupervised methods, as well as vanilla TICL models. ProFiT offers a practical way to improve tabular anomaly detection under limited labeled data conditions and vast amounts of unlabeled data.

**Strengths:**

1. Minimal Label Dependency for Real-World Applicability
ProFiT completes fine-tuning using only unlabeled target-domain data, fully addressing the core pain point of anomaly detection—“difficulty in obtaining labels (few anomalous samples and high annotation costs)”. Compared to weakly supervised methods (requiring a small number of labels) and traditional supervised methods (relying on large-scale labels), it has stronger adaptability to practical scenarios.
2. Resolution of “Distribution Misalignment” in Pretrained Models
Aiming at the key flaw of existing Tabular In-Context Learning (TICL) models—inductive biases formed on synthetic tasks are misaligned with the data distribution of real target domains—ProFiT enables the model to relearn the underlying structure of target data through “proxy tasks based on target-domain features”, significantly improving domain adaptation capability.
3. Consistency Regularization Enhances Model Robustness
ProFiT introduces a regularization term based on Jensen–Shannon divergence to force alignment of prediction results from two different proxy views. This design reduces biases from single proxy tasks, prevents the model from overfitting to local features, makes the learned features more general, and ultimately improves the stability of anomaly detection.

**Weaknesses:**

1. Strong Dependence on Inter-Feature Correlations in Data
The core logic of proxy tasks is “sampling some features as targets and using correlated features as inputs”. If the target-domain data has weak inter-feature correlations (e.g., multi-dimensional independent features) or a large number of redundant features, proxy tasks cannot be effectively constructed. As a result, the model fails to capture data structure, leading to ineffective fine-tuning.
2. Unclear Generalization to “Unknown Anomaly Types”
The paper only verifies performance on benchmark datasets, but in real scenarios, anomaly types are often “unknown (Out-of-Distribution)”. ProFiT detects anomalies by learning the structure of normal data; if unknown anomalies have little difference from the structure of normal data, its detection capability may decline significantly. This aspect is not verified in the abstract.
3. Higher Computational Cost Than Vanilla Pretrained Models
The proposed ProFiT needs to construct “multiple proxy tasks” and align predictions from two proxy views for consistency. Compared to directly using vanilla TICL models, the training process requires more computing resources and time—especially for high-dimensional tabular data, the cost increase may be more significant.

**Questions:**

I have two questions that I am particularly concerned about regarding this paper. 1. A limited discussion on the applicable boundaries of feature selection based on mRMR？The abstract mentions that ProFiT builds the agent task by predicting the reserved features of the relevant feature subset selected by mRMR, but it does not discuss the applicability boundary of the mRMR method itself. In practical scenarios, if the target table data has extreme features, such as extremely weak inter-feature correlations, high-dimensional data with a large number of sparse features, or nonlinear feature dependencies that are difficult to capture by mRMR (a method focused on linear/statistical correlations), the quality of the selected feature subset will decline. This directly affects the effectiveness of the proxy task and the model's ability to learn the target domain structure. However, this paper fails to address how to handle such edge cases or verify the stability of mRMR across different data types.
2. Lack of adaptation to dynamic or multi-source table scenarios. One key limitation of the current ProFiT framework is that it cannot handle dynamic or multi-source table data. In real-world anomaly detection (for example, real-time fraud prevention, dynamic risk control), data often exhibits concept drift (data distribution over time) or comes from multiple related tables (for example, the combination of user transaction data and user configuration data). At present, ProFiT, which is designed for static single-table data, cannot adapt to these common scenarios, greatly limiting its practical application scope in complex industrial environments.

---

> ### Author Response · Authors · 2025-11-23
>
> Thanks for your questions ! I will respond to your questions one by one.
>
> **Q1: Does the strong dependence on inter-feature correlations lead to the method failing on datasets with weak correlations between features?**
>
> To address this concern, we first quantify how well the latent factor assumption holds on each dataset. We introduce a new metric, Latent Factor Strength (LFS), defined as follows:
> 1. We compute the original correlation, i.e., the mean absolute pairwise correlation of the raw feature correlation matrix.
> 2. We then perform Factor Analysis to extract latent factors and reconstruct the features. Based on the reconstruction residuals, we compute the residual correlation, i.e., the mean absolute pairwise correlation of the residual features.
> 3. We define
>
> $$
> LFS = 1- \frac{\text{residual correlation}}{\text{original correlation}},
> $$
>
> which measures the proportion of the original feature dependence that can be explained by latent factors. A higher LFS indicates that the latent factor assumption is better satisfied.
>
> As shown in the following table or appendix E, we observe a clear relationship between this metric and the performance of our method:
> - On datasets with **strong latent factor structure** (high LFS), our method consistently yields significant improvements, which matches our theoretical expectation.
> - On datasets where the **latent factor strength is moderate or weak but the feature dimensionality is high**, the method still brings noticeable gains. In these cases, the rich feature space allows us to construct informative proxy tasks even though the latent structure is not very clean.
> - Only when **both the latent factor strength is weak and the feature dimensionality is small **do we see our method become less effective. In such datasets, the constructed proxy tasks are less expressive, and the performance may be close to or slightly below the baseline.
>
> Therefore, when the latent factor assumption does not hold strongly, we mainly see reduced gains rather than catastrophic degradation, and clear performance drops appear only in the corner case of weak latent factors combined with low-dimensional features.
>
> | Datasets      | Dim  | Original Corr | Residual Corr | LFS    | F1 Impr. | PR Impr. |
> |---------------|------|---------------|---------------|--------|----------|----------|
> | Lymphography  | 18   | 0.1680        | 0.1075        | 36.01% | 0.0278   | 0.0554   |
> | WPBC          | 33   | 0.2905        | 0.1290        | 55.59% | 0.0334   | 0.0435   |
> | Breastw       | 9    | 0.6019        | 0.1372        | 77.21% | 0.0000   | 0.0000   |
> | Skin          | 3    | 0.6961        | 0.4181        | 39.94% | 0.0000   | 0.0000   |
> | InternetAds   | 1555 | 0.0183        | 0.0183        | 0.00%  | 0.0825   | 0.1031   |
> | Backdoor      | 196  | 0.1061        | 0.1392        | -31.20%| 0.0879   | 0.0414   |
> | Census        | 500  | 0.0297        | 0.0273        | 8.08%  | 0.0179   | 0.0199   |
> | OI            | 27   | 0.0946        | 0.0870        | 8.03%  | -0.0017  | 0.0002   |
> | Shuttle       | 9    | 0.1885        | 0.1778        | 5.68%  | 0.0204   | -0.0032  |

---

> ### Author Response · Authors · 2025-11-23
>
> **Q2: Does the reliance on learning the structure of normal data in ProFiT affect its ability to detect unknown anomalies in real-world scenarios?**
>
> Our method is explicitly designed for scenarios involving **unknown anomalies**. During fine-tuning, while we learn feature dependencies from a large amount of unlabeled normal data, the **key advantage** of ProFiT is its ability to **generalize quickly** to OOD anomalies with just a few labeled samples during inference. This is possible because, once the model has been fine-tuned to capture the structure of the normal data, it can use minimal supervision to adapt to new, unseen anomalies at inference time. Thus, our method is well-suited for real-world scenarios where labeled data for new anomaly classes is scarce.
>
> **Q3: Does the need to construct multiple proxy tasks and align predictions in ProFiT lead to higher computational costs compared to vanilla pretrained models?**
>
> No, our method is specifically designed to reduce the computational cost of adapting models to new data. While the original MotherNet requires **four weeks of pretraining on an A100** to achieve good performance, our approach only requires **a few minutes of fine-tuning per dataset** to achieve better performance based on the pretrained MotherNet. Thus, ProFiT significantly lowers the data adaptation cost without incurring the high computational overhead of retraining a model from scratch.

---

### Official Review · Reviewer_Bf7X · 2025-10-30

**Soundness:** 2
**Presentation:** 2
**Contribution:** 2
**Rating:** 4
**Confidence:** 4

**Summary:**

This paper focuses on the task of weak-supervised tabular anomaly detection. The proposed method ProFit, adapts tabular foundation model to downstream anomaly detection tasks via proxy-based fine-tuning.

**Strengths:**

S1: Adapting tabular foundation model to downstream tasks is important.
S2: The paper structure is clear.

**Weaknesses:**

Reproducibility perspective:
W1: The source code is not provided.

Problem setting perspective:
W2: The current setting frames Tabular Anomaly Detection (TAD) as a supervised learning problem with sparse positive labels. However, it remains unclear why only the unlabeled samples predicted as normal by the detector are used during inference. What if the samples predicted as anomalous are also incorporated? Utilizing the full set of unlabeled data, including those flagged as anomalies, could potentially improve model robustness and performance.

W3: Given that the problem is formulated as binary classification, several powerful tabular models—such as XGBoost, CatBoost—could serve as strong baselines. These methods are well-established in tabular data benchmarks and should be included for a comprehensive comparison.

W4: Based on the above, what if the labeled true anomalies is removed? It seems the proposed framework can also be adapted to unsupervised setting by relying solely on pseudo-labels generated by an anomaly detector.

W5: Minor. Although the paper focuses on weakly-supervised learning, semi-supervised methods (MCM[1], NPT-AD[2], DRL[3])—which are widely used in TAD and assume only normal samples are available during training—should also be included as baselines. Many such methods have been evaluated under similar "contamination" settings.
[1] MCM: Masked Cell Modeling for Anomaly Detection in Tabular Data. ICLR’24.
[2] Beyond Individual Input for Deep Anomaly Detection on Tabular Data. ICML’24.
[3] DRL: Decomposed Representation Learning for Tabular Anomaly Detection. ICLR’25.

Technical perspective:
W6: Towards the confused contribution of foundation model and the proposed meta learning mechanism. Both tabular foundation model and this method do the similar thing, which first pretrain on diverse tasks, and then transfer to previously unseen downstream supervised tasks at inference (this paper will transfer weakly-supervised anomaly detection to supervised learning as shown in Line 910 of manuscript). It is known that foundation model like TabPFNv2 trained on synthetic data performs well on real-world tabular data, including low-data and imbalanced regimes. Why then do these foundation models not suffice for this setting? If the problem is reformulated as binary classification (as in Line 910), could fine-tuning TabPFNv2 directly (with supervised loss, not with proxy-based fine-tuning) yield competitive results? If not, this would indicate a fundamental gap in the transferability of foundation model knowledge to this specific task. Moreover, it seems that theoretical framework of proxy-based fine-tuning can train a model such as Transformer from scratch on a single dataset. Thus what is the explicit benefit of using a pretrained foundation model over training from scratch on the target dataset? It is insufficient to only compare whether proxy-based fine-tuning is used on MotherNet as illustrated in Fig.3. Both more detailed explanation and ablation results (more types of variants and other foundation models like TabPFNv2 as noted above) should be included to validate the effectiveness of the proposed method.

W7: The uniqueness of the method in the context of anomaly detection is not clearly established.  It seems that this method can work on any imbalance tabular data setting, what is the unique relationship with tabular anomaly detection? What is the performance of this method on other tabular imbalance learning setting?

W8: Towards the fragile of assumption. The theoretical analysis relies on several assumptions (e.g., latent factor model, task compatibility). These are strong assumptions that columns are independent conditioned on latent factor. How are these assumptions validated, especially under complex real-world data distributions?

W9: Towards the confused training objective. This method selects one feature column as target to prediction via minimizing the cross-entropy loss. However, tabular data including many numerical columns. If the selected column is continuous, cross-entropy may not be appropriate. How does the method handle continuous features?

W10: The meta learning mechanism, that selecting various columns as labels and the remainder as features is not novel, which is used by previous papers in tabular domain, e.g., GTL[4], STUNT[5].

[4] From Supervised to Generative: A Novel Paradigm for Tabular Deep Learning with Large Language Models. KDD’24.
[5] STUNT: FEW-SHOT TABULAR LEARNING WITH SELF-GENERATED TASKS FROM UNLABELED TABLES. ICLR’23.

W11: The choice of foundation model and anomaly detector is not thoroughly justified. Why was MotherNet selected over other tabular foundation models (e.g., TabPFNv2, TabPFN)? Ablation studies with alternative models would help validate the generality of the approach. Similarly, why was iForest chosen as the pseudo-labeler? Would the results hold with other detectors?

W12: The number of labeled anomalies. For datasets with very few anomalies (e.g., WBC, WDBC, Wine), using 5 anomalous samples may already represent half of all available anomalies. Is this a fair comparison with other detection methods?

**Questions:**

Please see the weaknesses above.

---

> ### Author Response · Authors · 2025-11-23
>
> We thank the reviewer for the detailed and technically insightful comments. However, several key misunderstandings appear to underlie the concerns, and we clarify them below to facilitate further discussion.
>
> **1. Foundation models such as TabPFNv2 and our work address fundamentally different problems.**
>
>  TabPFN-like models aim to build universal supervised tabular models with strong out-of-the-box performance on downstream supervised tasks. In contrast, our work focuses on a setting where **no labeled downstream data is available**, and the goal is to improve downstream anomaly detection performance without any supervised fine-tuning. Thus, although both frameworks involve pretraining and transfer, the transfer scenarios are categorically different:
> - Foundation models assume availability of supervised downstream labels.
> - Our framework must operate in a **weakly supervised / label-deficient regime**, where only a few labeled anomalies are given (and sometimes none).
>
> **2. Why not directly fine-tune TabPFNv2 with supervised loss?**
>
>  Fine-tuning TabPFNv2 indeed improves its performance **when sufficient labeled data is available**, which agrees with prior literature. However, in the low-label and highly imbalanced setting of weakly supervised anomaly detection, supervised fine-tuning of foundation models **fails to fully exploit downstream structure**, and their gains are limited.
>
> **3. Why not train a Transformer from scratch using the proposed proxy-based mechanism?**
>
>  In principle, the proposed proxy-based fine-tuning framework can train a model from scratch. However, the cost is prohibitive:
> - MotherNet was pretrained for **four weeks on an 80GB A100**, which is a high training cost.
> - Foundation models already provide reasonable feature priors. Starting from these pretrained models dramatically reduces training costs while enabling substantial performance gains through ProFiT.
>
> Thus, fine-tuning a foundation model, rather than training it from scratch, reduces training costs.
>
> Next, I will respond to your questions one by one.
>
> **Q1: The source code is not provided.**
>
> We will release the source code upon acceptance of the paper.

---

> ### Author Response · Authors · 2025-11-23
>
> **Q2:why was iForest chosen as the pseudo-labeler? Would the results hold with other detectors?**
>
> In this work, we use iForest as the anomaly detector primarily for its balance between efficiency and effectiveness. We investigate the impact of different support set pseudo-labeling strategies, including a strategy without any detector, on performance. Experiments were conducted on 35 anomaly detection datasets, and we compared three strategies for selecting pseudo-normal samples from unlabeled data:
>
> **1. IForest TopK.**
>
>  IForest is applied to the unlabeled data, and the $K$ samples with the lowest anomaly scores are selected as pseudo-normal. Although these samples have the highest confidence, their limited diversity leads to relatively lower performance.
>
> **2. IForest RandomK 80%.**
>
>  IForest is used again, but this time we retain the lowest 80% of the anomaly scores and then randomly select $K$ pseudo-normal samples from this subset. While introducing some noise, the increased sample diversity significantly improves performance compared to the IForest TopK strategy.
>
> **3. Random.**
>
> As a detector-free baseline, we randomly sample $K$ pseudo-normal points from all unlabeled samples without using any anomaly detector. While some of these samples can be abnormal, this strategy maximizes sample diversity. Interestingly, Random outperforms the other two strategies on average, highlighting the importance of diversity in the support set.
>
> As shown in the following table or appendix F,  ProFiT consistently outperforms  MotherNet under all three strategies. ProFiT consistently improves upon the baseline regardless of the pseudo-labeling strategy used for normal samples. This demonstrates that:
> - The ProFiT model is robust to different pseudo-labeling strategies, showing stable performance even without a specific anomaly detector;
> - The diversity of the support set plays a more crucial role than focusing on the most confident normal samples.
>
> Although we have not tested other anomaly detectors in this work, the consistent performance of ProFiT using the detector-free Random strategy shows **the importance of sample diversity over strict accuracy and the choice of detector is less critical**.
>
> | Method             | F1 MotherNet | F1 ProFiT | PR MotherNet | PR ProFiT |
> |--------------------|--------------|-----------|--------------|-----------|
> | IForest Topk       | 0.5677       | 0.5573    | 0.5904       | 0.6085    |
> | IForest RandomK 80%| 0.6696       | 0.6770    | 0.6954       | 0.7084    |
> | Random             | 0.7014       | 0.7128    | 0.7354       | 0.7475    |
> | Unsupervised       | 0.3402       | 0.3260    | 0.3660       | 0.3610    |

---

> ### Author Response · Authors · 2025-11-23
>
> **Q3: If the labeled true anomalies are removed, can the proposed framework operate in a fully unsupervised setting by relying solely on pseudo-labels generated by an anomaly detector?**
>
> As shown in the table in Q2, we also explored extending our method to an unsupervised approach, where both normal and abnormal samples are pseudo-labeled using iForest. However, this approach resulted in a **significant performance drop**. The main reason is that abnormal samples are much fewer than normal samples in the dataset, and accurate pseudo-labeling of abnormal samples is crucial for guiding the model. When using an unsupervised detector, the labeling accuracy for abnormal samples is too low, which negatively impacts the model’s performance.
>
> In contrast, the larger number of normal samples allows the model to tolerate some noise introduced by the unsupervised pseudo-labeling process without significantly affecting performance. This highlights the importance of accurately labeling abnormal samples for effective model performance. While normal sample diversity plays a significant role in improving robustness, **the quality of abnormal sample labels remains essential for maintaining high model performance.**
>
> **Q4: Why does the current TAD setting only use unlabeled samples predicted as normal during inference, and could incorporating those predicted as anomalous improve robustness and performance?**
>
> Our method is designed for the weakly supervised anomaly detection (WSAD) setting, where only a small number of labeled anomalous samples are available. To remain consistent with existing WSAD protocols, we use only the labeled anomalies during inference.
>
> However, due to the architectural design of the TICL model, the inference stage requires both normal and anomalous samples as inputs. Therefore, we rely on an unsupervised anomaly detector to select samples predicted as normal for constructing the support set.
>
> Incorporating samples predicted as anomalous into the support set is problematic: such samples typically contain a high proportion of pseudo-anomalies, which degrades the quality of the support set and negatively affects model performance. The results in Q2 & Q3 confirm that including pseudo-labeled anomalous samples leads to a **clear performance drop**.

---

> ### Author Response · Authors · 2025-11-23
>
> **Q5: Since the problem is formulated as a binary classification task, why are strong tabular baselines such as XGBoost and CatBoost not included for comprehensive comparison?**
>
> Although the problem can be cast as a binary classification task, our setting falls under weakly supervised anomaly detection, where only a very small number of labeled anomalies are available. In contrast, XGBoost and CatBoost are designed for fully supervised learning and typically require a substantial amount of labeled data to perform well.
>
> For completeness, we additionally evaluated XGBoost and CatBoost under the same data setting as our method. As shown in the following table, both models achieve inferior performance compared with ProFiT, despite the task being formulated as binary classification. This further demonstrates the advantage of our approach in low-label and weak-supervision scenarios.
>
> | Method    | F1 Score     | AUC PR      |
> |-----------|--------------|-------------|
> | XGBoost   | 0.4160  | 0.5459 |
> | CatBoost  | 0.4107  | 0.6721 |
> | MotherNet | 0.6696  | 0.6953 |
> | ProFiT    | 0.6770  | 0.7083 |

---

> ### Author Response · Authors · 2025-11-23
>
> **Q6: What is the performance of this method on other tabular imbalance learning settings?**
>
> Our method is not limited to weakly supervised anomaly detection and can also be applied to other imbalanced tabular learning settings. We are conducting the corresponding experiments. Specifically, we selected several datasets from CC70 for evaluation. The results indicate that our approach can still improve model performance on non-anomaly-detection tabular datasets, suggesting that the method has a certain level of generality.
>
> | Dataset           | F1 MotherNet | F1 ProFiT | PR MotherNet | PR ProFiT |
> |-------------------|--------------|-----------|---------------|-----------|
> | PC4               | 0.3333       | 0.3889    | 0.3687        | 0.4584    |
> | KC2               | 0.4545       | 0.4545    | 0.4304        | 0.4735    |
> | KC1               | 0.3438       | 0.3438    | 0.3672        | 0.4150    |
> | PC1               | 0.4286       | 0.4286    | 0.5169        | 0.5304    |
> | BankMarketing     | 0.4178       | 0.4178    | 0.3786        | 0.3852    |
> | Nomao             | 0.7584       | 0.7422    | 0.7494        | 0.7948    |
> | Dresses Sales     | 0.4762       | 0.6667    | 0.5124        | 0.6795    |
> | Credit Approval   | 0.8462       | 0.7692    | 0.8673        | 0.8688    |
> | Sick              | 0.5217       | 0.6957    | 0.5638        | 0.7406    |
> | Bioresponse       | 0.6029       | 0.6029    | 0.5840        | 0.5974    |
> | Spambase          | 0.6868       | 0.6923    | 0.7911        | 0.8087    |
> | PhishingWebsites  | 0.8920       | 0.8920    | 0.9488        | 0.9563    |
> | Tic Tac Toe       | 0.7460       | 0.7302    | 0.7146        | 0.7327    |
> | **Average**       | 0.5776   | **0.6019**| 0.5995    | **0.6493**|

---

> ### Author Response · Authors · 2025-11-23
>
> **Q7:The theoretical analysis relies on strong assumptions (e.g., latent factor models, task compatibility, conditional independence of columns). How are these assumptions justified or validated under complex real-world tabular data distributions?**
>
>
> To address this concern, we first quantify how well the latent factor assumption holds on each dataset. We introduce a new metric, Latent Factor Strength (LFS), defined as follows:
> 1. We compute the original correlation, i.e., the mean absolute pairwise correlation of the raw feature correlation matrix.
> 2. We then perform Factor Analysis to extract latent factors and reconstruct the features. Based on the reconstruction residuals, we compute the residual correlation, i.e., the mean absolute pairwise correlation of the residual features.
> 3. We define
> $$
> LFS = 1- \frac{\text{residual correlation}}{\text{original correlation}},
> $$
>
> which measures the proportion of the original feature dependence that can be explained by latent factors. A higher LFS indicates that the latent factor assumption is better satisfied.
>
> As shown in the following table or appendix E, we observe a clear relationship between this metric and the performance of our method:
> - On datasets with **strong latent factor structure** (high LFS), our method consistently yields significant improvements, which matches our theoretical expectation.
> - On datasets where the **latent factor strength is moderate or weak but the feature dimensionality is high**, the method still brings noticeable gains. In these cases, the rich feature space allows us to construct informative proxy tasks even though the latent structure is not very clean.
> - Only when **both the latent factor strength is weak and the feature dimensionality is small **do we see our method become less effective. In such datasets, the constructed proxy tasks are less expressive, and the performance may be close to or slightly below the baseline.
>
> Therefore, when the latent factor assumption does not hold strongly, we mainly see **reduced gains** rather than catastrophic degradation, and clear performance drops appear only in the corner case of weak latent factors combined with low-dimensional features.
>
> | Datasets      | Dim  | Original Corr | Residual Corr | LFS    | F1 Impr. | PR Impr. |
> |---------------|------|---------------|---------------|--------|----------|----------|
> | Lymphography  | 18   | 0.1680        | 0.1075        | 36.01% | 0.0278   | 0.0554   |
> | WPBC          | 33   | 0.2905        | 0.1290        | 55.59% | 0.0334   | 0.0435   |
> | Breastw       | 9    | 0.6019        | 0.1372        | 77.21% | 0.0000   | 0.0000   |
> | Skin          | 3    | 0.6961        | 0.4181        | 39.94% | 0.0000   | 0.0000   |
> | InternetAds   | 1555 | 0.0183        | 0.0183        | 0.00%  | 0.0825   | 0.1031   |
> | Backdoor      | 196  | 0.1061        | 0.1392        | -31.20%| 0.0879   | 0.0414   |
> | Census        | 500  | 0.0297        | 0.0273        | 8.08%  | 0.0179   | 0.0199   |
> | OI            | 27   | 0.0946        | 0.0870        | 8.03%  | -0.0017  | 0.0002   |
> | Shuttle       | 9    | 0.1885        | 0.1778        | 5.68%  | 0.0204   | -0.0032  |

---

> ### Author Response · Authors · 2025-11-23
>
> **Q8: How to process numerical columns?**
>
> For numerical columns, we follow the preprocessing strategy used in MotherNet and TabPFN. Specifically, we first sort the numerical values and then randomly select quantile boundaries. The values are then discretized into bins according to these quantile thresholds, converting each numerical column into a categorical attribute.
>
> **Q9: What is the difference between ProFiT and GTL or STUNT?**
>
> GTL and STUNT differ from ProFiT both in motivation and in technical design.
>
> 1. **Difference from GTL.**
>
> GTL constructs generative prediction tasks by randomly selecting table columns and predicting their values. This task formulation is conceptually closer to generative tabular modeling methods such as AnoLLM, and does not aim to adapt a pretrained foundation model to a weakly supervised anomaly detection setting. In contrast, ProFiT focuses on designing proxy tasks specifically for fine-tuning pretrained tabular models in scenarios where labeled downstream data is extremely scarce.
>
> 2. **Difference from STUNT.**
>
> While STUNT and our method share the high-level idea of using pseudo-labeled meta-learning–style training, the problem settings are fundamentally different.
> - STUNT trains a model entirely from scratch, requiring substantial computation.
> - ProFiT adapts an existing pretrained foundation model, which is significantly more efficient and practical for real-world anomaly detection.
>
> The success of STUNT empirically supports the effectiveness of pseudo-task–based adaptation. However, our work goes beyond empirical validation:
>
> - We provide a **theoretical analysis** that explains why proxy-task–based fine-tuning is effective.
> - We further **extend this framework to the fine-tuning of powerful pretrained tabular models**, demonstrating that it yields substantial gains in low-label and imbalance settings.
>
> Thus, although the methods share inspiration at a conceptual level, ProFiT tackles a different problem, uses a different computational regime, and contributes new theoretical understanding and practical extensions to pretrained models.

---

> ### Author Response · Authors · 2025-11-23
>
> **Q10: Why choose MotherNet as the foundation model instead of alternatives like TabPFN?**
>
> We chose MotherNet primarily because, in our preliminary experiments, it offered a substantially **lower inference cost** compared to TabPFN and TabPFNv2 under the TICL architecture. Although MotherNet is more expensive to pretrain, this cost is incurred only once, whereas its lightweight inference makes it attractive for real-world anomaly detection applications.
>
> Our focus in this work is therefore not on pretraining a new foundation model, but on **how to adapt an existing pretrained model to downstream weakly supervised tasks at low cost.** For this reason, most experiments are conducted on MotherNet.
>
> Importantly, ProFiT is not tied to MotherNet. We further applied ProFiT to TabPFNv2 and conducted the corresponding experiments. As shown in the table below, ProFiT can also be effectively used to fine-tune TabPFNv2, leading to notable performance improvements.
>
> | Dataset           | F1 TabPFNv2 | F1 TabPFNv2+ProFiT | PR TabPFNv2 | PR TabPFNv2+ProFiT |
> |-----------------|:-----------:|:-------------------:|:-----------:|:--------------------:|
> | ALOI              | 0.0242      | 0.0703              | 0.0357      | 0.0395               |
> | Annthyroid        | 0.6879      | 0.7110              | 0.7306      | 0.7526               |
> | Breastw           | 0.9452      | 0.9452              | 0.9641      | 0.9736               |
> | Cardio            | 0.8036      | 0.8214              | 0.9047      | 0.9009               |
> | Cardiotocography  | 0.6809      | 0.6950              | 0.7682      | 0.7631               |
> | Census            | 0.3065      | 0.3231              | 0.2590      | 0.2631               |
> | Fault             | 0.5694      | 0.6029              | 0.5857      | 0.6119               |
> | Http              | 0.9971      | 0.9985              | 0.9987      | 0.9986               |
> | InternetAds       | 0.4105      | 0.4421              | 0.3640      | 0.3626               |
> | Letter            | 0.1818      | 0.1818              | 0.1844      | 0.1904               |
> | Mammography       | 0.5882      | 0.6000              | 0.5512      | 0.5516               |
> | Pima              | 0.5823      | 0.6076              | 0.6251      | 0.6325               |
> | Satellite         | 0.7131      | 0.7244              | 0.8487      | 0.8567               |
> | Skin              | 0.7983      | 0.8196              | 0.9084      | 0.8915               |
> | Thyroid           | 0.7778      | 0.8148              | 0.9198      | 0.9169               |
> | Vertebral         | 0.3636      | 0.5455              | 0.3572      | 0.5032               |
> | WPBC              | 0.2667      | 0.2000              | 0.2693      | 0.2970               |
> | Yeast             | 0.4610      | 0.4610              | 0.4577      | 0.4652               |
> | **Average**       | 0.5643      | 0.5869              | 0.5962      | 0.6095               |

---

> ### Author Response · Authors · 2025-11-23
>
> **Q11: Is using 5 labeled anomalies a fair comparison with other detection methods?**
>
> Yes, the comparison is fair. In our paper, all methods are evaluated under the exact same labeling and the same random seeds, ensuring a strictly controlled experimental setup. Therefore, each method receives precisely five labeled anomalies, and no approach has access to additional supervision.
>
> We use a fixed number of labeled anomalies (k = 5) to reflect an extremely weakly supervised setting. While this number corresponds to a non-negligible fraction in a few very small datasets, on the majority of benchmarks, such as Census, Donors, and Skin. The five labeled anomalies represent far less than 0.1% of all anomalies, and sometimes even below 0.01%. This consistent labeling across all methods guarantees a fair comparison.

---

> ### Author Response · Authors · 2025-11-27
>
> **Q12: Why does the paper omit semi-supervised TAD baselines that are commonly evaluated under similar contamination settings?**
>
> To provide a more comprehensive comparison, we have additionally included MCM and DRL as baselines, following the reviewer’s suggestion. We evaluate both methods using their official implementations and recommended hyperparameters. The data splits and contamination levels were aligned with our experimental setup to ensure fairness.
>
> It is worth noting that these semi-supervised approaches are typically designed for settings where only normal samples are available, and naturally perform better when the training data is clean. Under our setting, however, the contamination rate equals the true anomaly ratio of each dataset, which can substantially degrade their performance.
>
> The results show that ProFiT, using only 5 labeled anomalies, outperforms both MCM and DRL on the majority of datasets (see Appendix for full results). This further demonstrates the effectiveness of our proxy-based fine-tuning approach under weak-supervision conditions.
>
> | Dataset | F1 DRL | F1 MCM | F1 ProFiT | PR DRL | PR MCM | PR ProFiT |
> |--------|:--------:|:--------:|:-----------:|:--------:|:--------:|:-----------:|
> | ALOI | 0.0286 | **0.0813** | 0.0308 | 0.0291 | **0.0531** | 0.0382 |
> | Annthyroid | 0.2023 | 0.2428 | **0.7168** | 0.1752 | 0.2196 | **0.6664** |
> | Breastw | 0.8767 | **0.9589** | **0.9589** | 0.9543 | 0.9856 | **0.9902** |
> | Cardio | 0.3214 | 0.4107 | **0.5893** | 0.3125 | 0.4347 | **0.5007** |
> | Cardiotocography | 0.4326 | 0.4894 | **0.6099** | 0.3816 | 0.4290 | **0.7123** |
> | Celeba | **0.1302** | 0.0756 | 0.0455 | **0.0605** | 0.0522 | 0.0302 |
> | Census | 0.0380 | 0.0656 | **0.1608** | 0.0678 | 0.0784 | **0.1403** |
> | Donors | 0.0311 | 0.0331 | **0.8600** | 0.1097 | 0.0862 | **0.9423** |
> | Fault | 0.4545 | **0.4976** | 0.4258 | 0.4756 | **0.4973** | 0.4366 |
> | Http | 0.0161 | 0.0587 | **0.9882** | 0.2509 | 0.6029 | **0.9842** |
> | Ionosphere | **0.8108** | 0.1081 | 0.7838 | **0.8688** | 0.2408 | 0.8367 |
> | Landsat | 0.2481 | 0.1830 | **0.3659** | 0.2580 | 0.2018 | **0.3676** |
> | Letter | **0.3939** | 0.1515 | 0.0303 | **0.4104** | 0.1175 | 0.0751 |
> | Magic | 0.4753 | **0.6408** | 0.4852 | 0.5635 | **0.7203** | 0.4848 |
> | Mammography | 0.2000 | 0.3294 | **0.5412** | 0.1310 | 0.2244 | **0.5183** |
> | Mnist | **0.5519** | 0.4057 | 0.3632 | **0.5544** | 0.3898 | 0.3292 |
> | Optdigits | 0.0000 | 0.0000 | **0.8140** | 0.0256 | 0.0407 | **0.8951** |
> | PageBlocks | 0.2986 | 0.2500 | **0.5417** | 0.3227 | 0.2168 | **0.4816** |
> | Pendigits | 0.0455 | 0.1818 | **0.7273** | 0.0326 | 0.0911 | **0.8186** |
> | Pima | 0.4684 | 0.5316 | **0.6329** | 0.4667 | 0.5193 | **0.6497** |
> | Satellite | 0.5497 | 0.5304 | **0.6619** | 0.5864 | 0.6771 | **0.8066** |
> | Satimage-2 | 0.6087 | 0.7391 | **0.8696** | 0.5295 | 0.7007 | **0.8741** |
> | Shuttle | 0.9159 | 0.9039 | **0.951** | 0.8728 | 0.7982 | **0.9736** |
> | Skin | 0.1923 | 0.0022 | **0.7826** | 0.2504 | 0.1595 | **0.7938** |
> | Smtp | **0.6154** | **0.6154** | 0.4615 | 0.4399 | **0.5982** | 0.4331 |
> | SpamBase | 0.5172 | 0.5960 | **0.7636** | 0.5508 | 0.6046 | **0.8387** |
> | Thyroid | 0.3704 | 0.4444 | **0.8889** | 0.3293 | 0.3547 | **0.9236** |
> | Vertebral | 0.0000 | 0.0000 | **0.6364** | 0.1589 | 0.1270 | **0.6621** |
> | WBC | 0.3333 | 0.0000 | **1.0000** | 0.425 | 0.2619 | **1.0000** |
> | WDBC | 0.6667 | **1.0000** | **1.0000** | 0.7292 | **1.0000** | **1.0000** |
> | Wilt | 0.1190 | 0.0000 | **0.5714** | 0.1162 | 0.0399 | **0.4468** |
> | Wine | 0.0000 | **1.0000** | **1.0000** | 0.2250 | **1.0000** | **1.0000** |
> | WPBC | **0.3333** | 0.2000 | **0.3333** | 0.3444 | 0.3076 | **0.4325** |
> | Yeast | 0.3312 | 0.3052 | **0.3701** | 0.3181 | 0.3090 | **0.3710** |
> | Average | 0.3405 | 0.3539 | **0.6165** | 0.3626 | 0.3865 | **0.6310** |

---

### Official Review · Reviewer_KCY4 · 2025-11-01

**Soundness:** 3
**Presentation:** 4
**Contribution:** 3
**Rating:** 6
**Confidence:** 4

**Summary:**

ProFiT is an effective unsupervised fine-tuning framework for tabular anomaly detection under limited labels, leveraging proxy tasks to capture feature relationships. The method is theoretically grounded and empirically strong, though its reliance on certain assumptions and hyperparameter sensitivity warrants further clarification.

**Strengths:**

This paper tackles an important challenge in anomaly detection under label scarcity. The proposed method, ProFiT, uses unlabeled data to create proxy tasks by predicting a held-out feature from correlated subsets. The feature selection strategy (based on correlation and redundancy) is well-motivated, and the addition of a JS-consistency loss across proxy subsets helps regularize training.
The theoretical justification is thoughtful. By linking proxy-task performance to downstream generalization through a latent factor model, the authors provide a clear intuition for why their method should work.
Empirical results are strong. Across 35 tabular datasets, ProFiT consistently outperforms both classical and recent weakly-supervised baselines. Gains in AUCPR and F1 are meaningful, especially with as few as 5 labeled anomalies.
Presentation-wise, the paper is cleanly written and structured. Figures and appendices help clarify the method and setup. The approach is practical and applicable in settings like fraud or risk monitoring, where labeled anomalies are scarce.

**Weaknesses:**

The theory relies on a strong latent factor assumption. While it helps the analysis, it's unclear how realistic this is for complex tabular data. A short discussion would help.
ProFiT assumes access to a pre-trained tabular model (e.g., MotherNet). It’s unclear how well the method works without it, or whether it’s portable to other backbones.
Hyperparameters like the subset size k or consistency loss weight λ aren’t thoroughly explored. Ablation or guidance would improve reproducibility.
The final inference step uses iForest to mine pseudo-normals. While practical, it introduces an unsupervised heuristic that could fail in noisy settings. Some robustness analysis would help.
The paper does not compare to simpler alternatives like masked prediction which could serve as baseline proxy-task learners.

**Questions:**

1. What happens if the latent factor assumption doesn't hold? Do you observe degraded performance in such settings?

2. Did you test different anomaly detectors besides iForest for support set selection? Does performance vary?

3. Can your method operate without any labeled anomalies (i.e., K=0)? Could proxy-task variance or consistency be used to rank anomalies?

4. What were your actual choices for k and M, and how sensitive is performance to them?

5. How does the base TICL model (e.g., MotherNet) perform without ProFiT? Including this would help isolate the fine-tuning gain.

---

> ### Author Response · Authors · 2025-11-23
>
> Thanks for your questions ! I will respond to your questions one by one.
>
> **Q1: What happens if the latent factor assumption doesn't hold? Do you observe degraded performance in such settings?**
>
> To address this concern, we first quantify how well the latent factor assumption holds on each dataset. We introduce a new metric, Latent Factor Strength (LFS), defined as follows:
> 1. We compute the original correlation, i.e., the mean absolute pairwise correlation of the raw feature correlation matrix.
> 2. We then perform Factor Analysis to extract latent factors and reconstruct the features. Based on the reconstruction residuals, we compute the residual correlation, i.e., the mean absolute pairwise correlation of the residual features.
> 3. We define
> $$
> LFS = 1- \frac{\text{residual correlation}}{\text{original correlation}},
> $$
>
> which measures the proportion of the original feature dependence that can be explained by latent factors. A higher LFS indicates that the latent factor assumption is better satisfied.
>
> As shown in the following table or appendix E, we observe a clear relationship between this metric and the performance of our method:
> - On datasets with **strong latent factor structure** (high LFS), our method consistently yields significant improvements, which matches our theoretical expectation.
> - On datasets where the **latent factor strength is moderate or weak but the feature dimensionality is high**, the method still brings noticeable gains. In these cases, the rich feature space allows us to construct informative proxy tasks even though the latent structure is not very clean.
> - Only when **both the latent factor strength is weak and the feature dimensionality is small **do we see our method become less effective. In such datasets, the constructed proxy tasks are less expressive, and the performance may be close to or slightly below the baseline.
>
> Therefore, when the latent factor assumption does not hold strongly, we mainly see reduced gains rather than catastrophic degradation, and clear performance drops appear only in the corner case of weak latent factors combined with low-dimensional features.
>
> | Datasets      | Dim  | Original Corr | Residual Corr | LFS    | F1 Impr. | PR Impr. |
> |---------------|------|---------------|---------------|--------|----------|----------|
> | Lymphography  | 18   | 0.1680        | 0.1075        | 36.01% | 0.0278   | 0.0554   |
> | WPBC          | 33   | 0.2905        | 0.1290        | 55.59% | 0.0334   | 0.0435   |
> | Breastw       | 9    | 0.6019        | 0.1372        | 77.21% | 0.0000   | 0.0000   |
> | Skin          | 3    | 0.6961        | 0.4181        | 39.94% | 0.0000   | 0.0000   |
> | InternetAds   | 1555 | 0.0183        | 0.0183        | 0.00%  | 0.0825   | 0.1031   |
> | Backdoor      | 196  | 0.1061        | 0.1392        | -31.20%| 0.0879   | 0.0414   |
> | Census        | 500  | 0.0297        | 0.0273        | 8.08%  | 0.0179   | 0.0199   |
> | OI            | 27   | 0.0946        | 0.0870        | 8.03%  | -0.0017  | 0.0002   |
> | Shuttle       | 9    | 0.1885        | 0.1778        | 5.68%  | 0.0204   | -0.0032  |

---

> ### Author Response · Authors · 2025-11-23
>
> **Q2: Did you test different anomaly detectors besides iForest for support set selection? Does performance vary?**
>
> In this work, we use iForest as the anomaly detector primarily for its balance between efficiency and effectiveness. We investigate the impact of different support set pseudo-labeling strategies, including a strategy without any detector, on performance. Experiments were conducted on 35 anomaly detection datasets, and we compared three strategies for selecting pseudo-normal samples from unlabeled data:
> 1. IForest TopK.
>
>  IForest is applied to the unlabeled data, and the $K$ samples with the lowest anomaly scores are selected as pseudo-normal. Although these samples have the highest confidence, their limited diversity leads to relatively lower performance.
>
> 2. IForest RandomK 80%.
>
>  IForest is used again, but this time we retain the lowest 80% of the anomaly scores and then randomly select $K$ pseudo-normal samples from this subset. While introducing some noise, the increased sample diversity significantly improves performance compared to the IForest TopK strategy.
>
> 3. Random.
>
> As a detector-free baseline, we randomly sample $K$ pseudo-normal points from all unlabeled samples without using any anomaly detector. While some of these samples can be abnormal, this strategy maximizes sample diversity. Interestingly, Random outperforms the other two strategies on average, highlighting the importance of diversity in the support set.
>
> As shown in the following table or appendix F,  ProFiT consistently outperforms  MotherNet under all three strategies. ProFiT consistently improves upon the baseline regardless of the pseudo-labeling strategy used for normal samples. This demonstrates that:
> - The ProFiT model is robust to different pseudo-labeling strategies, showing stable performance even without a specific anomaly detector;
> - The diversity of the support set plays a more crucial role than focusing on the most confident normal samples.
>
> Although we have not tested other anomaly detectors in this work, the consistent performance of ProFiT using the detector-free Random strategy shows the importance of sample diversity over strict accuracy and the choice of detector is less critical.
>
> | Method             | F1 MotherNet | F1 ProFiT | PR MotherNet | PR ProFiT |
> |--------------------|--------------|-----------|--------------|-----------|
> | IForest Topk       | 0.5677       | 0.5573    | 0.5904       | 0.6085    |
> | IForest RandomK 80%| 0.6696       | 0.6770    | 0.6954       | 0.7084    |
> | Random             | 0.7014       | 0.7128    | 0.7354       | 0.7475    |
> | Unsupervised       | 0.3402       | 0.3260    | 0.3660       | 0.3610    |

---

> ### Author Response · Authors · 2025-11-23
>
> **Q3: Can your method operate without any labeled anomalies (i.e., K=0)?**
>
> As shown in the table in Q2. We also explored extending our method to an unsupervised approach, where both normal and abnormal samples are pseudo-labeled using iForest. However, this approach resulted in a significant performance drop. The main reason is that abnormal samples are much fewer than normal samples in the dataset, and accurate pseudo-labeling of abnormal samples is crucial for guiding the model. When using an unsupervised detector, the labeling accuracy for abnormal samples is too low, which negatively impacts the model’s performance.
>
> In contrast, the larger number of normal samples allows the model to tolerate some noise introduced by the unsupervised pseudo-labeling process without significantly affecting performance. This highlights the importance of accurately labeling abnormal samples for effective model performance. While normal sample diversity plays a significant role in improving robustness, the quality of abnormal sample labels remains essential for maintaining high model performance.

---

> ### Author Response · Authors · 2025-11-23
>
> **Q4: What were your actual choices for $K$ and $M$, and how sensitive is performance to them?**
>
> In our setting, $K$ denotes the number of labeled examples used at inference time. The results reported in Table 1 correspond to $K = 5$. To further analyze the sensitivity to $K$, Table 2 presents the F1 and AUC-PR scores under different choices of $K$. Overall, increasing $K$ provides the model with more reliable labeled guidance and leads to consistent performance gains.
>
> Regarding $M$, it refers to the number of proxy tasks sampled during fine-tuning. In our implementation, $M$ is not a fixed hyperparameter; instead, for each training iteration we dynamically sample proxy tasks from the original table. Thus, the effective number of sampled proxy tasks corresponds to the total number of training iterations. This design allows the model to see a diverse set of proxy tasks without requiring manual tuning of $M$.

---

> ### Author Response · Authors · 2025-11-23
>
> **Q5: How does the base TICL model (e.g., MotherNet) perform without ProFiT? Including this would help isolate the fine-tuning gain.**
>
> Figure 3 in our paper already reports the performance gap between the original MotherNet and the MotherNet fine-tuned with ProFiT. As shown, ProFiT consistently improves performance (F1 score and AUC-PR) across most datasets compared to the base model without fine-tuning.
>
> **Q6: Why not compare to simpler alternatives like masked prediction?**
>
> Our work focuses on unsupervised fine-tuning of an existing pretrained model rather than retraining a new one from scratch. Therefore, when designing the proxy task, we must ensure that the task formulation can be directly integrated into the TICL framework. Many masked-prediction–based alternatives, such as MCM, are not compatible with TICL because their training objectives require architectural modifications or supervision signals that TICL does not support. For this reason, these methods cannot be adopted as fair or feasible baselines for our setting.

---

### Meta-Review · Area_Chair_Sb51 · 2025-12-30

**Summary:**

This paper presents a novel unsupervised fine-tuning method of tabular models for label-scarce anomaly detection. The method -Profit- uses only unlabeled target data to adjust pretrained tabular models for anomaly detection. The methods consists in building multiple proxy tasks by sampling different features as targets and  using correlation measures to capture the structure of target data. A regularization aligning predictions from two different proxy views is considered by means of the Jensen-Shannon divergence. An experimental evaluation on tabular anomaly detection benchmarks shows the effectiveness of the method on weakly-supervised and unsupervised setting.

Based on the initial reviews, the following comments were raised:
-Reviewer KCY4 indicates that the paper addresses an important challenge in anomaly detection when labels are scarce. The method is well-motivated. Empirical results obtained on 35 datasets are strong. The paper is clearly written and structured.
Among the weaknesses, the reviewer mentions that the method relies on a strong latent factor assumption and the paper lacks a discussion on how realistic the assumption is for complex tabular data. The adaptation of the method to other models or even without pre-trained models is unclear. The experimental evaluation did not explore that much the possible hyper-parameters values and ablation would have been welcomed. Some heuristics of the non use of simple bases are not justified.

-Bf7X identifies among the strengths that the task is important and that the paper structure is clear.
Among the weaknesses, the reviewer mentions: reproducibility issues (code not provided), the justification of using only the unlabeled samples as normal is not convincing, other anomaly detection models should have been considered (e.g. XGBoost), the impact of true anomalies removal is not discussed, the specificity of use with pre-trained models not sufficiently discussed, the assumptions taken by the method aren't sufficiently justified, how the method would handle continuous feature in the column selection process is unclear, the column selection approach has been used by previous methods, the foundation model used was not justified, in some highly imbalanced datasets the choice of the number of labeled anomalies might be a strong bias.

-J37h identifies as strengths: stronger adaptability to practical scenarios, resolution of distribution misalignment with proxy based approach, regularization with Jensen-Shannon term for improving robustness.
On the other hand, he mentions the following weaknesses: dependence on inter-features correlations, out-of-distribution anomalies are not addressed, high computational costs.

-rJk6 identifies 4 strengths: paper well written, experiments sound and extensive, ablation studies limited but relevant, paper theoretically based and well-founded.
On the other hand, as weaknesses: presence of a few typos, key information missing for the main text of the paper, the proposed pretext task is a special case of mask reconstruction, work not currently reproducible (code not provided)


Overall, the paper proposes an interesting contribution to tabular anomaly detection. The method is theoretically founded, supported by theoretical results with large experimental evaluation. On the other hand the paper misses some justification on the assumption of the methods, the application to other pre-trained models and other settings, plus some paper improvement. The code is not provided.
During rebuttal, authors have addressed many concerns of the reviewers with additional justifications and experiments, event not all the issues were addressed/answered.
The paper is borderline can be accepted if there is enough room.
However, the code is not provided, which makes the paper below than other submissions for reproducibility, there are still some concerns and the paper should worth another round of reviews.
I propose rejection.


Additional note: the reference of [(Shanmugam, 2001) for task] in Assumption 3.1 on latent factor model (line 213) appears to be completely false since the cited paper refers to a small review on Pearl's book without any mention on latent factor model.

**Reviewer Concerns:**

For reviewer KCY4, authors have addressed all his questions. They provided in particular an answer on the impact of the latent factor assumption which I guess answer one important weakness raised by the reviewer. Thy also provide an answer on the use of the iForest heuristic and non-use of simple models. However, issues related to the hyper-parameter exploration, and use with other models were not fully addressed in my opinion.

-For reviewer Bf7X, authors have provided a detailed answer. They provide justifications on the issues related to the use of other baseless (like XGBoost with additional experiment), they explain why they consider the specific pre-trained models setting, provided justification on the assumption on the latent factor - task similarity is not really discussed. They also explain the selection for continuous feature column, the choice of one foundation model but provided additional experiment with another one and the use of the model to other tasks, the choice of the number anomalies. The answer is globally complete, but the lack of code, the explanation of the impact of the use of abnormal samples of good quality might be an assumption that deserves mode discussion.

-For reviewer J37h, authors answer to the 3 weaknesses by providing the same answer on the inter-feature correlation as with the other reviewer, they answer that the method can generalize to OOD detection but this is not supported by experiments as far as I can see in the answer, and that the method does not imply a high computational costs. Two questions were asked during the reviews, but these two questions were not addressed by the authors in the rebuttal.

 -For reviewer rJK6, authors have provided the same answer for the latent factor assumption, they have taken the remarks of structure and typos of the reviewer into account and provided answers on the comparison with mask reconstruction tasks and extension to other setups with additional results. The reviewer mentioned in the discussion other recent papers to take into account but most of them were not added by the authors in the revision. Overall, the reviewer indicated that authors have addressed most of his concerns

**Reviewer Scores:**

KCY4 provided a score of 6, some of his important issues were a priori addressed but not of them. The reviewer should have at least kept his score, it not clear for me if he would have increased it.

Bf7X gave a score of 4, I think he could have increased his score to 5 but not higher.

J37h gave a score of 2. Authors have a doubt that this review was generated by an LLM, this is maybe possible, but the authors did not justify why they did not answer to the 2 questions asked in the review. At least one strong weakness has been addressed for me (inter-feature correlation). The reviewer might increase his score but probably not higher than a 4 due to the absence of answer to some questions and limited answers to weakness 2.

-rJK6 increased gave a 6 and indicated that authors addresses most of his concerns that he would updated his score.

---

### Decision · Program_Chairs · 2026-01-26

Reject